# Context Clues: Evaluating Long Context Models for Clinical Prediction Tasks on EHRs

**Michael Wornow**[*1], **Suhana Bedi**[*1], **Miguel Angel Fuentes Hernandez**[1], **Ethan Steinberg**[12],
**Jason Alan Fries**[1], **Christopher Ré**[1], **Sanmi Koyejo**[1], **Nigam H. Shah**[1]
[1]Stanford University [2]Prealize Health

## Abstract

Foundation Models (FMs) trained on Electronic Health Records (EHRs) have achieved state-of-the-art results on numerous clinical prediction tasks. However, most existing EHR FMs have context windows of <1k tokens. This prevents them from modeling full patient EHRs which can exceed 10k's of events. Recent advancements in subquadratic long-context architectures (e.g., Mamba) offer a promising solution. However, their application to EHR data has not been well-studied. We address this gap by presenting the first systematic evaluation of the effect of context length on modeling EHR data. We find that longer context models improve predictive performance – our Mamba-based model surpasses the prior state-of-the-art on 9/14 tasks on the EHRSHOT prediction benchmark. For clinical applications, however, model performance alone is insufficient – robustness to the unique properties of EHR is crucial. Thus, we also evaluate models across three previously underexplored properties of EHR data: (1) the prevalence of "copy-forwarded" diagnoses which creates artificial repetition of tokens within EHR sequences; (2) the irregular time intervals between EHR events which can lead to a wide range of timespans within a context window; and (3) the natural increase in disease complexity over time which makes later tokens in the EHR harder to predict than earlier ones. Stratifying our EHRSHOT results, we find that higher levels of each property correlate negatively with model performance (e.g., a 14% higher Brier loss when making predictions for the most versus least irregular patients), but that longer context models are more robust to more extreme levels of these properties. Our work highlights the potential for using long-context architectures to model EHR data, and offers a case study for identifying new challenges in modeling sequential data motivated by domains outside of natural language. We release our model checkpoints and code at: https://github.com/som-shahlab/long_context_clues

## 1 Introduction

Foundation Models (FMs) (Bommasani et al., 2021) trained on Electronic Health Records (EHRs) have achieved state-of-the-art results on numerous clinical prediction tasks (Odgaard et al., 2024; Yang et al., 2023). Such models can improve patient outcomes via early detection of disease and risk stratification (Steinberg et al., 2023). As an EHR is simply a list of chronologically-ordered clinical events (see Figure 1a), it can be modeled as a sequence of tokens. Instead of subwords or image patches, however, tokens represent clinical events like diagnoses and procedures (McDermott et al., 2023). This approach has enabled the application of transformer architectures originally developed for natural language processing (NLP) such as BERT (Rasmy et al., 2021; Li et al., 2020; Odgaard et al., 2024) and GPT (Steinberg et al., 2021; Pang et al., 2024; Kraljevic et al., 2024) to EHR data.

A critical choice in FM design is context length – i.e. how many tokens of input the model can ingest. Longer context lengths have shown a consistent positive impact on FM performance across various domains by enabling models to reference and reason over more information (Xiong et al., 2023). Given the typical hospital's limited compute resources, however, transformer-based EHR FMs have been limited to processing short context lengths (i.e., 512 tokens) due to the quadratic scaling of attention with input length (Vaswani et al., 2017). As a single patient's EHR can contain 10k's of tokens, this greatly limits the amount of data that EHR FMs can consider. This is especially

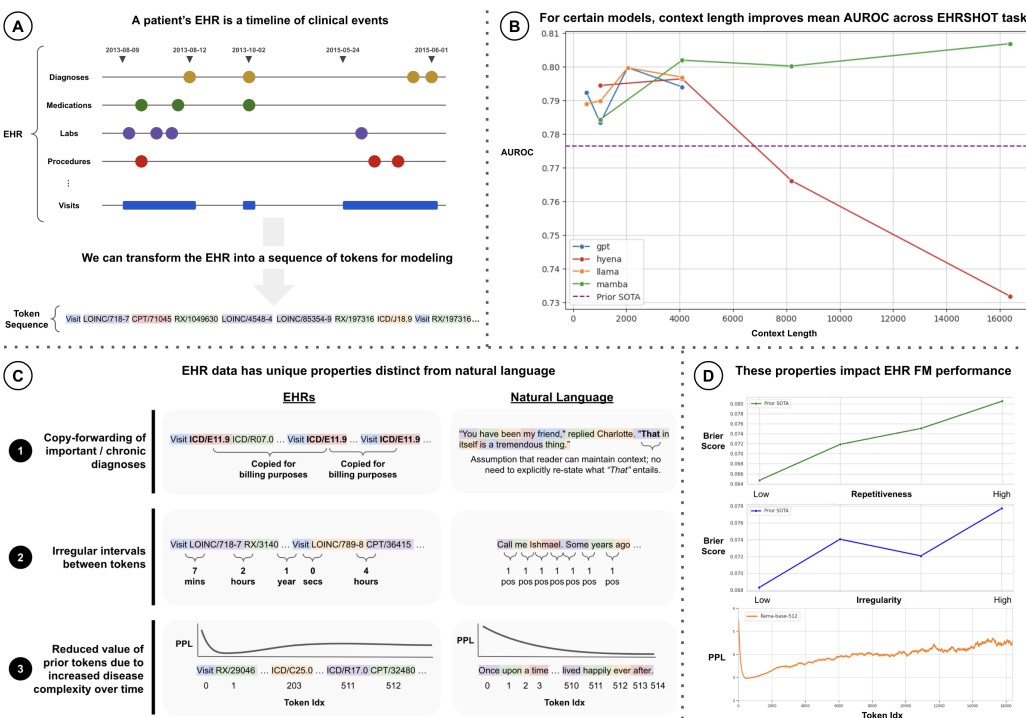

Figure 1: The central claims of this paper. **(a) EHRs are sequences:** An EHR is simply a timeline of clinical events that occur to a patient, and thus can be naturally represented as a sequence of tokens. **(b) Long context improves performance:** AUROC on clinical prediction tasks tends to increase with longer context lengths, with Hyena (red) being the notable exception. Overall, Mamba (green) at a context length of 16k achieves the highest average AUROC across 14 diverse clinical prediction tasks. **(c) EHR data has distinct properties:** In contrast to natural language, EHR data has unique properties whose implications remain under-explored in the ML literature. Here, we highlight three such attributes – copy-forwarding, irregular time intervals between tokens, and disease progression. **(d) EHRs properties present unique modeling challenges:** Stratifying patients by the degree to which they exhibit each EHR-specific property, we find that higher Brier scores (i.e., worse model performance) are associated with patients who have more repetitive (top) or irregular (middle) EHRs. Additionally, the perplexity of tokens later in a patient's timeline tends to be higher, even when conditioning on prior tokens (bottom).

true for the sickest patients – i.e. the ones of most interest to a hospital for prediction tasks – as they typically have high healthcare utilization and thus have very long timelines, as can be seen in the CDF plots of patient sequence length in Appendix Figure 6.

Recently developed *subquadratic* architectures such as Mamba (Gu & Dao, 2024) and Hyena (Poli et al., 2023a) that are optimized for long contexts offer a potential solution. As EHR FMs begin driving real-world care decisions, it is essential to better understand the implications of adapting these long context architectures for clinical prediction making.

However, their effectiveness on EHR data remains unclear. In contrast to natural language, EHR data exhibits specific types of token repetition and noise that complicate the expected benefits of longer contexts. We identify and present the first quantitative analysis of three such underexplored properties, as outlined in Figure 1c:

1. **Copy-forwarding** — key diagnoses are repeated across multiple visits due to billing practices, leading to artificial repetition of tokens in the EHR (Thornton et al., 2013).

2. **Irregular time intervals between tokens** — unlike in natural language where consecutive tokens are trivially 1 position apart, consecutive clinical events can be days or years apart, thus creating a wide range of timescales within a single context (McDermott et al., 2023).

3. **Disease progression** — later tokens in a patient's timeline are harder to predict as disease complexity tends to increase with age (Fabbri et al., 2015), even when conditioning on

prior tokens; this contrasts with natural language, in which later tokens in a prompt tend to exhibit lower perplexities (Peng et al., 2023b).

While several papers have introduced transformer-based EHR FMs, they typically only evaluate at a single context length of 512 tokens, as shown in Table 1. Evaluations of subquadratic architectures on EHR data have also been limited to one context length and do not consider "longitudinal" (i.e. full-length) EHRs (Fallahpour et al., 2024). To our knowledge, there has been no systematic evaluation of the impact of context length on state-of-the-art transformer and non-transformer architectures trained on longitudinal EHR data for clinical prediction tasks.

To address these gaps in the literature, our paper makes the following three contributions:

- **State-of-the-art (SOTA) Clinical Prediction Making with Subquadratic Architectures:** We train and evaluate two transformer-based – GPT (Brown et al., 2020) and Llama (Team, 2024) – and two subquadratic – Mamba (Gu & Dao, 2024) (state space models) and Hyena (Poli et al., 2023a) (long convolutions) – architectures. We are among the first to train the latter three at the scale of millions of patients' EHRs. *We achieve SOTA AUROC scores on 9/14 tasks from the EHRSHOT clinical prediction benchmark using a Mamba-based model*. These results highlight the potential for subquadratic models to process EHR data.

- **Increased Performance with Longer Contexts:** We evaluate the impact of context length (ranging from 512 to 16k tokens) on 14 clinical risk prediction tasks. As shown in Figure 1b, *model performance tends to increase with longer contexts* (with the exception of Hyena, whose performance degrades sharply). While we observe smaller gains than in other fields, these results represent a first step towards improved clinical prediction making by leveraging larger amounts of medical history.

- **Quantifying Difficulties in Modeling EHRs v. Natural Language:** Beyond AUROC, we measure how 3 EHR-specific properties — copy-forwarding, irregular inter-token time intervals, and disease progression — impact models at different context lengths. As shown in Figure 1d, *these EHR-specific properties negatively correlate with model performance*, e.g., patients with the most irregular timelines achieve a Brier score 14% worse than patients with the least irregular timelines. However, we find that *longer context models are more robust* to patients exhibiting higher degrees of these properties.

Our work aims to realize the benefits of long context models in healthcare. More broadly, as sequence modeling architectures designed for natural language are increasingly applied to external domains such as molecular sequences (Nguyen et al., 2023a; 2024), climate (Bodnar et al., 2024; Nguyen et al., 2023b), and time series (Cohen et al., 2024), we hope our analysis serves as a general blueprint for taking a data-centric lens on adapting such models for non-NLP domains. We **release the full weights of our pretrained models on HuggingFace and our code at the Github repo here**: https://github.com/som-shahlab/long_context_clues

## 2 BACKGROUND

In this section, we motivate the application of long-context foundation models to electronic health record data and summarize related work.

### 2.1 FOUNDATION MODELS FOR EHRS

Foundation Models (FMs) are large-scale deep learning models trained on extensive amounts of unlabeled data via unsupervised learning (Bommasani et al., 2021). An electronic health record (EHR) provides comprehensive documentation of patient interactions with the healthcare system, including diagnoses, medications, procedures, lab results, etc. (Ambinder, 2005). In this work, we only consider **structured EHR data** – i.e. we ignore notes and images – as structured EHR data is simpler to deidentify and thus share with the community for open science (Negash et al., 2023).

As seen in Table 1, many architectures for sequence modeling have been re-applied to EHR data. Most utilize transformer-based architectures such as BERT (Devlin et al., 2019) or GPT (Brown et al., 2020) with a context length of 512. Pretrained on millions of EHRs using objectives such as

causal or masked language modeling, these EHR FMs are state-of-the-art on many clinical prediction tasks (Yang et al., 2023; Odgaard et al., 2024; Wornow et al., 2023).

| Model | Context Length(s) | Architecture(s) | Subquadratic? |
|---|---|---|---|
| CEHR-BERT (Pang et al., 2021) | 300 | BERT | |
| Med-BERT (Rasmy et al., 2021) | 512 | BERT | |
| BEHRT (Li et al., 2020) | 512 | BERT | |
| CORE-BEHRT (Odgaard et al., 2024) | 512 | BERT | |
| ForeSight (Kraljevic et al., 2024) | 256 | GPT | |
| CLMBR (Steinberg et al., 2021) | 512 | GPT | |
| CEHR-GPT (Pang et al., 2024) | 512 | GPT | |
| ETHOS (Renc et al., 2024) | 2048 | GPT | |
| TranformEHR (Yang et al., 2023) | 512 | T5 | |
| MOTOR (Steinberg et al., 2023) | 512 | Custom | |
| UniHPF (Hur et al., 2024b) | 8192 | Custom | |
| GenHPF (Hur et al., 2024a) | 8192 | Custom | |
| EHRMamba (Fallahpour et al., 2024) | 2048 | Mamba | ✓ |
| Our Work | 512 - 16,384 | Mamba, Llama, Hyena, GPT | ✓ |

Table 1: Comparison to prior work on sequence modeling for EHR data

## 2.2 LONG CONTEXT FMS

Context length is the number of input tokens that a model can ingest. Longer contexts have shown to positively impact FM performance by enabling models to reason over more information (Xiong et al., 2023). Token-level perplexity typically decreases as context length increases, reflecting improved model comprehension of longer sequences (Press et al., 2022; Chen et al., 2023; Peng et al., 2023b).

Theoretically, conditioning on more of a patient's medical history should also enable better clinical decisions. Unfortunately, transformers scale quadratically with context length (Vaswani et al., 2017), which makes processing long sequences computationally expensive. This is an especially important consideration for resource-constrained hospitals hoping to deploy such models. To remedy this, *subquadratic* architectures optimized for processing longer contexts have been proposed (Tay et al., 2020; Wang et al., 2024). They replace the $O(n^2)$ attention mechanism in transformers with linear or log-linear alternatives such as state space models (Gu & Dao, 2024; Goel et al., 2022), long convolutions (Poli et al., 2023a), linear attention (Peng et al., 2023a; Katharopoulos et al., 2020), or recurrent subunits (De et al., 2024). Despite strong results in NLP (Xu, 2024) and biology (Nguyen et al., 2023a), these architectures remain largely untested on EHR data.

## 2.3 RELATED WORK

The impact of context length on EHR FMs for clinical prediction tasks remains largely unexplored. Many papers have evaluated the trade-offs of BERT (Odgaard et al., 2024; Rasmy et al., 2021; Li et al., 2020) and GPT-based (Kraljevic et al., 2024; Pang et al., 2024) architectures on EHR data. However, they typically only consider one context length up to 512 tokens. In contrast, our work examines the impact of multiple context lengths up to 16,384 tokens.

These works also do not consider state-of-the-art subquadratic architectures. To our knowledge, only one work – EHRMamba (Fallahpour et al., 2024) – has done so. However, the authors only consider a single context length of 2048, and do not train or evaluate on longitudinal (i.e. full-length) EHRs, instead focusing on the more limited ICU setting. In contrast, our work evaluates Mamba (Gu & Dao, 2024) on 8x longer context lengths and longitudinal EHR tasks.

Several studies have combined fixed context window transformers with a preliminary retrieval step that selects the most relevant events across a patient's entire timeline (Kim et al., 2023; Zhu et al., 2024). However, they only consider fixed context windows and benchmark against weaker long context models such as S4 (Gu et al., 2022) and Performer (Choromanski et al., 2022).

## 3 METHODS

Our goal is to measure how non-transformer architectures, context length, and the unique properties of EHR data impact performance on clinical prediction tasks. We pretrain 16 models across four

architectures and six context lengths on the structured EHR data of 2.5M patients. We evaluate each model on 14 binary classification tasks from the EHRSHOT benchmark (Wornow et al., 2023), as detailed in Section 3.2 We stratify our results on the degree to which each patient exhibits 3 EHR-specific properties – token repetition due to copy-forwarding, irregularity of time intervals between tokens, and increased complexity of tokens due to disease progression – which we hypothesize may influence the efficacy of longer context models.

## 3.1 MODEL TRAINING

Here, we provide details on our training dataset, tokenization strategy, and model architectures.

### 3.1.1 PROBLEM SETUP

In this paper, we focus exclusively on the structured data within a longitudinal (i.e. full-length) EHR – i.e., diagnoses, medications, lab tests, procedures, visits, and other observational data. Our dataset consists of $n$ patients $X = \{X_1, ..., X_n\}$. For each patient $i$ we have their structured EHR data $X_i$, which is composed of a sequence of chronologically ordered clinical events $X_{ij}$:

$$X_i = \{X_{i1}, X_{i2}, ..., X_{i|X_i|}\}$$

We refer to $X_i$ as a "patient timeline", where each clinical event is a tuple of the form $(t_{ij}, c_{ij}, v_{ij})$. Here, $t_{ij}$ is the timestamp, $c_{ij} \in \mathcal{C}$ is a medical code drawn from a fixed medical ontology ($\mathcal{C}$), and $v_{ij} \in \mathcal{V}_c \cup \mathcal{V}_n \cup \emptyset$ is an optional value, either categorical ($\mathcal{V}_c$) or numeric ($\mathcal{V}_n$):

$$X_{ij} = (t_{ij}, c_{ij}, v_{ij})$$

Events are sorted by time such that $t_{ij} \leq t_{i(j+1)} \forall j$. This formulation of EHR data is also referred to as the "event stream format" (McDermott et al., 2023).

For our experiments, we use a dataset of deidentified longitudinal EHRs sourced from an academic medical center that have been formatted under the OMOP Common Data Model (Sciences & Informatics, 2021). We refer to this dataset as **EHR-OMOP**. We use 2.5M patients (covering 3.5B clinical events) for training, and hold out 0.5M patients as a validation set. The average patient has 1,364 total and 237 unique events. Additional information can be found in Appendix Section A.

### 3.1.2 TOKENIZATION

Given a patient timeline $X_i$, we must convert it into a sequence of tokens $T_i$ that our models can ingest. Thus, we must map each $X_{ij} = (t_{ij}, c_{ij}, v_{ij})$ to some set of token(s) $T_{ij} = \{T_{ij1}, ..., T_{ijk}\}$. We use the same vocabulary used by the prior SOTA model on the benchmark we use for evaluation, EHRSHOT (Wornow et al., 2023). Each clinical "event" in a patient's timeline has a single "code" associated with it. Each "code" then gets converted into a single "token" within our vocabulary via the following process. First, all unique codes $c \in \mathcal{C}$ that occur at least once in our training dataset are assigned a unique token. Second, all codes that are associated with categorical values are assigned a unique token for each possible associated categorical value. Third, all codes associated with numerical values are assigned a unique token for each decile within the range of values attained in our training dataset. After sorting all tokens by their information content, the top 39811 tokens were kept as our vocabulary, and all models share this same vocabulary. Please see Appendix Section D for additional details on the token generation and selection process.

### 3.1.3 ARCHITECTURES

We evaluate four models – GPT (Brown et al., 2020), Llama (Team, 2024), Mamba (Gu & Dao, 2024), and Hyena (Poli et al., 2023a) – at the 120 million parameter scale using their default HuggingFace implementations. (see Appendix Section C for details on each architecture and Appendix Table 6 for exact configurations). We evaluate each model across various context lengths $L \in \mathcal{L}$, with $\mathcal{L} = \{512, 1k, 2k, 4k\}$ for the transformer-based models (GPT and Llama) and $\mathcal{L} = \{1k, 4k, 8k, 16k\}$ for the subquadratic models (Mamba and Hyena). The ranges are different given the poor computational scaling of transformers and our limited compute.

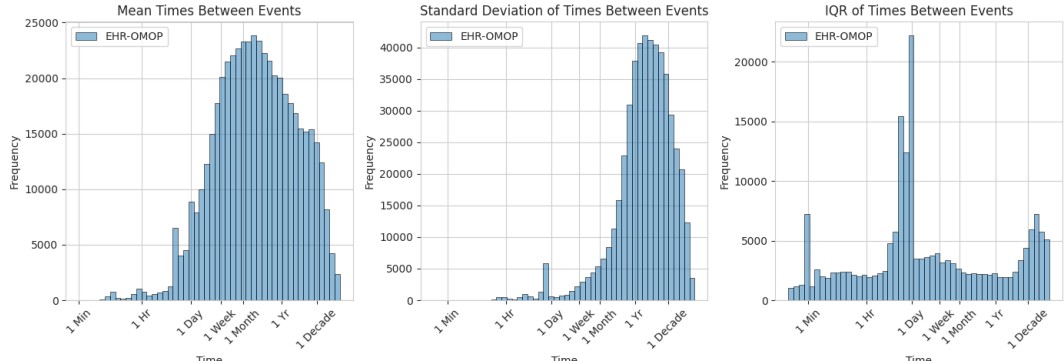

Figure 2: EHR data exhibits a high degree of variation in time intervals between events. From left to right, we measure the mean, standard deviation, and inter-quartile range (IQR) of time intervals between events, reflecting the irregular timing of clinical interactions "EHR-OMOP" (blue) is the 0.5M patients in the EHR-OMOP validation set. The x-axis (log scale) represents the metric in seconds, ranging from $10^1$ to $10^9$. The y-axis measures the number of sequences with those values. Here, we focus on event intervals to capture the temporal structure of clinical encounters and highlight patterns in patient healthcare utilization.

For pretraining, we employ an autoregressive next-token prediction objective with cross entropy loss. We sample one subsequence of $\min\{L, |T_i|\}$ tokens from each patient $i$'s timeline per epoch and train each model for 2 billion tokens.

## 3.2 EVALUATION

We use the EHRSHOT clinical prediction benchmark for all of our downstream evaluations (Wornow et al., 2023). EHRSHOT consists of 15 clinical prediction tasks based on a dataset of 7k patients' longitudinal EHRs. The primary evaluation metric is AUROC, and Brier scores are also reported. We only consider binary classification tasks, thus we exclude the multilabel *Chest X-Ray Findings* task. We use the remaining 14 tasks from the EHRSHOT benchmark for our evaluations, which are broadly grouped into three categories: *Operational Outcomes* includes predicting ICU Transfer, 30-day Readmission, and Long Length-of-Stay; *Anticipating Lab Test Results* involves predicting if a thrombocytopenia, hyperkalemia, hypoglycemia, hyponatremia, or anemia lab will be abnormal; and *Assignment of New Diagnoses* requires predicting whether a patient will get a new diagnosis of hypertension, hyperlipidemia, pancreatic cancer, celiac disease, or lupus within the next year. For additional details on all 14 tasks, including precise definitions, label counts, statistics on the number of tokens per patient, and evaluation methodology, please see Appendix Section A.

For our evaluations, we use the same context length that was used during pretraining. We thus sample the last $\min\{L, |T_i|\}$ tokens for each patient prior to the relevant prediction time for a task, then take the embedding of the last token in that sequence as our representation for that patient. We evaluate our models under the zero-shot, few-shot, and "All" data setting, with detailed results for zero- and few-shot evaluation provided in Appendix Sections G and H. All EHRSHOT scores reported in the main results use the "All" data setting. To be consistent with the original EHRSHOT benchmark, we do not finetune our base models – instead, we train a logistic regression head on top of the frozen representations created for each patient. Additional details are in Appendix Section A.

## 3.3 EHR-SPECIFIC PROPERTIES

In the following subsections, we define metrics to quantify three properties of EHR data that distinguish it from modalities such as natural language – repetitiveness due to copy-forwarding, irregular intervals of time between events, and a natural trend towards increased token complexity over time due to disease progression. Please see Figure 1c for an overview. We believe this analysis provides an interesting counterpoint to most ML research being conducted on natural language sequences.

For all three metrics, we first apply them to the EHR-OMOP validation dataset to measure the extent to which a large corpus of real-world EHR data exhibits these properties. Second, we apply two of the EHR-specific metrics – repetitiveness and irregularity – to the EHRSHOT dataset to stratify

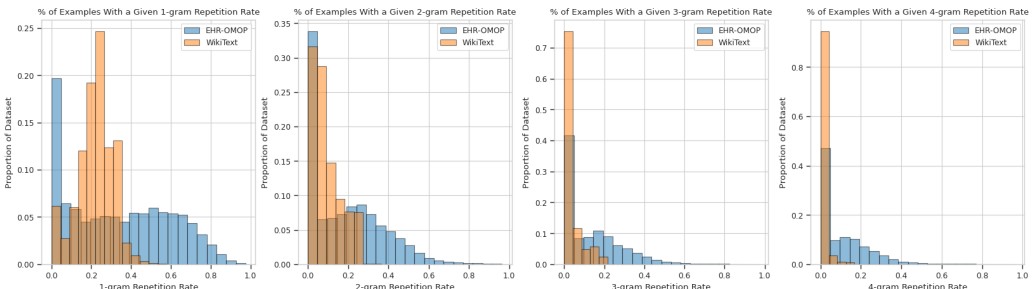

Figure 3: EHR data exhibits a higher degree of repetition than natural language, as measured by $n$-gram repetition rates. From left to right, we measure $n = 1, 2, 3, 4$. "EHR-OMOP" (blue) is the 0.5M patients in the EHR-OMOP validation dataset, "WikiText" (orange) is the WikiText-103 training dataset of high quality Wikipedia articles (Merity et al., 2016). We analyze $n$-gram repetition at the event level to reflect the structure of recurring clinical events, capturing patterns unique to EHR data. The x-axis represents the $n$-gram repetition rate (i.e., percentage of $n$-grams that are repeated at least once within a sequence, where higher is more repetitive) and the y-axis shows the frequency of sequences with that repetition rate in each dataset.

individual patients based on how much they exhibit each property. This stratification allows us to assess how model performance varies across different levels of these properties, and to what extent longer context models can maintain robust performance.

### 3.3.1 COPY-FORWARDING LEADS TO NOISY TOKEN REPETITION

**EHR v. NLP.** Copy-forwarding refers to the practice of recording the same diagnosis across multiple visits, typically for chronic conditions or billing purposes (Thornton et al., 2013; Calder et al., 2024; Weis & Levy, 2014). This leads to higher levels of event repetition within the EHR. We hypothesize that repetition could worsen model performance by crowding information out of a limited context window. A long context model might be better equipped to handle this range of possibilities.

**Metrics.** To quantify the prevalence of copy-forwarding in a sequence, we calculate its $n$-gram repetition rate (RR), i.e., the proportion of $n$-grams in the sequence that are repeated at least once. Please see Appendix Section F.1 for details. A higher RR implies a more repetitive sequence.

### 3.3.2 TIME INTERVALS BETWEEN EVENTS ARE HIGHLY IRREGULAR

**EHR v. NLP.** In natural language, consecutive tokens uniformly have the same "distance" of 1 position. In EHR data, however, a patient might wait days, weeks, or even years between visits to the hospital (McDermott et al., 2023). This means consecutive EHR events can have vastly different "distances" in time. We hypothesize that patients with more "irregular" sequences, i.e., a greater variety of inter-event time intervals, are more difficult to model as they present a more complex mix of timespans over which a model must reason. This could pose particular challenges to long context models given they observe an even broader range of events (and thus inter-event timespans).

**Metrics.** We quantify irregularity as the standard deviation of time intervals between every pair of consecutive events. A higher standard deviation implies a more irregular sequence. Please see Appendix Section F.2 for more details.

### 3.3.3 DISEASE PROGRESSION CAUSES INCREASED TOKEN COMPLEXITY OVER TIME

**EHR v. NLP.** Disease progression refers to the evolving nature of a patient's health over time. As people age, they experience an increase in the variety, frequency, and complexity of diseases they experience due to declining immunity and the increased likelihood of developing comorbidities (Fabbri et al., 2015). In natural language, earlier tokens tend to help in predicting later tokens, and thus perplexity is inversely correlated with a token's position in a prompt (Kaplan et al., 2020). Since disease becomes more complex over time, however, it was unclear if this trend holds for EHR data.

**Metrics.** To quantify disease complexity over time, we apply our trained EHR FMs to calculate the median perplexity at each token position across a sample of 20,000 patients from the EHR-OMOP validation set. Please see Appendix Section F.3 for additional experimental details.

| Metric | Model | Context Length | Q1 | Q2 | Q3 | Q4 |
|---|---|---|---|---|---|---|
| Repetitiveness (1-gram RR) | Mamba | 1k | 0.0644 | 0.0737 | 0.0744 | 0.0790 |
| | | 16k | **0.0605** | **0.0670** | **0.0700** | **0.0746** |
| | Llama | 512 | 0.0640 | 0.0710 | 0.0743 | 0.0792 |
| | | 4k | **0.0627** | **0.0687** | **0.0721** | **0.0770** |
| | CLMBR-t-base | 512 | 0.0647 | 0.0719 | 0.0751 | 0.0805 |
| Irregularity (Standard Deviation) | Mamba | 1k | 0.0693 | 0.0729 | 0.0731 | 0.0764 |
| | | 16k | **0.0641** | **0.0678** | **0.0679** | **0.0723** |
| | Llama | 512 | 0.0694 | 0.0730 | 0.0713 | 0.0749 |
| | | 4k | **0.0664** | **0.0705** | **0.0694** | **0.0740** |
| | CLMBR-t-base | 512 | 0.0683 | 0.0741 | 0.0721 | 0.0777 |

Table 2: Comparison of average Brier scores of models across all 14 EHRSHOT tasks. Patients are bucketed by repetitiveness (top) and irregularity (bottom). Q1/Q2/Q3/Q4 are the 1st through 4th quartiles of patients ranked by each metric. For example, Q1 contains the least repetitive / irregular patients while Q4 contains the most repetitive / irregular patients. **Bolded** values show a statistically significant win rate of at least 50% of the longer context model over the shorter context model at a specific quartile. Only Mamba, Llama, and CLMBR-t-base (the prior SOTA) are shown for space – see Appendix Table 14 for results on all models.

## 4 RESULTS

First, we evaluate each of our models on the 14 EHRSHOT clinical prediction tasks. Overall results are shown in Figure 1b, and per-task results in Appendix Figure 9. Our best performing model is Mamba with a context length of 16k tokens. It achieves the highest average AUROC across all tasks, beating the prior state-of-the-art by 0.03 points. Second, we analyze how three EHR-specific properties – event repetition from copy-forwarding, irregularly spaced inter-event times, and disease progression – impact model performance. After stratifying EHRSHOT patients into quartiles by each property, we find that each property negatively correlates with model performance. However, longer context models exhibit more robustness as they perform better across all quartiles.

### 4.1 LONGER CONTEXTS IMPROVE PREDICTION MAKING FOR CERTAIN ARCHITECTURES

Our best performing model is Mamba at its maximum context length of 16k tokens, with a mean AUROC of 0.807 (+0.03 points over prior SOTA). This can be seen in Figure 1b. Each line represents a separate model architecture. The y-axis is mean AUROC across the 14 EHRSHOT tasks, and the x-axis is the context length. The dotted purple line is the AUROC (0.777) achieved by the best overall prior model, CLMBR-t-base, which had a context length of 512 tokens (Wornow et al., 2023).

Several trends appear in Figure 1b. Both Mamba (green) and Llama (orange) show increased performance at longer context lengths, demonstrating the value of additional EHR data when making clinical predictions. In contrast, Hyena (red) exhibits a sharp decrease in performance after exceeding a context length of 4k. This shows that including more tokens into the context does not always improve performance across architectures. The impact of context length on GPT (blue) appears less clear, which could be due to its usage of absolute positional embeddings (see Section 4.4 for additional analysis). Results on individual tasks are in Appendix Figure 9.

To more explicitly model the passage of time, we also train a version of our models using the Artificial Time Tokens (ATT) technique proposed in CEHR-BERT (Pang et al., 2021). However, as shown in Appendix Figure 12, we see slightly worse performance with this tokenization strategy.

### 4.2 COPY-FORWARDING CREATES NOISY REPETITION HARMING MODEL PERFORMANCE

**EHR-OMOP Analysis.** We measure the $n$-gram repetition rate (RR) across all 0.5M EHR-OMOP validation patients and plot the frequency of each observed RR in Figure 3 in blue. We perform the same calculations on the WikiText-103 dataset and overlay them in orange as "WikiText" as a point of comparison (Merity et al., 2016). While a significant number of patients have no repeated $n$-grams in their records due to their short length (see Appendix Figure 8 for a recreation of this plot that excludes patients with less than 20 total events), we see that EHR data still exhibits a much higher degree of repetition than does natural language, especially when considering the repetition of 3-grams and 4-grams. For more details on $n$-gram RRs, see Appendix Section F.1.

**EHRSHOT Stratification.** Next, we evaluated how the repetitiveness of a patient's timeline affects model performance on the EHRSHOT benchmark using Brier score. Using 1-gram repetition rate

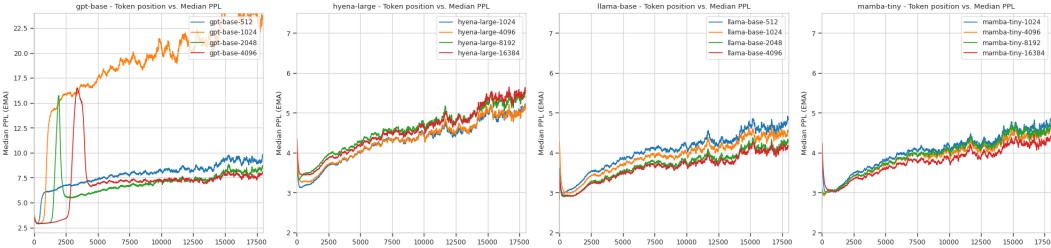

Figure 4: Median perplexity (PPL) by token position for different models – GPT (far left), Hyena (middle left), Llama (middle right), Mamba (far right) – across varying context lengths (lines). The x-axis represents token position, and the y-axis shows the median PPL at each position measured across 20k EHR-OMOP patients. We analyze PPL by token rather than by event to capture the model's handling of the specific information content in each encoded token.Note that the upward trend in PPL is almost immediate, even within the first hundred tokens of each model's context window.

as the metric, patients were grouped into quartiles from Q1 (lowest) to Q4 (highest). 1d (top) show that increased repetition reduces the performance of CLMBR-t-base.

We repeated this analysis with the EHR FMs trained in this work 2 (top). Model performance consistently degrades as repetition increases, indicating that highly repetitive sequences are more challenging to model. Notably, longer context versions of Mamba and Llama achieve significantly lower Brier scores across all quartiles compared to their shorter counterparts.

### 4.3 IRREGULAR INTER-TOKEN TIME INTERVALS ARE HARDER TO MODEL

**EHR-OMOP Analysis.** We first quantify the degree to which EHR data exhibits irregularity in the intervals of time between consecutive events. Figure 2 shows three different metrics for irregularity – the mean, standard deviation, and interquartile range of inter-event times for each individual patient – for the EHR-OMOP validation set in blue. The x-axis of each plot is on a log scale, illustrating the large range of inter-event times across patients. Most patients appear to have a standard deviation of inter-event times between $10^7$ and $10^8$ seconds (i.e. 115 days to 3.2 years).

**EHRSHOT Stratification.** Next, we measured how patient timeline irregularity impacts model performance on the EHRSHOT benchmark using Brier score. Evaluating CLMBR-t-base across quartiles of patient irregularity (using the standard deviation of inter-event times as the metric), we found that performance generally degrades (higher Brier scores) as irregularity increases 1d (middle), indicating that irregular sequences are harder to model.

Table 2 extends this analysis to the EHR FMs trained in this work. While model performance still degrades with increased irregularity, longer context versions of Mamba and Llama consistently outperform their shorter counterparts across all quartiles.

### 4.4 DISEASE PROGRESSION EFFECTS ARE BETTER MODELED WITH LONGER CONTEXTS

**EHR-OMOP Analysis.** Figure 4 shows that tokens later in a patient's timeline are more difficult to predict (higher perplexity), even when conditioning on all prior tokens. This contrasts with natural language, where later tokens tend to have lower perplexity (Kaplan et al., 2020; Peng et al., 2023b). We hypothesize this is because diseases naturally become more complex and varied with aging. This degrades the predictive utility of past medical history as primary diagnoses change over time.

Longer context versions of Mamba and Llama consistently achieve lower perplexities across all token positions compared to shorter contexts, with the gap widening at later tokens. This suggests that a more complete view of the patient's timeline helps handle increasing token complexity due to aging. In contrast, Hyena's longer context models perform worse, replicating our original EHRSHOT results. For GPT, results are mixed: longer contexts (2k and 4k) achieve lower perplexities at later tokens but exhibit significant spikes. This appears to be caused by GPT's usage of absolute positional embeddings – replacing them with rotary positional embeddings (ROPE) (Su et al., 2024) mitigated these spikes as seen in Appendix Figure 11. Thus, despite its popularity in the EHR FM community (see Table 1), we recommend discontinuing the GPT architecture in favor of Llama or other more modern decoder-only architectures.

## 5 DISCUSSION

In this study, we evaluated the impact of context length on clinical prediction tasks across four models—Mamba, Llama, GPT, and Hyena—trained on longitudinal EHR data. We are the first to pretrain and release the full weights of these non-GPT architectures at the scale of millions of EHRs. With a context length of 16k tokens, Mamba achieved the highest average AUROC across 14 prediction tasks on the EHRSHOT benchmark, surpassing the prior state-of-the-art by +0.03 points. In addition to the best performance, Mamba also offers faster training, quicker inference, and the potential to support longer contexts (Gu & Dao, 2024). Notably, longer context versions of Mamba and Llama performed well in handling EHR-specific issues like token repetition due to copy-forwarding, irregular inter-token time intervals, and increased token complexity from disease progression. This improvement, however, wasn't universal, as Hyena's performance declined significantly beyond 4k tokens, underscoring the need to empirically validate each architecture for long context use.

**Limitations / Future Work:** While our findings highlight the potential for long-context models to successfully model EHR data, several limitations should be considered. First, we did not evaluate transformer-based models at context lengths beyond 4k tokens due to limited computational resources. Running a vanilla 16k transformer takes roughly 16x more compute/memory than at a context length of 4k, which was a core motivator for the development of the subquadratic architectures evaluated in this work. Second, model sizes were kept consistent across architectures to isolate the impact of context length. Preliminary findings suggest smaller Mamba models with 16k tokens perform well, which may reduce the need for larger models unsuitable for resource-constrained settings. Future work should quantify the impact of model size on performance. Third, our evaluations focused on clinical risk prediction tasks, but broader clinical tasks (e.g., phenotyping, treatment selection) merit further consideration. Fourth, our pretraining dataset was sourced from a single institution due to data privacy concerns, which may limit generalizability. Fifth, we explored only three EHR-specific properties. Future research could extend this to more attributes of EHR data – e.g., partial observation due to underdiagnosis or miscoding (Pivovarov et al., 2014; Che et al., 2018), multimodal signals (Soenksen et al., 2022), and event-associated metadata (McDermott et al., 2023). Sixth, we focused on the impact of these EHR-specific properties on downstream evaluations, but they may also have effects on pretraining convergence and stability, which we leave to future work. Seventh, while the metrics we introduce offer a novel lens for examining EHR data, they are fairly simple and could be improved with additional context. For example, having our repetition metric distinguish between meaningful and non-meaningful repetition (e.g., a repeated lab test in an ICU stay is likely more informative than a repeated diagnosis code of a chronic condition like hypertension) could improve model performance in high-repetition settings. And for the irregularity metric, disease status may influence the regularity of time intervals between events (e.g. a cancer patient may exhibit more regular visits than a patient suffering from acute cardiovascular events), which future work could explore by stratifying results based on specific disease phenotypes. Eighth, other promising transformer alternatives, such as linear attention models (Arora et al., 2024), hybrid architectures (Poli et al., 2023b; Lieber et al., 2024), and recurrent models (Peng et al., 2023a), should be explored in future work that builds upon the framework introduced here.

## 6 CONCLUSION

Long context models have unlocked a broad range of natural language applications through their ability to ingest and reason over massive amounts of information. Translating these gains to EHR data could benefit patients by enabling the modeling of an entire lifetime. Thus, we present the first systematic evaluation of how context length impacts EHR modeling. We find that long context subquadratic models such as Mamba are capable of achieving state-of-the-art results on clinical prediction tasks. This represents a sharp break from prior work in EHR FMs, as shown in Table 1, which generally utilized BERT-based models limited to context windows of 512 tokens. We also find that longer context models are more robust to three distinct aspects of EHR data that had been underexplored in prior literature on sequence modeling. We hope our work inspires future efforts to identify interesting sequence modeling challenges from non-NLP domains and encourages further research towards applying non-transformer architectures to structured EHR data.

ACKNOWLEDGMENTS

NHS acknowledges support by the Mark and Debra Leslie Endowment for AI in Healthcare, the Clinical Excellence Research Center at Stanford Medicine, and Technology and Digital Solutions at Stanford Healthcare. JF was supported in part by a Stanford AIMI-HAI Partnership Grant. CR acknowledges the support of NIH under No. U54EB020405 (Mobilize), NSF under Nos. CCF2247015 (Hardware-Aware), CCF1763315 (Beyond Sparsity), CCF1563078 (Volume to Velocity), and 1937301 (RTML); US DEVCOM ARL under Nos. W911NF-23-2-0184 (Long-context) and W911NF-21-2-0251 (Interactive Human-AI Teaming); ONR under Nos. N000142312633 (Deep Signal Processing); Stanford HAI under No. 247183; NXP, Xilinx, LETI-CEA, Intel, IBM, Microsoft, NEC, Toshiba, TSMC, ARM, Hitachi, BASF, Accenture, Ericsson, Qualcomm, Analog Devices, Google Cloud, Salesforce, Total, the HAI-GCP Cloud Credits for Research program, the Stanford Data Science Initiative (SDSI), and members of the Stanford DAWN project: Meta, Google, and VMWare. SK acknowledges support by NSF 2046795 and 2205329, IES R305C240046, the MacArthur Foundation, Stanford HAI, OpenAI, and Google. The U.S. Government is authorized to reproduce and distribute reprints for Governmental purposes notwithstanding any copyright notation thereon. Any opinions, findings, and conclusions or recommendations expressed in this material are those of the authors and do not necessarily reflect the views, policies, or endorsements, either expressed or implied, of NIH, ONR, or the U.S. Government.

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

## A  DATASET

Our primary dataset, "EHR-OMOP", is sourced from an academic medical center. It contains deidentified longitudinal EHR data formatted according to the Observational Medical Outcomes Partnership Common Data Model (OMOP-CDM) (Sciences & Informatics, 2021). All data is stripped of protected health information and deidentified at the institution level to comply with HIPAA and the Safe Harbor standard. The dataset is stored in a HIPAA-compliant compute environment. All patients included in EHR-OMOP sign a form consenting their records to be included in research purposes like this work. This study was conducted under an institution-wide IRB protocol that makes this deidentified dataset available for research purposes.

We use roughly 2.5M patients from EHR-OMOP for pretraining our models, and hold out 0.5M patients for conducting validation experiments.

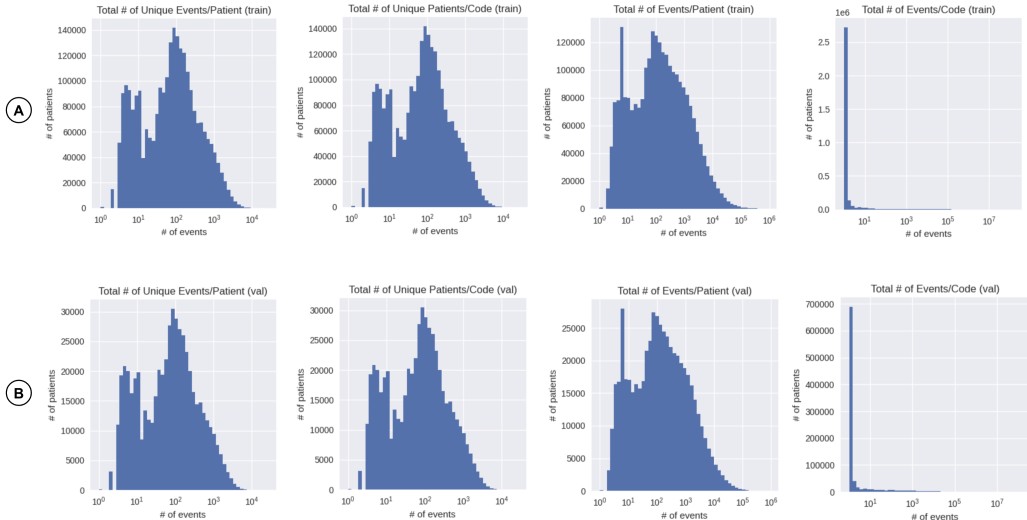

Figure 5: Distributions of patient data from the EHR-OMOP dataset across (A) training and (B) validation splits, showing both event-level and code-level counts. The x-axis is log-scaled to capture the wide range in the number of events per patient, the number of unique patients per code, and the distribution of events associated with each code.

| Training Split | Value | Validation Split | Value |
|---|---|---|---|
| *Overall counts* | | *Overall counts* | |
| Number of events | 3,501,210,238 | Number of events | 749,003,035 |
| Unique codes | 3,144,978 | Unique codes | 881,012 |
| Unique patients | 2,567,450 | Unique patients | 550,305 |
| | | | |
| *Events per patient* | | *Events per patient* | |
| Minimum | 1 | Minimum | 1 |
| Mean | 1,364 | Mean | 1,361 |
| Median | 121 | Median | 121 |
| Maximum | 890,048 | Maximum | 638,708 |
| | | | |
| *Unique events per patient* | | *Unique events per patient* | |
| Minimum | 1 | Minimum | 1 |
| Mean | 237 | Mean | 237 |
| Median | 76 | Median | 76 |
| Maximum | 26,131 | Maximum | 18,561 |

Table 3: Summary statistics for the EHR-OMOP training (left) and validation (right) splits.

## B EVALUATION

### B.1 TASKS

For all of our model evaluations, we use 14 binary clinical prediction tasks sourced from the EHRSHOT benchmark (Wornow et al., 2023). The definitions of these tasks are detailed in Appendix Table 4. We also provide label and patient counts in Appendix Table 5 for each task.

| Task Name | Task Type | Prediction Time | Time Horizon |
|---|---|---|---|
| **Operational Outcomes** | | | |
| Long Length of Stay | Binary | 11:59pm on day of admission | Admission duration |
| 30-day Readmission | Binary | 11:59pm on day of discharge | 30 days post-discharge |
| ICU Transfer | Binary | 11:59pm on day of admission | Admission duration |
| | | | |
| **Anticipating Lab Test Results** | | | |
| Thrombocytopenia | Binary | Immediately before result | Next result |
| Hyperkalemia | Binary | Immediately before result | Next result |
| Hypoglycemia | Binary | Immediately before result | Next result |
| Hyponatremia | Binary | Immediately before result | Next result |
| Anemia | Binary | Immediately before result | Next result |
| | | | |
| **Assignment of New Diagnoses** | | | |
| Hypertension | Binary | 11:59pm on day of discharge | 1 year post-discharge |
| Hyperlipidemia | Binary | 11:59pm on day of discharge | 1 year post-discharge |
| Pancreatic Cancer | Binary | 11:59pm on day of discharge | 1 year post-discharge |
| Celiac | Binary | 11:59pm on day of discharge | 1 year post-discharge |
| Lupus | Binary | 11:59pm on day of discharge | 1 year post-discharge |
| Acute MI | Binary | 11:59pm on day of discharge | 1 year post-discharge |

Table 4: The 14 clinical prediction tasks used for evaluating models in this work. *Prediction Time* is the precise time point (up to minute precision) in a patient's timeline when the prediction is made. *Time Horizon* is the length of time considered after the prediction time to determine whether an event occurs, i.e. we only consider a patient "positive" for a new diagnosis of pancreatic cancer if she receives that diagnosis within a year of being discharged. Table reproduced verbatim from (Wornow et al., 2023).

The definitions for each task are provided below (reproduced verbatim from (Wornow et al., 2023)).

**Operational Outcomes**. These tasks are related to hospital operations. They are defined as follows:

- **Long Length of Stay**: Predict whether a patient's total length of stay during a visit to the hospital will be at least 7 days. The prediction time is at 11:59pm on the day of admission, and visits that last less than one day (i.e. discharge occurs on the same day of admission) are ignored.

- **30-day Readmission**: Predict whether a patient will be re-admitted to the hospital within 30 days after being discharged from a visit. The prediction time is at 11:59pm on the day of admission, and admissions where a readmission occurs on the same day as the corresponding discharge are ignored.

- **ICU Transfer**: Predict whether a patient will be transferred to the ICU during a visit to the hospital. The prediction time is at 11:59pm on the day of admission, and ICU transfers that occur on the same day as admission are ignored.

**Anticipating Lab Test Results**. These tasks are related to lab value prediction. The prediction time is immediately before the lab result is recorded. They are defined as follows:

- **Thrombocytopenia**: Predict whether a thrombocytopenia lab comes back as normal ($>=150$ $10^9$/L) or abnormal (any other reading). We consider all lab results coded as LOINC/LP393218-5, LOINC/LG32892-8, or LOINC/777-3.

- **Hyperkalemia**: Predict whether a hyperkalemia lab comes back as normal ($<=5.5$ mmol/L), or abnormal (any other reading). We consider all lab results coded as LOINC/LG7931-1, LOINC/LP386618-5, LOINC/LG10990-6, LOINC/6298-4, or LOINC/2823-3.

| Task Name | Train | | Val | | Test | |
|---|---|---|---|---|---|---|
| | # Patients (# Positive) | # Labels (# Positive) | # Patients (# Positive) | # Labels (# Positive) | # Patients (# Positive) | # Labels (# Positive) |
| **Operational Outcomes** | | | | | | |
| Long Length of Stay | 1377 (464) | 2569 (681) | 1240 (395) | 2231 (534) | 1238 (412) | 2195 (552) |
| 30-day Readmission | 1337 (164) | 2608 (370) | 1191 (159) | 2206 (281) | 1190 (151) | 2189 (260) |
| ICU Transfer | 1306 (107) | 2402 (113) | 1157 (84) | 2052 (92) | 1154 (75) | 2037 (85) |
| **Anticipating Lab Test Results** | | | | | | |
| Thrombocytopenia | 2084 (906) | 68776 (22714) | 1981 (807) | 54504 (17867) | 1998 (853) | 56338 (19137) |
| Hyperkalemia | 2038 (456) | 76349 (1829) | 1935 (428) | 60168 (1386) | 1958 (405) | 63653 (1554) |
| Hypoglycemia | 2054 (511) | 122108 (1904) | 1950 (433) | 95488 (1449) | 1970 (435) | 100568 (1368) |
| Hyponatremia | 2035 (1294) | 81336 (23877) | 1930 (1174) | 64473 (17557) | 1956 (1224) | 67028 (19274) |
| Anemia | 2092 (1484) | 70501 (49028) | 1992 (1379) | 56224 (38498) | 2002 (1408) | 58155 (39970) |
| **Assignment of New Diagnoses** | | | | | | |
| Hypertension | 792 (129) | 1259 (182) | 781 (128) | 1247 (175) | 755 (129) | 1258 (159) |
| Hyperlipidemia | 923 (137) | 1684 (205) | 863 (140) | 1441 (189) | 864 (133) | 1317 (172) |
| Pancreatic Cancer | 1376 (128) | 2576 (155) | 1242 (46) | 2215 (53) | 1246 (29) | 2220 (56) |
| Celiac | 1392 (48) | 2623 (62) | 1252 (8) | 2284 (11) | 1255 (13) | 2222 (21) |
| Lupus | 1377 (79) | 2570 (104) | 1238 (24) | 2225 (33) | 1249 (19) | 2243 (20) |
| Acute MI | 1365 (130) | 2534 (175) | 1234 (112) | 2176 (145) | 1235 (115) | 2127 (144) |

Table 5: The number of unique patients and total labels for each split of the 14 EHRSHOT tasks evaluated in this work. The prevalence of positive patients/labels is shown in parenthesis. Table reproduced from (Wornow et al., 2023), with updates to reflect the latest version of the EHRSHOT dataset.

- **Hypoglycemia**: Predict whether a hypoglycemia lab comes back as normal ($>=3.9$ mmol/L) or abnormal (any other reading). We consider all lab results coded as SNOMED/33747003, LOINC/LP416145-3, or LOINC/14749-6.

- **Hyponatremia**: Predict whether a hyponatremia lab comes back as normal ($>=135$ mmol/L) or abnormal (any other reading). We consider all lab results coded as LOINC/LG11363-5, LOINC/2951-2, or LOINC/2947-0.

- **Anemia**: Predict whether an anemia lab comes back as normal ($>=120$ g/L) or abnormal (any other reading). We consider all lab results coded as LOINC/LP392452-1.

**Assignment of New Diagnoses**. These tasks are related to predicting the first diagnosis of a disease. The prediction time is at 11:59pm on the day of discharge from an inpatient visit, and we count any diagnosis that occurs within 365 days post-discharge as a positive outcome. We ignore all discharges in which the patient already has an existing diagnosis of a disease. The tasks are defined as follows:

- **Hypertension**: Predict whether the patient will have her first diagnosis of essential hypertension within the next year. We define hypertension as an occurrence of the code SNOMED/59621000, as well as its children codes in our ontology.

- **Hyperlipidemia**: Predict whether the patient will have her first diagnosis of hyperlipidemia within the next year. We define hyperlipidemia as an occurrence of the code SNOMED/55822004, as well as its children codes in our ontology.

- **Pancreatic Cancer**: Predict whether the patient will have her first diagnosis of pancreatic cancer within the next year. We define pancreatic cancer as an occurrence of the code SNOMED/372003004, as well as its children codes in our ontology.

- **Celiac**: Predict whether the patient will have her first diagnosis of celiac disease within the next year. We define celiac disease as an occurrence of the code SNOMED/396331005, as well as its children codes in our ontology.

- **Lupus**: Predict whether the patient will have her first diagnosis of lupus within the next year. We define lupus as an occurrence of the code SNOMED/55464009, as well as its children codes in our ontology.

- **Acute MI**: Predict whether the patient will have her first diagnosis of an acute myocardial infarction within the next year. We define myocardial infarction as an occurrence of the code SNOMED/57054005, as well as its children codes in our ontology.

## B.2 EVALUATION PROCEDURE

Each model $m \in \mathcal{M}$ outputs an embedding for each token in its input sequence. Our goal is to aggregate these outputs into a unified representation $R_i$ for each patient $i$ which captures key patterns in their disease trajectory. We will then use this representation $R_i$ to finetune a logistic regression head for our downstream binary classification prediction tasks.

We define two functions. First, we define $S : \mathbf{R}^{n \times d} \to \mathbf{R}^{k \times d}$ to select a subset of $k$ vectors from a set of $n$ vectors. Second, we define $A : \mathbf{R}^{n \times d} \to \mathbf{R}^d$ to aggregate a set of $n$ $d$-dimensional vectors into a single vector. Thus:

$$R_i = A(S(m(\{T_{ik}, ..., T_{i(k+L)}\})))$$

Initial experiments indicated that setting $A$ to simply return the last vector in the sequence (i.e. the most recent token in a patient's timeline) and $S$ to the most recent $L$ tokens in a patient's timeline prior to the timepoint at which the prediction for a task is made performed the best. Thus, we have:

$$R_i = \text{mean}(m(\{T_{i,|T_i|-L}, ..., T_{i|T_i|}\}))$$

Finally, we fit a logistic regression head $H$ on top of these representations in order to apply them to binary prediction tasks. This yields a final prediction $P_i$ of:

$$P_i = H(R_i)$$

which provides the model's estimate for the probability that a specific clinical event occurs within a task-defined window of time for this patient $i$ based on their current representation $R_i$.

## B.3 PATIENT STATISTICS

In Appendix Figure 6, we plot the CDF of the number of **raw clinical events** and **tokens** preceding each prediction time for a given task across train/val/test splits. The blue line represents all prediction times, the orange line corresponds to only predictions associated with a positive label. Note that not every clinical event corresponds to a token in our vocabulary, hence many events are dropped during the tokenization process.

## B.4 TASK-LEVEL RESULTS

We present plots of each model's performance on the 14 individual EHRSHOT tasks in Appendix Figure 9. Additionally, we provide raw numbers on the AUROC differences between each model and the prior SOTA model, CLMBR-t-base, for each task in Appendix Tables 7, 8, 9, 10. We report bootstrapped 95% confidence intervals over 1,000 resamples of the test set for each AUROC difference. Across all context lengths, our results for Mamba are shown in Appendix Table 7, Llama in Appendix Table 8, GPT in Appendix Table 9, and Hyena in Appendix Table 10.

## C MODEL ARCHITECTURES

In this section, we present the mathematical formulations and detailed architectural descriptions of the four models used in our experiments: GPT, Mamba, Llama, and Hyena.

### C.1 GPT

GPT (Generative Pre-trained Transformer) is a transformer-based autoregressive model that uses self-attention to process input sequences. (Brown et al., 2020) The main operation is the scaled dot-product attention:

$$\text{Attention}(Q, K, V) = \text{softmax}\left(\frac{QK^\top}{\sqrt{d_k}}\right)V \tag{1}$$

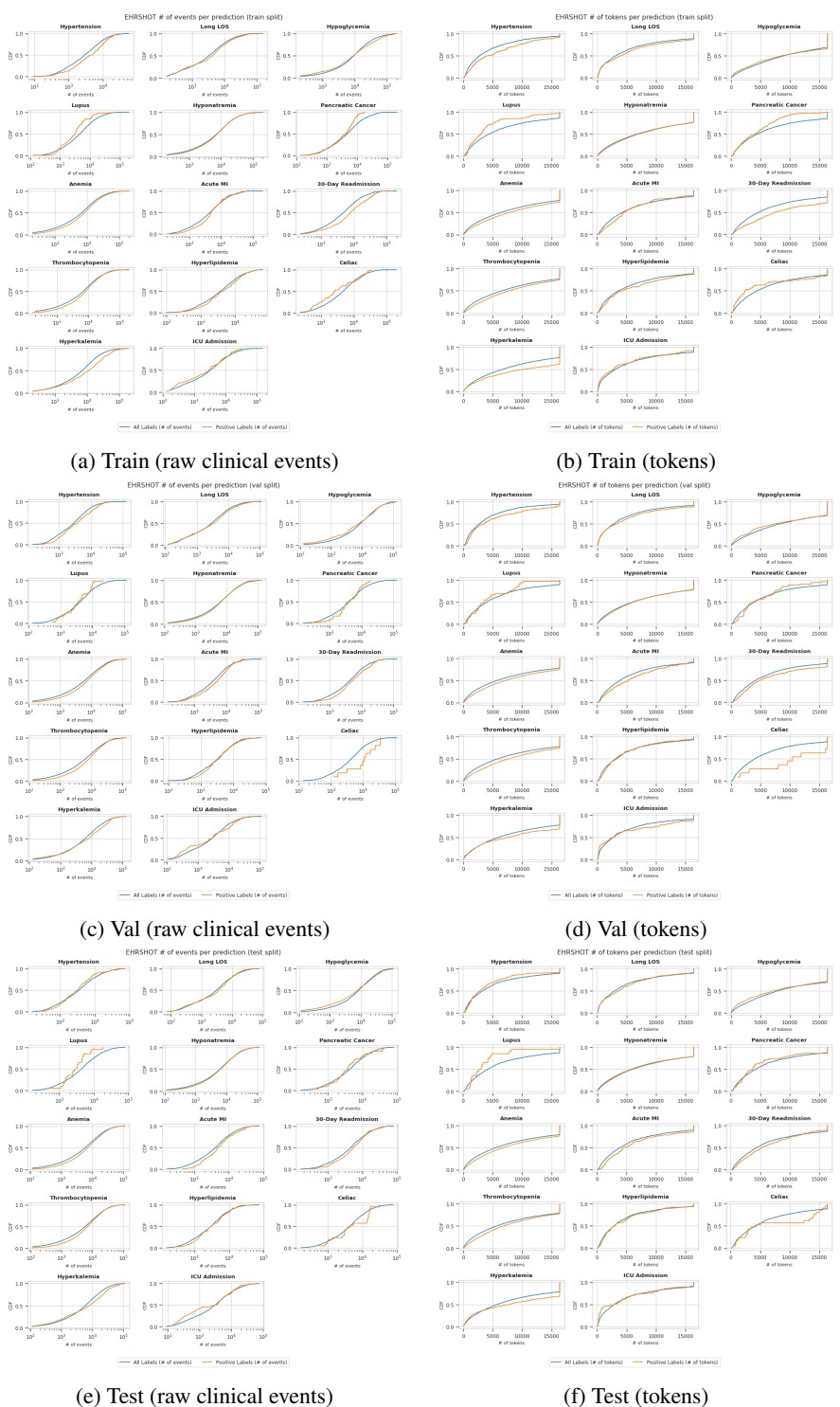

(a) Train (raw clinical events)

(b) Train (tokens)

(c) Val (raw clinical events)

(d) Val (tokens)

(e) Test (raw clinical events)

(f) Test (tokens)

Figure 6: For each EHRSHOT task, we plot the CDF of the number of **raw clinical events** (left column) and **tokens** (right column) available to the model when making its prediction. In other words, the number of events/tokens preceding each label's prediction time point. The blue line represents all prediction times, while the orange line represents only predictions associated with a positive label. Note that unlike the raw event counts, all token counts are capped at the maximum context length of the models we test (16k), hence the spike at the end of the CDF.

Here, $Q$, $K$, and $V$ are the query, key, and value matrices, respectively, and $d_k$ is the dimensionality of the key vectors. The transformer block consists of multi-head attention and a position-wise feed-forward network:

$$\text{MultiHead}(Q, K, V) = \text{Concat}(\text{head}_1, ..., \text{head}_h)W^O \tag{2}$$

$$\text{head}_i = \text{Attention}(QW_i^Q, KW_i^K, VW_i^V) \tag{3}$$

where $W_i^Q$, $W_i^K$, $W_i^V$, and $W^O$ are learned projection matrices. After attention, GPT applies a position-wise feed-forward network consisting of two fully connected layers with ReLU activations:

$$\text{FFN}(x) = \text{ReLU}(xW_1 + b_1)W_2 + b_2 \tag{4}$$

The quadratic complexity of self-attention with respect to input length makes it challenging to scale GPT to long context lengths. In our experiments, we use GPT variants with context lengths up to 4096 tokens.

## C.2 LLAMA

Llama is a transformer-based model that shares the core structure of GPT but incorporates optimizations for training efficiency and scalability (Team, 2024). The model uses the same attention mechanism as GPT, but with several architectural modifications, such as an increased hidden state dimension, fewer normalization layers, and relative positional embeddings to improve its performance.

The forward pass for each transformer block in Llama follows the same formulation as GPT, combining self-attention with a feed-forward network:

$$\mathbf{h}_{t+1} = \text{LayerNorm}(\mathbf{h}_t + \text{MultiHead}(\mathbf{h}_t, \mathbf{h}_t, \mathbf{h}_t)) \tag{5}$$
$$\mathbf{h}_{t+2} = \text{LayerNorm}(\mathbf{h}_{t+1} + \text{FFN}(\mathbf{h}_{t+1})) \tag{6}$$

Llama utilizes rotary positional embeddings (RoPE) (Su et al., 2024), which encode relative positional information directly into the self-attention mechanism without requiring absolute positional encodings:

$$\text{RoPE}(q, k, i) = \cos(i\theta)q + \sin(i\theta)k \tag{7}$$

Here, $q$ and $k$ are the query and key vectors, and $\theta$ is a frequency parameter. We evaluate Llama on context lengths of up to 4096 tokens.

## C.3 MAMBA

Mamba is a state-space model (SSM)-based architecture designed to handle long sequences efficiently. It replaces self-attention with state-space layers, which provide linear scaling with respect to input length. Mamba leverages the continuous-time state-space model to capture long-range dependencies:

$$\mathbf{x}_{t+1} = A\mathbf{x}_t + B\mathbf{u}_t \tag{8}$$
$$\mathbf{y}_t = C\mathbf{x}_t + D\mathbf{u}_t \tag{9}$$

where $\mathbf{x}_t$ is the hidden state, $\mathbf{u}_t$ is the input at time $t$, $\mathbf{y}_t$ is the output, and $A$, $B$, $C$, and $D$ are learned matrices. This allows Mamba to model long sequences with linear complexity, making it ideal for processing the lengthy and complex event streams in EHR data.

In our experiments, we evaluate Mamba with context lengths of up to 16k tokens. Mamba's efficiency allows it to process long patient histories without the computational overhead of traditional transformer models.

## C.4 HYENA

The Hyena architecture introduces an efficient mechanism for handling long sequences by utilizing implicit long convolutions and multiplicative gating (Poli et al., 2023a).

The input sequence is denoted by $\mathbf{x}(t)$, where $t$ represents the sequence position. The convolution operation applied in Hyena can be described by the following equation:

$$\mathbf{y}(t) = \sum_{i=0}^{L-1} \mathbf{h}(i) \cdot \mathbf{x}(t-i)$$

where $\mathbf{x}(t)$ is the input at time step $t$, $\mathbf{h}(i)$ is the convolution filter of length $L$, $\mathbf{y}(t)$ is the output at time step $t$, and $L$ is the length of the filter.

The key difference between Hyena and traditional attention mechanisms is the use of implicit convolutions, which avoid the quadratic complexity of the attention mechanism.

To further enhance the expressivity of the model, Hyena applies multiplicative gating after the convolution operation. This gating mechanism can be expressed as:

$$\mathbf{z}(t) = \sigma(\mathbf{W}_1 \cdot \mathbf{y}(t)) \odot \mathbf{W}_2 \cdot \mathbf{y}(t)$$

where:

- $\mathbf{z}(t)$ is the gated output,
- $\sigma$ is a non-linear activation function (e.g., sigmoid),
- $\mathbf{W}_1$ and $\mathbf{W}_2$ are learnable weight matrices,
- $\odot$ represents element-wise multiplication.

This combination of implicit long convolutions and multiplicative gating allows the Hyena model to process sequences with log-linear complexity in their length.

## D TOKENIZATION

We follow the tokenization strategy used by the CLMBR-t-base model which had achieved the highest average AUROCs on the EHRSHOT benchmark (Wornow et al., 2023). This tokenization strategy is described in detail in (Steinberg et al., 2021).

Given a patient timeline $X_i$, our goal is to convert it into a sequence of tokens $T_i$ that our models can ingest. Thus, we must map each $X_{ij} = (t_{ij}, c_{ij}, v_{ij})$ to some set of token(s) $T_{ij} = \{T_{ij1}, ..., T_{ijk}\}$ where $T_{ijk} \in \mathbb{T}$.

For encoding the $t_{ij}$ component of each clinical event $X_{ij}$, we utilize positional encodings based on the token position $j$, as prior studies have shown minimal benefits from directly embedding absolute time information (Yang et al., 2023).

For handling the $v_{ij}$ component of $X_{ij}$, we define the following function $g$ to map clinical events to tokens by handling each of the three possible cases for the types of values that $v_{ij}$ can take on separately:

$$g(X_{ij}) = \begin{cases} g_v(c_{ij}) & \text{if } v_{ij} \in \emptyset, \\ g_c(c_{ij}, v_{ij}) & \text{if } v_{ij} \in \mathcal{V}_c, \\ g_n(c_{ij}, v_{ij}) & \text{if } v_{ij} \in \mathcal{V}_n. \end{cases}$$

Thus, the same clinical event (e.g. a lab test for anemia) can be mapped to an arbitrary large set of finer-grained tokens (e.g. one token for all lab tests, one each for mild/moderate/severe, one each for a 10-point scale, etc.).

Following (Steinberg et al., 2021) we choose to employ a deciling strategy for all numerical $v_{ij}$, and we map each unique categorical $v_{ij}$ to its own token.

Let $D : \mathcal{C} \times \mathcal{V}_n \to \{x \in \mathbb{Z} \mid 0 \le x \le 9\}$ be a function that maps $v_{ij}$ to the decile it belongs to when considering all possible values that $c_{ij}$ is associated with in the training set. And let $G(\cdot)$ be a function that maps its input to some unique integer in the domain of our tokenizer's vocabulary.

Thus, we have that:

$$g_v(c_{ij}) = G(c_{ij})$$
$$g_c(c_{ij}, v_{ij}) = G(c_{ij}, v_{ij})$$
$$g_n(c_{ij}, v_{ij}) = G(c_{ij}, D(c_{ij}, v_{ij}))$$

Within our dataset, employing this tokenization strategy results in hundreds of thousands of potential unique codes. Many such codes, however, occur very infrequently. Thus, we select the top $k = 39811$ frequently occurring codes, following the same procedure outlined in (Steinberg et al., 2021). In addition, seven special tokens — [BOS], [EOS], [UNK], [SEP], [PAD], [CLS], and [MASK] — are included, resulting in a total vocabulary size of 39818 tokens. This yields an identical vocabulary to the one used by CLMBR-t-base in the original EHRSHOT benchmark (Wornow et al., 2023).

For positional embeddings, we use the default strategies for the various architectures we evaluate – e.g. absolute positional embeddings for GPT, rotary positional embeddings for Llama, none for Hyena beyond the Hyena positional embedding, and none for Mamba.

For completeness, we also evaluate the impact of injecting explicit temporal information into the patient timeline via **Artificial Time Tokens (ATTs)**, as proposed in CEHR-BERT Pang et al., 2021 and used in other works (Pang et al., 2024; Renc et al., 2024). In brief, we create artificial tokens to represent various time intervals (days, weeks, months, etc.) and inject these tokens between consecutive visits to represent the interval of time between them:

$$\mathrm{ATT} = \begin{cases} D_n & \text{if gap } < 7 \text{ days (e.g., } D_1, ..., D_6), \\ W_n & \text{if } 7 \text{ days} \le \text{gap} < 28 \text{ days (e.g., } W_1, ..., W_4), \\ M_n & \text{if } 28 \text{ days} \le \text{gap} < 365 \text{ days (e.g., } M_1, ..., M_{12}), \\ LT & \text{if gap } \ge 365. \end{cases}$$

Furthermore, to clearly define the start and end of each visit, we enclose each visit $V_i$ with special tokens $\mathrm{VS}$ (Visit Start) and $\mathrm{VE}$ (Visit End). This approach allows us to represent a patient timeline as a structured sequence:

$$P = \{\mathrm{VS}, v_1, \mathrm{VE}, \mathrm{ATT}, \mathrm{VS}, v_2, \mathrm{VE}, \mathrm{ATT}, \dots, \mathrm{VS}, v_i, \mathrm{VE}\}$$

This enhancement directly embeds temporal patterns within the token sequence, providing contextual information about the intervals between clinical events. The results of these models trained using ATT tokens are shown in Appendix Figure 12. The figure shows that this tokenization strategy actually tended to reduce the performance of our models, and our best performing model remains Mamba-16k without ATTs.

## E  TRAINING

In this section, we describe the training of models used in our experiments. All model base configuration were taken from Huggingface, and can be found uder:

- GPT: https://huggingface.co/openai-community/gpt2
- Hyena: https://huggingface.co/LongSafari/hyenadna-large-1m-seqlen-hf
- Mamba: https://huggingface.co/state-spaces/mamba-130m-hf

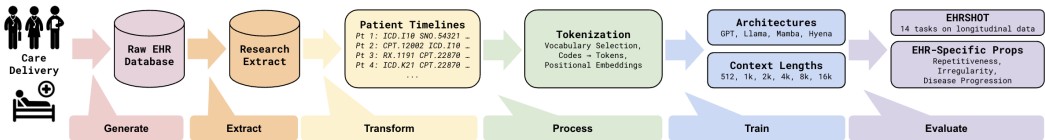

Figure 7: A high-level overview of our experimental pipeline, from data generation to final evaluation results.

- Llama: https://huggingface.co/meta-llama/Llama-3.1-8B-Instruct

Their base configurations were modified to standardize in terms of parameter count to make a fair comparison between them. These configuration changes are shown in Table 6.

| Model | Configuration | Value |
|---|---|---|
| GPT | | |
| | n positions | {512, 1k, 2k, 4k } |
| | learning rate | 2e-4 |
| | dim model | 768 |
| | num layers | 12 |
| | num heads | 12 |
| | Total Parameters | **116M** |
| Hyena | | |
| | max seq len | { 1k, 4k, 8k, 16k } |
| | learning rate | 2e-4 |
| | dim model | 768 |
| | num layers | 16 |
| | Total Parameters | **125M** |
| Mamba | | |
| | max seq len | { 1k, 4k, 8k, 16k } |
| | learning rate | 2e-4 |
| | dim model | 768 |
| | num layers | 24 |
| | num hidden layers | 24 |
| | state size | 16 |
| | Total Parameters | **121M** |
| Llama | | |
| | max position embeddings | {512, 1k, 2k, 4k } |
| | learning rate | 2e-4 |
| | hidden size | 768 |
| | intermediate size | 2688 |
| | num attention heads | 12 |
| | num hidden layers | 8 |
| | num key value heads | 4 |
| | Total Parameters | **123M** |

Table 6: Model configurations used for training. All models are designed to be roughly 120 million parameters. We use the same tokenizer and vocabulary size for all models.

For the pretraining of our models, we randomly sample a patient timeline of length equal to the lesser of the timeline length of the model's context length. To improve training stability and ensure GPU memory optimization, we employed gradient accumulation across multiple batches with a total number of tokens per step of 65,536.

All models were trained using the AdamW optimizer with the following parameters: $\beta_1 = 0.9$, $\beta_2 = 0.95$, $\lambda = 0.1$. We performed a hyperparameter sweep over learning rates between $1e-6$ and $1e-3$ for each model architecture before settling on the learning rates shown in Appendix Table 6. We employed a learning rate warm-up for the first 40,000 steps, after which the learning rate decayed to $1e-5$ as training progressed. This approach ensured smooth convergence while avoiding abrupt changes in training dynamics. Perplexity stabilized after one epoch, and we trained all models for 2 billion total tokens.

The training was conducted on a PHI-compliant shared cluster equipped with a heterogeneous mix of GPUs.The majority of experiments in this work were conducted on a set of V100s, with limited access to another 4 NVIDIA H100s and 16 NVIDIA A100s. The use of a secure, PHI-compliant environment ensured that all patient health information remained confidential and protected throughout the training process, adhering to stringent data privacy regulations.

## F  EHR-SPECIFIC PROPERTY METRICS

We define several metrics for quantifying the specific properties of longitudinal EHR data, such as the irregularity of inter-event time intervals, the repetitiveness of event sequences, and the complexity of tokens due to disease progression. These metrics help us understand the challenges posed by EHR data when used in predictive models.

### F.1  REPETITIVENESS

Due to liability, documentation requirements, billing practices, and other administrative processes, EHR data tends to have a high prevalence of "copy-forwarded" information – i.e. data that is copied-and-pasted from one visit to the next (Thornton et al., 2013; Calder et al., 2024; Weis & Levy, 2014). To quantify the level of "copy-forwarding" within a sequence, we calculate the $n$-gram repetition rate (RR) for each EHR sequence in our dataset using $n = 1, 2, 3, 4$.

We define the $n$-gram repetition rate as the proportion of $n$-grams in a given sequence that are repeated at least once. A higher repetition rate means a sequence is more repetitive. Formally, we define the $n$-gram repetition rate as follows:

$$\text{RR}_n(x) = \frac{\sum_{u \in \mathcal{U}(x)} \mathbb{I}[C(u, x) > 1]}{|\mathcal{U}(x)|}$$

where $\mathcal{U}(\S)$ is the set of unique $n$-grams in the sequence $x$ and $C(u, x) \in \mathbb{R}$ is the count of occurrences of the $n$-gram $u \in \mathcal{U}$ in the sequence $x$. We define $\mathbb{I}[\cdot]$ as the indicator random variable that is 1 if the condition inside the brackets is true, and 0 otherwise.

We calculate $n$-gram repetition rates for $n = 1, 2, 3, 4$ across all 0.5M patients in our EHR-OMOP validation dataset. In Figure 8, we compare the observed repetition rate in our EHR dataset to the repetition rates observed in the WikiText-103 corpus to demonstrate the higher levels of repetition in EHR sequence data. We repeat our analysis in Appendix Figure 8, but first remove patients with less than 20 total clinical events in order to give a more accurate picture of the level of repetition seen in the timelines of patients with "meaningful" levels of engagement with the healthcare system.

### F.2  IRREGULARITY

Irregularity in EHR data arises from uneven time intervals between clinical events for each patient (McDermott et al., 2023). We define three metrics to quantify the irregularity of a given patient's EHR sequence. These metrics help to capture the variability in timing between events, which is critical for models dealing with irregular time intervals.

**Standard deviation of inter-event times:** Let $X_i$ represent the sequence of clinical events for patient $i$. Let $t_{ij}$ represent the timestamp of the $j$-th event in $X_i$. Then the irregularity $I_\sigma^{(i)}$ of patient $i$ using the standard deviation of inter-event times is given by:

$$\Delta t_{ij} = t_{i(j+1)} - t_{ij}, \quad \forall j \in \{1, \dots, |X_i| - 1\}$$

$$\mu_i = \frac{1}{|X_i| - 1} \sum_{j=1}^{|X_i|-1} \Delta t_{ij}$$

$$I_\sigma^{(i)} = \sqrt{\frac{1}{|X_i| - 1} \sum_{j=1}^{|X_i|-1} (\Delta t_{ij} - \mu_i)^2}$$

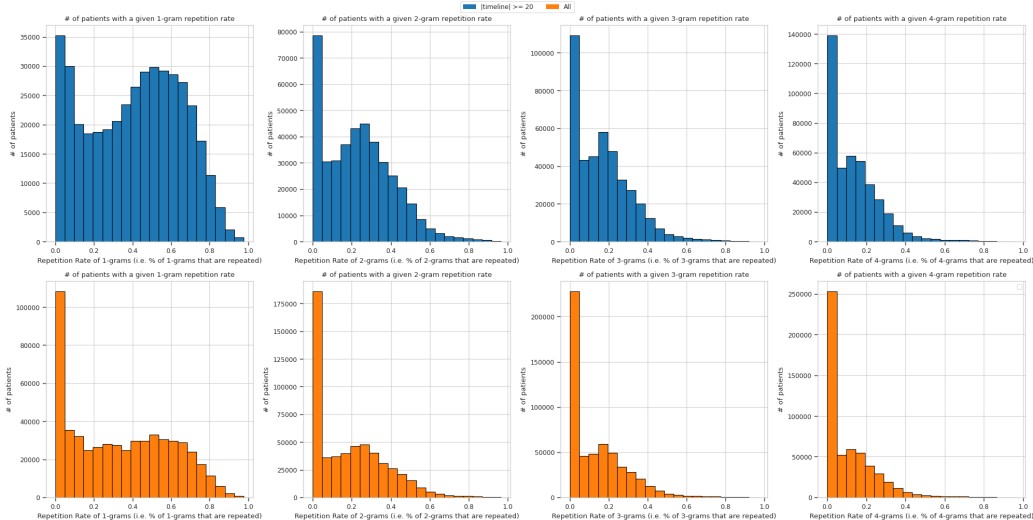

Figure 8: Distribution of $n$-gram repetition rates across patients in the EHR-OMOP validation set. We repeat our analysis from Figure 3 in the main text (reproduced in the bottom row in orange), but also include a version in which we first filter out all patients with less than 20 total events before generating our plots (top row in blue). This helps to clearly show that patients with "meaningful"-length encounters with the healthcare system tend to have highly repetitive EHR timelines. The x-axis represents the $n$-gram repetition rate (i.e. percentage of $n$-grams that are repeated at least once within a patient's EHR), and the y-axis shows the number of patients in each bin.

**Mean inter-event time:** We can also estimate irregularity as $I_\mu^{(i)}$, which represents the mean time between events and is given by:

$$I_\mu^{(i)} = \frac{1}{|X_i| - 1} \sum_{j=1}^{|X_i|-1} \Delta t_{ij}$$

**Interquartile range (IQR):** We can also estimate irregularity as $I_{IQR}^{(i)}$, which represents the interquartile range of the time intervals between events and is given by:

$$I_{IQR}^{(i)} = Q_{75}(\Delta t_{i1}, \ldots, \Delta t_{i(|X_i|-1)}) - Q_{25}(\Delta t_{i1}, \ldots, \Delta t_{i(|X_i|-1)})$$

where $Q_n(\cdot)$ returns the $n$-th percentile of its arguments.

### F.3 INCREASED TOKEN COMPLEXITY DUE TO DISEASE PROGRESSION

As patients age, their diseases become more complex and varied. Thus, we should expect to see tokens later in a patient's timeline to have higher perlexity than tokens earlier in a patient's timeline. In natural language, the uncertainty of later tokens in a document is reduced by conditioning on all prior tokens, such that later tokens in a prompt typically exhibit substantially lower perplexity than earlier words (Kaplan et al., 2020). We found that this trend did not hold with EHR data, per the experimental set-up described below.

To quantify how the complexity of disease changes over time, we used the median perplexity measured at each token position across patient EHRs. Under our hypothesis of disease progression, later tokens should have higher perplexities, even when conditioning on all prior tokens in a patient's medical history.

Perplexity measures the uncertainty in a model's predictions and is computed as:

$$\text{Perplexity}(x) = \exp\left(-\frac{1}{N}\sum_{i=1}^{N}\log P(x_i \mid x_{<i})\right)$$

Where $x_i$ is the current token and $P(x_i \mid x_{<i})$ is the predicted probability of the token given the preceding tokens.

More specifically, we start by sampling 20,000 patients from the EHR-OMOP validation set and tokenizing their full timelines. We use this set of patients for all of our subsequent evaluations.

We then select one of our trained models (e.g. Llama with a context length of 512). We use this model to run inference on the full length of each of these 20,000 patients' timelines. This yields a perplexity score for every token. For patient timelines that are longer than the model's context window, we use a sliding window of 32 tokens.

After running inference on all 20,000 patients with this model, we then calculate the median perplexity output by the model at each token positions. We use median rather than mean to reduce the influence of outliers, which we found to be problematic in early testing. We use these median perplexity scores as our official measurement for that token position's perplexity under that model. For our plots, we apply an exponential moving average over the past 250 token positions for smoothing.

### F.4 EHRSHOT STRATIFICATION

To stratify model performance on EHRSHOT by the repetitiveness of the underlying patient, we first calculate the 1-gram repetition rate (RR) for each patient in the EHRSHOT test set. After grouping the EHRSHOT test patients by the tasks they belong to, we then stratify the patients within each task by their associated 1-gram RR. We sort patients into 4 quartiles, with Q1 containing patients with the lowest RRs (i.e. the least repetitive patients) and Q4 containing patients with the highest RRs (i.e. the most repetitive patients). For each model and each quartile, we then calculate the average Brier score achieved by that model on all patients within the quartile. This yields one Brier score per quartile per model per task. We chose the Brier score as our performance metric because certain strata exhibited uniform labels, which rendered AUROC calculations infeasible. We repeat this process across all tasks and models.

To obtain a single "Q1" Brier Score for a specific model, we take an unweighted average of the previously calculated mean Brier score for the Q1 patients for each task. We repeat this process for Q2/Q3/Q4 to fill out the full row in the table for a specific model.

For testing the statistical significance of whether two models achieve different Brier scores for the same quartile, we perform 1,000 bootstrap samples over the EHRSHOT test set.

## G  FEW-SHOT LEARNING ON EHRSHOT

We define $k$-shot evaluation of a model $M$ on a specific task $T$ as follows:

1. **Training:** For each task $T$, we sample $k$ positive and $k$ negative examples from the training split of $T$ to train the model $M$.
2. **Validation:** An additional $k$ positive and $k$ negative examples are sampled from $T$'s validation split to tune hyperparameters for $M$ on $T$.
3. **Testing:** The best-performing version of $M$, based on validation results, is evaluated on the entire held-out test split of $T$. AUROC is recorded as the performance metric.

For tasks where the total number of unique positive examples is fewer than $k$, all positive examples are included in the training set, and positive examples are randomly resampled until $k$ training examples are achieved.

### G.1 EXPERIMENTAL SETUP

We considered values of $k \in \{8, 16, 32, 64, 128\}$ for all 14 EHRSHOT tasks, with one exception: for the *Celiac* prediction task, we limited $k \leq 64$ due to the dataset's constraint of only 62 posi-

tive training examples. This approach ensures fairness in evaluating performance across tasks with varying dataset sizes and class imbalances.

## G.2 RESULTS

As shown in Appendix Tables 13, 11, and 12 and Appendix Figure 10, our few-shot learning results indicate that model performance, as measured by AUROC, improves consistently as $k$ increases. Longer-context models, particularly Mamba, demonstrated notable gains even at lower values of $k$, underscoring their robustness in data-limited scenarios. This trend was consistent across most benchmark tasks, underscoring the utility of long-context architectures in low-resource settings. Our key observations are as follows:

- **Performance Gains with Context Length:** Longer context lengths generally led to better performance, with Mamba models achieving the highest AUROC scores across several $k$-shot settings, especially at 16,384 tokens.
- **Impact of Few-Shot Sample Size** ($k$)**:** All models showed improved performance with increasing $k$, but Mamba and Llama benefited more significantly at higher values of $k$ (64 and 128), consistently outperforming other models across tasks.

## H ZERO-SHOT LEARNING ON EHRSHOT

We also evaluate a subset of our models under the **zero-shot** setting, i.e we simply run inference on each model without any finetuning. This offers the practical benefit of not having to train or store any fine-tuned task-specific model heads.

### H.1 EXPERIMENTAL SETUP

We follow the procedure outlined in the ETHOS paper (Renc et al., 2024) for making our zero-shot predictions. In brief, we generate 20 synthetic timelines for each patient at the prediction time, measure the percentage of timelines in which the positive event for a task occurs, and then use that percentage as the probability that the patient experiences that positive event. For our zero-shot evaluations, we choose our two strongest models (Mamba and Llama) at their minimum and maximum context lengths, and evaluate them on three representative EHRSHOT tasks – new diagnosis of hypertension, 30-day readmission, and new diagnosis of acute MI.

### H.2 RESULTS

As shown in Appendix Table 15, our zero-shot results significantly lag behind the performance of our few-shot and finetuned models. None of the zero-shot models beat the prior SOTA model (CLMBR-t-base) on any of the three tasks evaluated. Additionally, results across context lengths appear mixed. This underscores the importance of finetuning for clinical prediction making, and suggests that our training pipeline is not optimally designed for zero-shot evaluations.

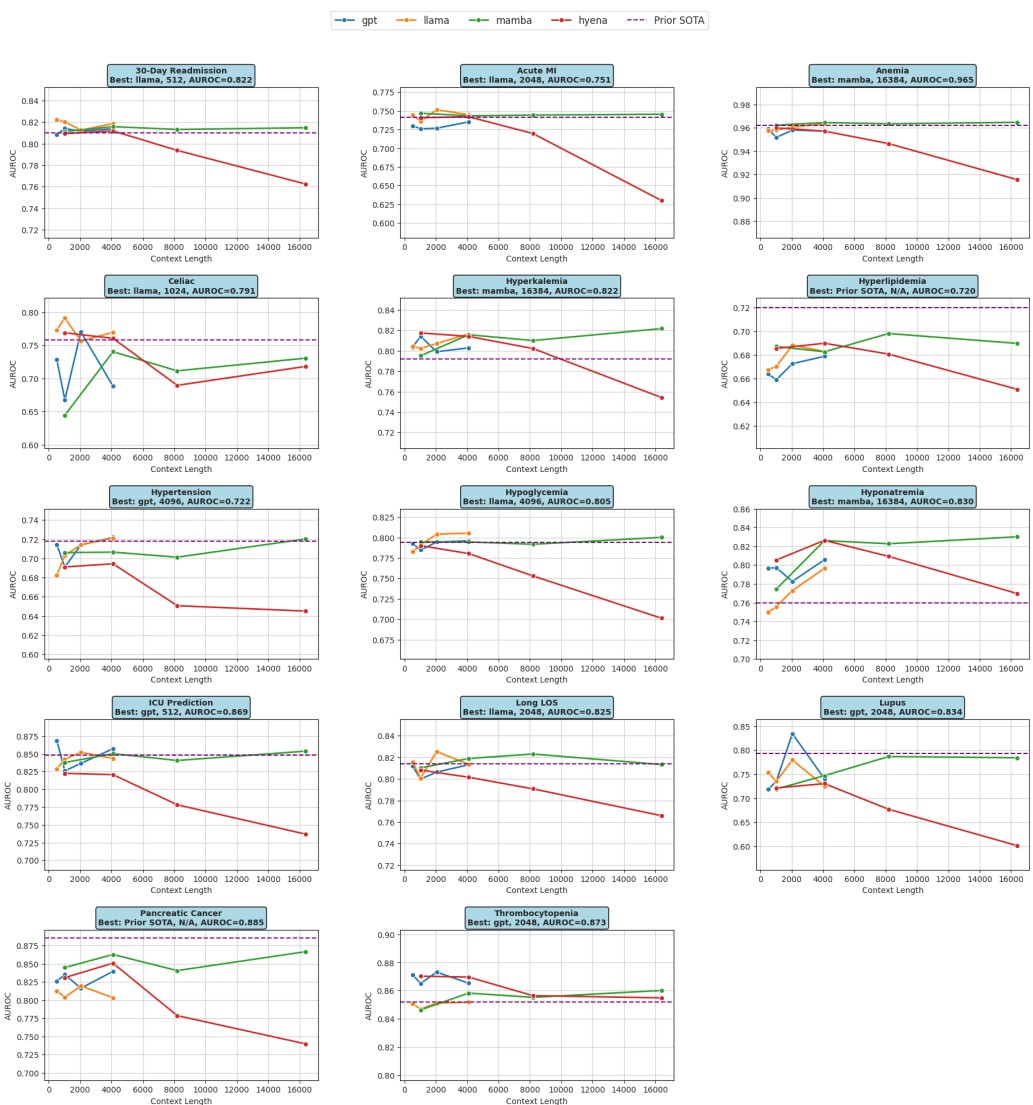

Figure 9: AUROC by context length and architecture across all 14 tasks evaluated from EHRSHOT. The highest scoring model for each task is listed above its plot. Note that the "Prior SOTA" is selected on a task-by-task basis, and thus is not necessarily the same model across plots.

| Model | Context Length | Task | Δ over CLMBR-t-base | 95% CI | Significant |
|---|---|---|---|---|---|
| mamba | 1024 | ICU Admission | -0.009 | (-0.039, 0.019) | |
| mamba | 1024 | Long LOS | -0.003 | (-0.018, 0.010) | |
| mamba | 1024 | 30-day Readmission | 0.001 | (-0.010, 0.013) | |
| mamba | 1024 | Anemia | 0.000 | (-0.001, 0.001) | |
| mamba | 1024 | Hyperkalemia | 0.003 | (-0.006, 0.013) | |
| mamba | 1024 | Hypoglycemia | 0.001 | (-0.011, 0.013) | |
| mamba | 1024 | Hyponatremia | 0.014 | (0.007, 0.022) | ✓ |
| mamba | 1024 | Thrombocytopenia | -0.005 | (-0.010, -0.001) | ✓ |
| mamba | 1024 | Acute MI | 0.017 | (-0.007, 0.040) | |
| mamba | 1024 | Celiac | 0.102 | (-0.076, 0.262) | |
| mamba | 1024 | Hyperlipidemia | 0.020 | (-0.010, 0.050) | |
| mamba | 1024 | Hypertension | -0.011 | (-0.034, 0.011) | |
| mamba | 1024 | Lupus | -0.030 | (-0.115, 0.052) | |
| mamba | 1024 | Pancreatic Cancer | 0.032 | (-0.008, 0.071) | |
| mamba | 4096 | ICU Admission | 0.004 | (-0.024, 0.029) | |
| mamba | 4096 | Long LOS | 0.005 | (-0.010, 0.021) | |
| mamba | 4096 | 30-day Readmission | 0.006 | (-0.006, 0.017) | |
| mamba | 4096 | Anemia | 0.002 | (0.001, 0.003) | ✓ |
| mamba | 4096 | Hyperkalemia | 0.024 | (0.014, 0.034) | ✓ |
| mamba | 4096 | Hypoglycemia | 0.001 | (-0.012, 0.013) | |
| mamba | 4096 | Hyponatremia | 0.066 | (0.057, 0.075) | ✓ |
| mamba | 4096 | Thrombocytopenia | 0.007 | (0.002, 0.011) | ✓ |
| mamba | 4096 | Acute MI | 0.014 | (-0.009, 0.036) | |
| mamba | 4096 | Celiac | 0.198 | (0.115, 0.288) | ✓ |
| mamba | 4096 | Hyperlipidemia | 0.015 | (-0.034, 0.057) | |
| mamba | 4096 | Hypertension | -0.010 | (-0.033, 0.010) | |
| mamba | 4096 | Lupus | -0.003 | (-0.091, 0.086) | |
| mamba | 4096 | Pancreatic Cancer | 0.049 | (0.017, 0.081) | ✓ |
| mamba | 8192 | ICU Admission | -0.007 | (-0.033, 0.018) | |
| mamba | 8192 | Long LOS | 0.009 | (-0.006, 0.024) | |
| mamba | 8192 | 30-day Readmission | 0.003 | (-0.010, 0.016) | |
| mamba | 8192 | Anemia | 0.001 | (0.000, 0.002) | ✓ |
| mamba | 8192 | Hyperkalemia | 0.018 | (0.008, 0.029) | ✓ |
| mamba | 8192 | Hypoglycemia | -0.002 | (-0.014, 0.010) | |
| mamba | 8192 | Hyponatremia | 0.063 | (0.053, 0.072) | ✓ |
| mamba | 8192 | Thrombocytopenia | 0.004 | (-0.001, 0.008) | |
| mamba | 8192 | Acute MI | 0.014 | (-0.008, 0.036) | |
| mamba | 8192 | Celiac | 0.173 | (0.083, 0.312) | ✓ |
| mamba | 8192 | Hyperlipidemia | 0.030 | (-0.011, 0.068) | |
| mamba | 8192 | Hypertension | -0.016 | (-0.036, 0.003) | |
| mamba | 8192 | Lupus | 0.038 | (-0.029, 0.113) | |
| mamba | 8192 | Pancreatic Cancer | 0.027 | (-0.010, 0.062) | |
| mamba | 16384 | ICU Admission | 0.007 | (-0.028, 0.040) | |
| mamba | 16384 | Long LOS | 0.013 | (-0.005, 0.029) | |
| mamba | 16384 | 30-day Readmission | 0.005 | (-0.008, 0.017) | |
| mamba | 16384 | Anemia | 0.002 | (0.001, 0.003) | ✓ |
| mamba | 16384 | Hyperkalemia | 0.030 | (0.019, 0.042) | ✓ |
| mamba | 16384 | Hypoglycemia | 0.006 | (-0.006, 0.019) | |
| mamba | 16384 | Hyponatremia | 0.070 | (0.061, 0.079) | ✓ |
| mamba | 16384 | Thrombocytopenia | 0.008 | (0.004, 0.013) | ✓ |
| mamba | 16384 | Acute MI | 0.016 | (-0.005, 0.036) | |
| mamba | 16384 | Celiac | 0.194 | (0.108, 0.333) | ✓ |
| mamba | 16384 | Hyperlipidemia | 0.023 | (-0.013, 0.058) | |
| mamba | 16384 | Hypertension | 0.003 | (-0.018, 0.023) | |
| mamba | 16384 | Lupus | 0.037 | (-0.056, 0.132) | |
| mamba | 16384 | Pancreatic Cancer | 0.053 | (0.024, 0.087) | ✓ |

Table 7: Performance of Mamba across all context lengths on the 14 EHRSHOT tasks. The column "Δ over CLMBR-t-base" contains the increase in AUROC relative to CLMBR-t-base, the prior SOTA model on EHRSHOT. The column "95% CI" contains a bootstrapped confidence interval calculated over 1,000 samples of the test set. The column "Significant" contains a checkmark if the CI does not intersect with 0.

| Model | Context Length | Task | Δ over CLMBR-t-base | 95% CI | Significant |
|-------|---------------|------|---------------------|--------|-------------|
| llama | 512 | ICU Admission | -0.018 | (-0.052, 0.015) | |
| llama | 512 | Long LOS | 0.002 | (-0.014, 0.017) | |
| llama | 512 | 30-day Readmission | 0.012 | (0.000, 0.024) | ✓ |
| llama | 512 | Anemia | -0.004 | (-0.005, -0.003) | ✓ |
| llama | 512 | Hyperkalemia | 0.012 | (0.004, 0.020) | ✓ |
| llama | 512 | Hypoglycemia | -0.011 | (-0.022, 0.001) | |
| llama | 512 | Hyponatremia | -0.010 | (-0.016, -0.004) | ✓ |
| llama | 512 | Thrombocytopenia | -0.001 | (-0.006, 0.004) | |
| llama | 512 | Acute MI | 0.015 | (-0.006, 0.037) | |
| llama | 512 | Celiac | 0.227 | (0.111, 0.356) | ✓ |
| llama | 512 | Hyperlipidemia | 0.001 | (-0.018, 0.020) | |
| llama | 512 | Hypertension | -0.035 | (-0.057, -0.012) | ✓ |
| llama | 512 | Lupus | 0.005 | (-0.084, 0.095) | |
| llama | 512 | Pancreatic Cancer | 0.001 | (-0.044, 0.046) | |
| llama | 1024 | ICU Admission | -0.005 | (-0.042, 0.032) | |
| llama | 1024 | Long LOS | -0.013 | (-0.034, 0.005) | |
| llama | 1024 | 30-day Readmission | 0.010 | (-0.002, 0.024) | |
| llama | 1024 | Anemia | -0.004 | (-0.005, -0.003) | ✓ |
| llama | 1024 | Hyperkalemia | 0.010 | (0.002, 0.019) | ✓ |
| llama | 1024 | Hypoglycemia | -0.003 | (-0.014, 0.008) | |
| llama | 1024 | Hyponatremia | -0.004 | (-0.010, 0.001) | |
| llama | 1024 | Thrombocytopenia | -0.005 | (-0.009, -0.000) | ✓ |
| llama | 1024 | Acute MI | 0.007 | (-0.014, 0.029) | |
| llama | 1024 | Celiac | 0.250 | (0.149, 0.359) | ✓ |
| llama | 1024 | Hyperlipidemia | 0.003 | (-0.016, 0.021) | |
| llama | 1024 | Hypertension | -0.014 | (-0.033, 0.003) | |
| llama | 1024 | Lupus | -0.014 | (-0.102, 0.079) | |
| llama | 1024 | Pancreatic Cancer | -0.007 | (-0.053, 0.037) | |
| llama | 2048 | ICU Admission | 0.005 | (-0.023, 0.033) | |
| llama | 2048 | Long LOS | 0.014 | (-0.003, 0.029) | |
| llama | 2048 | 30-day Readmission | 0.010 | (-0.003, 0.023) | |
| llama | 2048 | Anemia | -0.002 | (-0.003, -0.001) | ✓ |
| llama | 2048 | Hyperkalemia | 0.015 | (0.005, 0.025) | ✓ |
| llama | 2048 | Hypoglycemia | 0.011 | (-0.002, 0.023) | |
| llama | 2048 | Hyponatremia | 0.013 | (0.005, 0.020) | ✓ |
| llama | 2048 | Thrombocytopenia | -0.000 | (-0.006, 0.004) | |
| llama | 2048 | Acute MI | 0.022 | (-0.001, 0.044) | |
| llama | 2048 | Celiac | 0.212 | (0.083, 0.343) | ✓ |
| llama | 2048 | Hyperlipidemia | 0.021 | (-0.005, 0.049) | |
| llama | 2048 | Hypertension | -0.003 | (-0.025, 0.018) | |
| llama | 2048 | Lupus | 0.031 | (-0.049, 0.119) | |
| llama | 2048 | Pancreatic Cancer | 0.007 | (-0.042, 0.053) | |
| llama | 4096 | ICU Admission | -0.003 | (-0.026, 0.021) | |
| llama | 4096 | Long LOS | -0.004 | (-0.018, 0.010) | |
| llama | 4096 | 30-day Readmission | 0.013 | (0.002, 0.026) | ✓ |
| llama | 4096 | Anemia | 0.001 | (0.000, 0.002) | ✓ |
| llama | 4096 | Hyperkalemia | 0.024 | (0.016, 0.033) | ✓ |
| llama | 4096 | Hypoglycemia | 0.012 | (-0.000, 0.022) | |
| llama | 4096 | Hyponatremia | 0.036 | (0.028, 0.046) | ✓ |
| llama | 4096 | Thrombocytopenia | 0.000 | (-0.004, 0.005) | |
| llama | 4096 | Acute MI | 0.015 | (-0.008, 0.038) | |
| llama | 4096 | Celiac | 0.226 | (0.097, 0.365) | ✓ |
| llama | 4096 | Hyperlipidemia | 0.016 | (-0.002, 0.036) | |
| llama | 4096 | Hypertension | 0.004 | (-0.013, 0.021) | |
| llama | 4096 | Lupus | -0.023 | (-0.097, 0.049) | |
| llama | 4096 | Pancreatic Cancer | -0.008 | (-0.056, 0.033) | |

Table 8: Performance of Llama across all context lengths on the 14 EHRSHOT tasks. The column "Δ over CLMBR-t-base" contains the increase in AUROC relative to CLMBR-t-base, the prior SOTA model on EHRSHOT. The column "95% CI" contains a bootstrapped confidence interval calculated over 1,000 samples of the test set. The column "Significant" contains a checkmark if the CI does not intersect with 0.

| Model | Context Length | Task | Δ over CLMBR-t-base | 95% CI | Significant |
|---|---|---|---|---|---|
| gpt2 | 512 | ICU Admission | 0.022 | (-0.005, 0.050) | |
| gpt2 | 512 | Long LOS | -0.002 | (-0.017, 0.012) | |
| gpt2 | 512 | 30-day Readmission | -0.002 | (-0.013, 0.009) | |
| gpt2 | 512 | Anemia | -0.003 | (-0.004, -0.002) | ✓ |
| gpt2 | 512 | Hyperkalemia | 0.011 | (0.001, 0.021) | ✓ |
| gpt2 | 512 | Hypoglycemia | -0.001 | (-0.014, 0.012) | |
| gpt2 | 512 | Hyponatremia | 0.037 | (0.028, 0.046) | ✓ |
| gpt2 | 512 | Thrombocytopenia | 0.020 | (0.015, 0.025) | ✓ |
| gpt2 | 512 | Acute MI | 0.001 | (-0.022, 0.027) | |
| gpt2 | 512 | Celiac | 0.181 | (0.063, 0.295) | ✓ |
| gpt2 | 512 | Hyperlipidemia | -0.004 | (-0.047, 0.043) | |
| gpt2 | 512 | Hypertension | -0.003 | (-0.021, 0.014) | |
| gpt2 | 512 | Lupus | -0.031 | (-0.110, 0.050) | |
| gpt2 | 512 | Pancreatic Cancer | 0.014 | (-0.028, 0.054) | |
| gpt2 | 1024 | ICU Admission | -0.021 | (-0.052, 0.009) | |
| gpt2 | 1024 | Long LOS | -0.014 | (-0.032, 0.004) | |
| gpt2 | 1024 | 30-day Readmission | 0.004 | (-0.009, 0.015) | |
| gpt2 | 1024 | Anemia | -0.011 | (-0.012, -0.009) | ✓ |
| gpt2 | 1024 | Hyperkalemia | 0.022 | (0.011, 0.033) | ✓ |
| gpt2 | 1024 | Hypoglycemia | -0.009 | (-0.022, 0.004) | |
| gpt2 | 1024 | Hyponatremia | 0.037 | (0.028, 0.046) | ✓ |
| gpt2 | 1024 | Thrombocytopenia | 0.013 | (0.009, 0.019) | ✓ |
| gpt2 | 1024 | Acute MI | -0.003 | (-0.027, 0.021) | |
| gpt2 | 1024 | Celiac | 0.125 | (0.007, 0.274) | ✓ |
| gpt2 | 1024 | Hyperlipidemia | -0.008 | (-0.053, 0.036) | |
| gpt2 | 1024 | Hypertension | -0.026 | (-0.049, -0.005) | ✓ |
| gpt2 | 1024 | Lupus | -0.016 | (-0.090, 0.062) | |
| gpt2 | 1024 | Pancreatic Cancer | 0.022 | (-0.009, 0.050) | |
| gpt2 | 2048 | ICU Admission | -0.010 | (-0.040, 0.021) | |
| gpt2 | 2048 | Long LOS | -0.008 | (-0.022, 0.006) | |
| gpt2 | 2048 | 30-day Readmission | 0.002 | (-0.011, 0.014) | |
| gpt2 | 2048 | Anemia | -0.004 | (-0.005, -0.003) | ✓ |
| gpt2 | 2048 | Hyperkalemia | 0.007 | (-0.003, 0.017) | |
| gpt2 | 2048 | Hypoglycemia | 0.001 | (-0.013, 0.013) | |
| gpt2 | 2048 | Hyponatremia | 0.023 | (0.015, 0.029) | ✓ |
| gpt2 | 2048 | Thrombocytopenia | 0.021 | (0.016, 0.027) | ✓ |
| gpt2 | 2048 | Acute MI | -0.003 | (-0.030, 0.024) | |
| gpt2 | 2048 | Celiac | 0.227 | (0.037, 0.433) | ✓ |
| gpt2 | 2048 | Hyperlipidemia | 0.005 | (-0.014, 0.025) | |
| gpt2 | 2048 | Hypertension | -0.002 | (-0.021, 0.017) | |
| gpt2 | 2048 | Lupus | 0.085 | (0.005, 0.165) | ✓ |
| gpt2 | 2048 | Pancreatic Cancer | 0.004 | (-0.032, 0.037) | |
| gpt2 | 4096 | ICU Admission | 0.011 | (-0.021, 0.044) | |
| gpt2 | 4096 | Long LOS | -0.001 | (-0.014, 0.014) | |
| gpt2 | 4096 | 30-day Readmission | 0.004 | (-0.009, 0.015) | |
| gpt2 | 4096 | Anemia | -0.005 | (-0.006, -0.004) | ✓ |
| gpt2 | 4096 | Hyperkalemia | 0.011 | (0.001, 0.021) | ✓ |
| gpt2 | 4096 | Hypoglycemia | 0.003 | (-0.011, 0.015) | |
| gpt2 | 4096 | Hyponatremia | 0.046 | (0.036, 0.055) | ✓ |
| gpt2 | 4096 | Thrombocytopenia | 0.014 | (0.009, 0.018) | ✓ |
| gpt2 | 4096 | Acute MI | 0.006 | (-0.022, 0.033) | |
| gpt2 | 4096 | Celiac | 0.149 | (0.041, 0.278) | ✓ |
| gpt2 | 4096 | Hyperlipidemia | 0.012 | (-0.018, 0.043) | |
| gpt2 | 4096 | Hypertension | 0.004 | (-0.015, 0.024) | |
| gpt2 | 4096 | Lupus | -0.008 | (-0.095, 0.088) | |
| gpt2 | 4096 | Pancreatic Cancer | 0.027 | (-0.008, 0.062) | |

Table 9: Performance of GPT across all context lengths on the 14 EHRSHOT tasks. The column "Δ over CLMBR-t-base" contains the increase in AUROC relative to CLMBR-t-base, the prior SOTA model on EHRSHOT. The column "95% CI" contains a bootstrapped confidence interval calculated over 1,000 samples of the test set. The column "Significant" contains a checkmark if the CI does not intersect with 0.

| Model | Context Length | Task | Δ over CLMBR-t-base | 95% CI | Significant |
|---|---|---|---|---|---|
| hyena | 1024 | ICU Admission | -0.026 | (-0.064, 0.013) | |
| hyena | 1024 | Long LOS | -0.006 | (-0.020, 0.011) | |
| hyena | 1024 | 30-day Readmission | -0.001 | (-0.012, 0.010) | |
| hyena | 1024 | Anemia | -0.002 | (-0.003, -0.001) | ✓ |
| hyena | 1024 | Hyperkalemia | 0.026 | (0.015, 0.036) | ✓ |
| hyena | 1024 | Hypoglycemia | -0.004 | (-0.015, 0.008) | |
| hyena | 1024 | Hyponatremia | 0.045 | (0.036, 0.055) | ✓ |
| hyena | 1024 | Thrombocytopenia | 0.019 | (0.014, 0.024) | ✓ |
| hyena | 1024 | Acute MI | 0.011 | (-0.015, 0.038) | |
| hyena | 1024 | Celiac | 0.224 | (0.095, 0.367) | ✓ |
| hyena | 1024 | Hyperlipidemia | 0.018 | (-0.000, 0.037) | |
| hyena | 1024 | Hypertension | -0.026 | (-0.053, -0.003) | ✓ |
| hyena | 1024 | Lupus | -0.026 | (-0.116, 0.055) | |
| hyena | 1024 | Pancreatic Cancer | 0.019 | (-0.022, 0.060) | |
| hyena | 4096 | ICU Admission | -0.026 | (-0.058, 0.004) | |
| hyena | 4096 | Long LOS | -0.012 | (-0.030, 0.006) | |
| hyena | 4096 | 30-day Readmission | 0.002 | (-0.012, 0.013) | |
| hyena | 4096 | Anemia | -0.005 | (-0.006, -0.004) | ✓ |
| hyena | 4096 | Hyperkalemia | 0.022 | (0.013, 0.033) | ✓ |
| hyena | 4096 | Hypoglycemia | -0.013 | (-0.027, 0.001) | |
| hyena | 4096 | Hyponatremia | 0.066 | (0.056, 0.078) | ✓ |
| hyena | 4096 | Thrombocytopenia | 0.018 | (0.013, 0.023) | ✓ |
| hyena | 4096 | Acute MI | 0.013 | (-0.013, 0.040) | |
| hyena | 4096 | Celiac | 0.216 | (0.077, 0.370) | ✓ |
| hyena | 4096 | Hyperlipidemia | 0.023 | (-0.012, 0.057) | |
| hyena | 4096 | Hypertension | -0.023 | (-0.050, 0.002) | |
| hyena | 4096 | Lupus | -0.019 | (-0.110, 0.056) | |
| hyena | 4096 | Pancreatic Cancer | 0.038 | (-0.011, 0.092) | |
| hyena | 8192 | ICU Admission | -0.069 | (-0.106, -0.032) | ✓ |
| hyena | 8192 | Long LOS | -0.023 | (-0.041, -0.004) | ✓ |
| hyena | 8192 | 30-day Readmission | -0.017 | (-0.033, -0.002) | ✓ |
| hyena | 8192 | Anemia | -0.016 | (-0.018, -0.014) | ✓ |
| hyena | 8192 | Hyperkalemia | 0.010 | (0.000, 0.022) | ✓ |
| hyena | 8192 | Hypoglycemia | -0.041 | (-0.056, -0.025) | ✓ |
| hyena | 8192 | Hyponatremia | 0.049 | (0.039, 0.059) | ✓ |
| hyena | 8192 | Thrombocytopenia | 0.005 | (-0.001, 0.010) | |
| hyena | 8192 | Acute MI | -0.009 | (-0.038, 0.022) | |
| hyena | 8192 | Celiac | 0.154 | (-0.013, 0.352) | |
| hyena | 8192 | Hyperlipidemia | 0.014 | (-0.026, 0.052) | |
| hyena | 8192 | Hypertension | -0.066 | (-0.108, -0.030) | ✓ |
| hyena | 8192 | Lupus | -0.073 | (-0.189, 0.025) | |
| hyena | 8192 | Pancreatic Cancer | -0.033 | (-0.088, 0.018) | |
| hyena | 16384 | ICU Admission | -0.110 | (-0.147, -0.075) | ✓ |
| hyena | 16384 | Long LOS | -0.048 | (-0.068, -0.029) | ✓ |
| hyena | 16384 | 30-day Readmission | -0.048 | (-0.067, -0.026) | ✓ |
| hyena | 16384 | Anemia | -0.047 | (-0.051, -0.043) | ✓ |
| hyena | 16384 | Hyperkalemia | -0.038 | (-0.054, -0.023) | ✓ |
| hyena | 16384 | Hypoglycemia | -0.093 | (-0.109, -0.075) | ✓ |
| hyena | 16384 | Hyponatremia | 0.010 | (-0.002, 0.021) | |
| hyena | 16384 | Thrombocytopenia | 0.003 | (-0.005, 0.011) | |
| hyena | 16384 | Acute MI | -0.100 | (-0.145, -0.053) | ✓ |
| hyena | 16384 | Celiac | 0.176 | (0.029, 0.318) | ✓ |
| hyena | 16384 | Hyperlipidemia | -0.016 | (-0.069, 0.034) | |
| hyena | 16384 | Hypertension | -0.071 | (-0.125, -0.023) | ✓ |
| hyena | 16384 | Lupus | -0.145 | (-0.268, -0.017) | ✓ |
| hyena | 16384 | Pancreatic Cancer | -0.073 | (-0.148, 0.006) | |

Table 10: Performance of Hyena across all context lengths on the 14 EHRSHOT tasks. The column "Δ over CLMBR-t-base" contains the increase in AUROC relative to CLMBR-t-base, the prior SOTA model on EHRSHOT. The column "95% CI" contains a bootstrapped confidence interval calculated over 1,000 samples of the test set. The column "Significant" contains a checkmark if the CI does not intersect with 0.

| Model | Context Length | k | | | | | |
|---|---|---|---|---|---|---|---|
| | | 8 | 16 | 32 | 64 | 128 | All |
| gpt2 | 512 | **0.661** | **0.714** | **0.747** | **0.779** | **0.794** | **0.830** |
| gpt2 | 1024 | 0.634 | 0.697 | 0.732 | 0.758 | 0.774 | 0.813 |
| gpt2 | 2048 | 0.654 | 0.704 | 0.743 | 0.771 | 0.792 | 0.818 |
| gpt2 | 4096 | 0.657 | 0.706 | 0.742 | 0.769 | 0.791 | 0.828 |
| llama | 512 | 0.672 | **0.716** | 0.741 | 0.767 | 0.786 | 0.822 |
| llama | 1024 | 0.662 | 0.707 | 0.737 | 0.769 | 0.788 | 0.821 |
| llama | 2048 | **0.674** | 0.714 | **0.757** | **0.784** | 0.799 | **_0.833_** |
| llama | 4096 | 0.665 | 0.709 | 0.756 | 0.782 | **0.800** | 0.826 |
| mamba | 1024 | 0.668 | 0.719 | 0.745 | 0.774 | 0.786 | 0.820 |
| mamba | 4096 | 0.681 | 0.730 | 0.754 | 0.784 | 0.796 | 0.828 |
| mamba | 8192 | 0.676 | 0.728 | 0.753 | 0.782 | 0.800 | 0.826 |
| mamba | 16384 | **_0.685_** | **_0.734_** | **_0.761_** | **_0.791_** | **_0.804_** | **0.831** |
| hyena | 1024 | **0.655** | **0.705** | **0.739** | **0.761** | **0.778** | **0.813** |
| hyena | 4096 | 0.631 | 0.681 | 0.725 | 0.747 | 0.773 | 0.811 |
| hyena | 8192 | 0.622 | 0.669 | 0.698 | 0.727 | 0.750 | 0.788 |
| hyena | 16384 | 0.587 | 0.629 | 0.651 | 0.676 | 0.705 | 0.755 |

Table 11: **Few-Shot Evaluation:** Average AUROC score for each model and context length across all *Operational Outcomes* tasks and $k$-shot settings. The highest AUROC across all models for each $k$ is **bolded underlined**, and the maximum value within each model across context lengths for each $k$ is **bolded**.

| Model | Context Length | k | | | | | |
|---|---|---|---|---|---|---|---|
| | | 8 | 16 | 32 | 64 | 128 | All |
| gpt2 | 512 | 0.603 | 0.634 | 0.670 | 0.695 | 0.713 | 0.730 |
| gpt2 | 1024 | 0.610 | 0.644 | 0.672 | 0.691 | 0.711 | 0.719 |
| gpt2 | 2048 | **0.621** | **0.654** | **0.684** | **0.709** | **0.726** | **0.756** |
| gpt2 | 4096 | 0.616 | 0.642 | 0.678 | 0.700 | 0.722 | 0.734 |
| llama | 512 | 0.606 | 0.635 | 0.665 | 0.687 | 0.721 | 0.739 |
| llama | 1024 | 0.615 | 0.644 | 0.670 | 0.692 | 0.708 | 0.740 |
| llama | 2048 | **0.624** | **0.653** | 0.675 | 0.694 | **0.728** | **0.751** |
| llama | 4096 | 0.621 | 0.646 | **0.679** | **0.695** | 0.721 | 0.741 |
| mamba | 1024 | 0.628 | 0.652 | 0.682 | 0.698 | 0.716 | 0.725 |
| mamba | 4096 | 0.630 | 0.658 | 0.689 | 0.704 | 0.726 | 0.747 |
| mamba | 8192 | 0.633 | 0.657 | 0.690 | 0.706 | 0.723 | 0.747 |
| mamba | 16384 | **_0.647_** | **_0.668_** | **_0.698_** | **_0.711_** | **_0.732_** | **_0.756_** |
| hyena | 1024 | **0.621** | **0.651** | **0.682** | **0.697** | **0.717** | 0.740 |
| hyena | 4096 | 0.608 | 0.638 | 0.666 | 0.680 | 0.709 | **0.745** |
| hyena | 8192 | 0.585 | 0.608 | 0.638 | 0.657 | 0.671 | 0.699 |
| hyena | 16384 | 0.540 | 0.553 | 0.578 | 0.597 | 0.636 | 0.664 |

Table 12: **Few-Shot Evaluation:** Average AUROC score for each model and context length across all *Assignment of New Diagnoses* tasks and $k$-shot settings. The highest AUROC across all models for each $k$ is **bolded underlined**, and the maximum value within each model across context lengths for each $k$ is **bolded**.

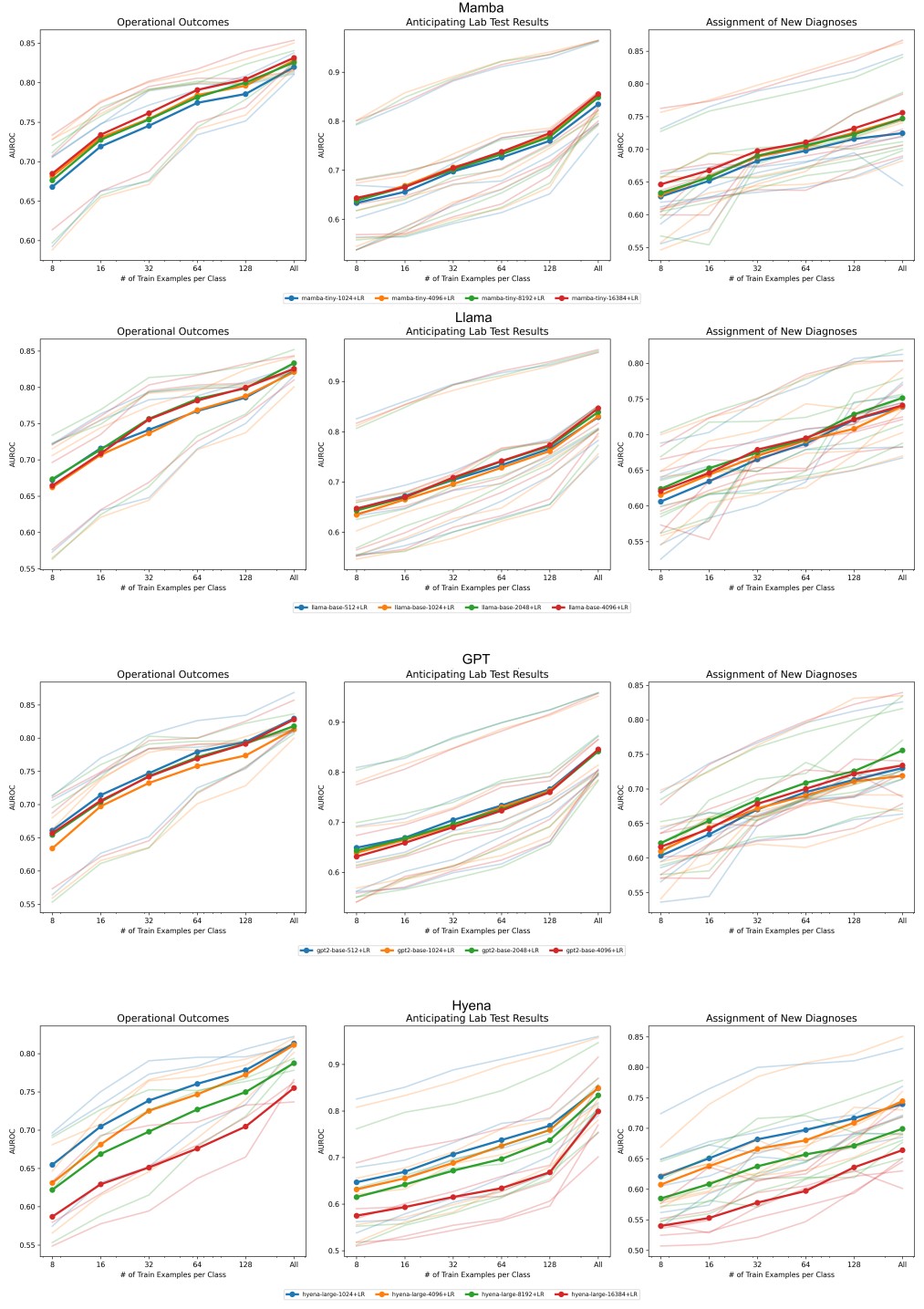

Figure 10: **Few-Shot Evaluation:** Average AUROC scores for each model and context length across all few-shot settings, aggregated for each EHRSHOT clinical prediction task group: *Operational Outcomes*, *Anticipating Lab Test Results*, and *Assignment of New Diagnoses*. Each row is a different model (from top to bottom: Mamba, Llama, GPT, Hyena) and each column is a task group. The x-axis shows the number of few-shot examples ($k$-shot), while the y-axis displays AUROC. Each line represents a different context length. Solid lines are AUROCs average across all subtasks within a task group, while lighter lines are the few-shot results for each individual subtask.

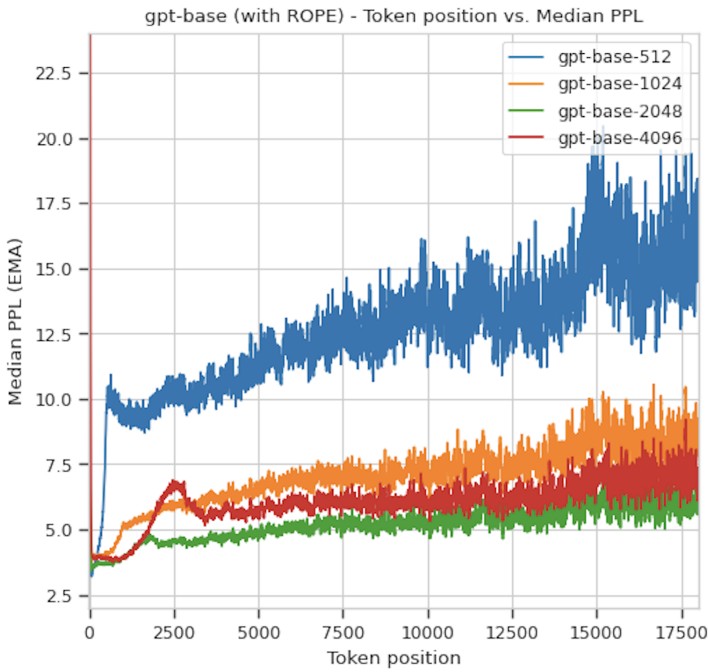

Figure 11: Reproduction of Figure 4 for the GPT architecture, but with rotary positional embeddings (ROPE) instead of absolute positional embeddings. All other aspects of the GPT architecture are kept the same. With ROPE, the perplexity curves appear more stable and do not exhibit the 10+ point perplexity spikes seen in Figure 4, but still mirror the trend of increased perplexity with increased sequence length.

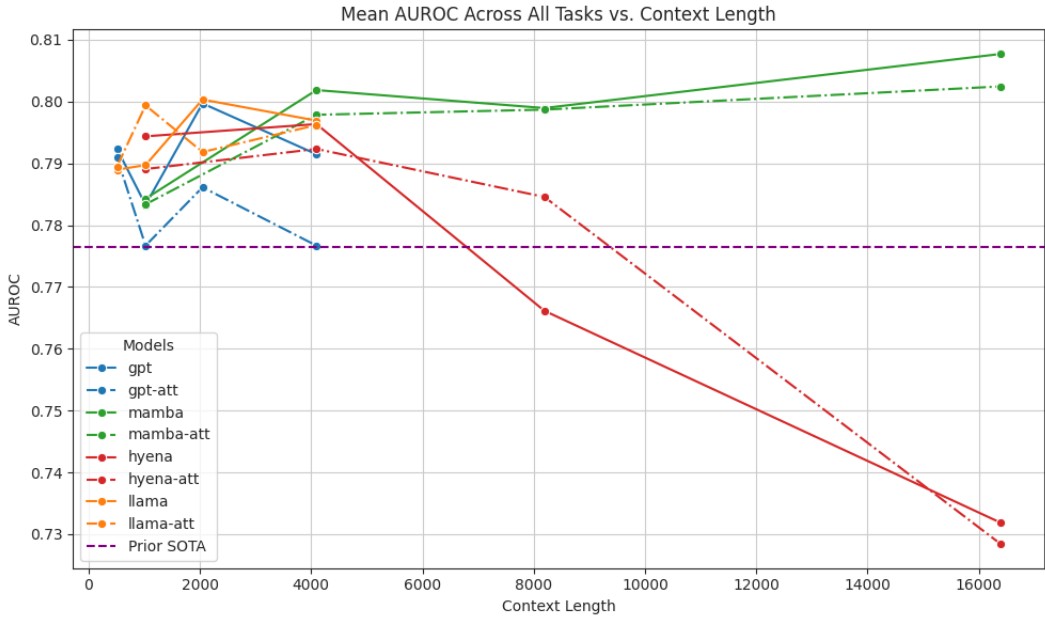

Figure 12: Reproduction of Figure 1b, but with models trained using **Artificial Time Tokens (ATTs)** (as defined in CEHR-BERT (Pang et al., 2021)) shown in dotted lines, and models trained without ATTs in solid lines. Overall, we see better performance without using ATT tokens. While the dotted lines closely follow the solid lines for Mamba and Hyena, the transformer models appear to have less stable performance at smaller contexts, potentially due to the injection of more tokens within each patient's timeline.

| Model | Context Length | k | | | | | |
|---|---|---|---|---|---|---|---|
| | | 8 | 16 | 32 | 64 | 128 | All |
| gpt2 | 512 | **0.649** | **0.669** | **0.704** | **0.733** | **0.766** | **0.845** |
| gpt2 | 1024 | 0.639 | 0.665 | 0.694 | 0.730 | 0.763 | 0.843 |
| gpt2 | 2048 | 0.643 | 0.667 | 0.696 | 0.726 | 0.761 | 0.841 |
| gpt2 | 4096 | 0.631 | 0.659 | 0.690 | 0.723 | 0.760 | **0.845** |
| llama | 512 | **0.647** | **0.672** | 0.704 | 0.733 | 0.767 | 0.829 |
| llama | 1024 | 0.635 | 0.665 | 0.696 | 0.728 | 0.762 | 0.831 |
| llama | 2048 | 0.643 | 0.669 | 0.707 | 0.741 | 0.772 | 0.839 |
| llama | 4096 | **0.647** | 0.670 | **0.709** | **0.742** | **0.773** | **0.847** |
| mamba | 1024 | 0.633 | 0.656 | 0.698 | 0.726 | 0.760 | 0.835 |
| mamba | 4096 | 0.640 | **0.669** | **0.706** | 0.734 | 0.770 | 0.852 |
| mamba | 8192 | 0.638 | 0.666 | 0.701 | 0.733 | 0.768 | 0.849 |
| mamba | 16384 | **0.644** | 0.666 | 0.705 | **0.738** | **0.776** | **0.855** |
| hyena | 1024 | **0.647** | **0.669** | **0.707** | **0.737** | **0.768** | 0.849 |
| hyena | 4096 | 0.632 | 0.655 | 0.688 | 0.725 | 0.759 | **0.850** |
| hyena | 8192 | 0.615 | 0.642 | 0.672 | 0.697 | 0.737 | 0.833 |
| hyena | 16384 | 0.575 | 0.594 | 0.615 | 0.634 | 0.668 | 0.799 |

Table 13: **Few-Shot Evaluation:** Average AUROC score for each model and context length across all *Anticipating Lab Test Results* tasks and $k$-shot settings. The highest AUROC across all models for each $k$ is **bolded underlined**, and the maximum value within each model across context lengths for each $k$ is **bolded**.

| Metric | Model | Context Length | Q1 | Q2 | Q3 | Q4 |
|---|---|---|---|---|---|---|
| Repetitiveness (1-gram RR) | Mamba | 1k | 0.0644 | 0.0737 | 0.0744 | 0.0790 |
| | | 16k | **0.0605** | **0.0670** | **0.0700** | **0.0746** |
| | Llama | 512 | 0.0640 | 0.0710 | 0.0743 | 0.0792 |
| | | 4k | **0.0627** | **0.0687** | **0.0721** | **0.0770** |
| | GPT | 512 | 0.0619 | 0.0691 | 0.0710 | 0.0765 |
| | | 4k | 0.0643 | 0.0692 | 0.0711 | 0.0765 |
| | Hyena | 1k | 0.0636 | 0.0681 | 0.0718 | 0.0776 |
| | | 16k | 0.0733 | 0.0759 | 0.0780 | 0.0822 |
| | CLMBR-t-base | 512 | 0.0647 | 0.0719 | 0.0751 | 0.0805 |
| Irregularity (Standard Deviation) | Mamba | 1k | 0.0693 | 0.0729 | 0.0731 | 0.0764 |
| | | 16k | **0.0641** | **0.0678** | **0.0679** | **0.0723** |
| | Llama | 512 | 0.0694 | 0.0730 | 0.0713 | 0.0749 |
| | | 4k | **0.0664** | **0.0705** | **0.0694** | 0.0740 |
| | GPT | 512 | 0.0654 | 0.0693 | 0.0703 | 0.0736 |
| | | 4k | 0.0653 | 0.0699 | 0.0701 | 0.0759 |
| | Hyena | 1k | 0.0666 | 0.0702 | 0.0692 | 0.0751 |
| | | 16k | 0.0698 | 0.0755 | 0.0788 | 0.0853 |
| | CLMBR-t-base | 512 | 0.0683 | 0.0741 | 0.0721 | 0.0777 |

Table 14: Comparison of average Brier scores for all models across all 14 EHRSHOT tasks. Patients are bucketed by repetitiveness (top) and irregularity (bottom). Q1/Q2/Q3/Q4 are the 1st through 4th quartiles of patients ranked by each metric. For example, Q1 contains the least repetitive / least irregular patients while Q4 contains the most repetitive / most irregular patients. **Bolded** values show a statistically significant win rate of at least 50% of the longer context model over the shorter context model at a specific quartile. This is identical to Table 2, but with all models shown.

.

| Model | Context Length | AUROC |
|---|---|---|
| **Hypertension** | | |
| CLMBR-t-base | 512 | **0.718** |
| Mamba | 1024 | 0.660 |
| Llama | 512 | 0.642 |
| Llama | 4096 | 0.609 |
| Mamba | 16384 | 0.563 |
| | | |
| **30-day Readmission** | | |
| CLMBR-t-base | 512 | **0.810** |
| Mamba | 1024 | 0.720 |
| Llama | 4096 | 0.710 |
| Llama | 512 | 0.705 |
| Mamba | 16384 | 0.643 |
| | | |
| **Acute MI** | | |
| CLMBR-t-base | 512 | **0.729** |
| Mamba | 16384 | 0.531 |
| Mamba | 1024 | 0.525 |
| Llama | 4096 | 0.52 |
| Llama | 512 | 0.51 |

Table 15: **Zero-Shot Evaluation:** AUROC scores for each model and context length for zero-shot evaluations across three EHRSHOT clinical prediction tasks. The zero-shot evaluations followed the procedure outlined in (Renc et al., 2024). Namely, 20 synthetic timelines were generated for each patient at each prediction timepoint. The probability that a patient experienced a positive event was calculated as the percentage of generated timelines that contained that positive event within the appropriate time horizon as defined by the relevant task.

.

