# OpenReview forum: "Context Clues: Evaluating Long Context Models for Clinical Prediction Tasks on EHR Data"
_ICLR.cc/2025/Conference — ICLR 2025 Poster_

### Official Review · Reviewer_uVBS · 2024-11-02

**Soundness:** 2
**Presentation:** 3
**Contribution:** 2
**Rating:** 5
**Confidence:** 4

**Summary:**

This paper investigates the advantages of long context models in the healthcare domain by evaluating the impact of context length on clinical prediction tasks using four models—Mamba, Llama, GPT, and Hyena—trained on millions of longitudinal EHR records. Additionally, the study assesses model robustness against three properties of EHR data: (1) "copy-forwarded" diagnoses that lead to artificial token repetition, (2) irregular time intervals between EHR events causing variable timespans within context windows, and (3) the increasing complexity of diseases over time, which complicates the prediction of later tokens compared to earlier ones. These factors highlight challenges associated with EHR data in clinical prediction tasks. The results indicate that a higher prevalence of each property negatively impacts model performance, while models with longer context inputs tend to be more robust (although not consistantly) to these issues.

**Strengths:**

1. The analysis is solid in its technical execution and experimental design
2. Generally, a rich technical/experimental paper

**Weaknesses:**

1.	The analysis is solid in its technical execution and experimental design; however, it does not introduce any new methods, models, or techniques, which limits its novelty.
2.	The importance of long context is questionable given the low frequency of extremely long contexts in EHRs. It seems somewhat expected that providing more information for each sample would improve results.
3.	While the paper states that disease progression increases token complexity over time. I am not very convinced how this property can be problematic with shorter input context compared to longer input context models.
4.	The paper suggests that model performance improves with longer contexts; however, this is not consistently reflected in the figures. If this is indeed the case, it stands to reason that having more information about a patient (i.e., longer context) should facilitate easier predictions.

**Questions:**

Could you elaborate on the paper's novelty?
Is the dataset utilized in the study publicly available?

---

> ### Author Response · Authors · 2024-11-18
> **Author Response**
>
> We sincerely appreciate your comprehensive review and the constructive feedback on our paper. We appreciate your recognition of the “rich technical” work in the paper and its ““solid” execution and experimental design. Your comments have helped us to significantly strengthen our work, and so we appreciate the time taken to provide your feedback. In addition to the Overall Response above, we provide responses to each of your comments below.
>
> ### "The analysis is solid in its technical execution and experimental design; however, it does not introduce any new methods, models, or techniques, which limits its novelty."
>
> We appreciate your recognition of the soundness of our experimental design and execution, while giving us the opportunity to address your concerns regarding the novelty of our work. Please see our **Overall Response** under _“Novelty”_ for details on the novel contributions of this paper, as well as comments below in response to your later question: “Could you elaborate on the paper's novelty?”
>
> ### "The importance of long context is questionable given the low frequency of extremely long contexts in EHRs. It seems somewhat expected that providing more information for each sample would improve results."
>
> Your point is well-taken, thus we have added **Appendix Figure 6** to show the distribution of patient sequence lengths in our evaluation dataset. This Figure shows the CDF of patient sequence lengths for all 14 tasks in the EHRSHOT benchmark. As can be seen across all tasks and splits, the median sequence length per task is between 5k - 10k tokens, and there is a long right tail approaching 100k clinical events per patient.
>
> We agree that most patients in an EHR database will have short timelines, and thus may not require long context models. However, the most “interesting” patients, i.e. the ones with the most need for healthcare, typically have longer sequences of medical history due to their frequent usage of the healthcare system (as can be seen in the characteristics of patients in the EHRSHOT benchmark). Thus, for the type of patient that a health system would most care about making predictions for, long context models appear particularly well-suited.
>
> We thank you for this comment, as this is a key motivating reason for our work which we did not sufficiently explain in our manuscript. Per your suggestion, we have also added the following text to **Section 1 (“Intro”)**
>
> > This is especially true for the sickest patients -- i.e. the ones of most interest to a hospital -- as they typically have high healthcare utilization and thus have very long timelines, as can be seen in the CDF plots of patient sequence length in Appendix Figure 6.
>
> ### "While the paper states that disease progression increases token complexity over time. I am not very convinced how this property can be problematic with shorter input context compared to longer input context models."
>
> A longer context model might be better suited for processing more complex disease tokens than shorter context models for several reasons. We see this in empirically in **Figure 4** -- as a patient’s timeline length increases (i.e. going from left to right on the x-axis) the longest context Llama and Mamba models (red) consistently exhibit lower perplexities than the shorter context models (blue), with the gap increasing as the patient timeline gets longer.
>
> There are several possible reasons for this. First, more complex diseases tend to have longer-term disease trajectories (e.g. some evidence shows that the first incidence of a cancer mutation can occur decades before cancer is ever detected in the human body), and certain diagnosis patterns can only be observed over the span of years (e.g. pancreatic cancer metastasizing to the lung). Thus, ingesting more patient history would allow a model to view the full trajectory of a complex disease, and thus better be able to model the later (more complex) tokens in a patient’s timeline. Second, ingesting more patient history provides a model with more context to make decisions. Thus, even for diseases that do not take years/decades to progress, providing a doctor with more of a patient’s medical history should intuitively lead to better predictions simply due to having more information available.
>
> (cont...)

---

> ### Author Response · Authors · 2024-11-18
> **Author Response (cont.)**
>
> (...cont)
>
> ### "The paper suggests that model performance improves with longer contexts; however, this is not consistently reflected in the figures. If this is indeed the case, it stands to reason that having more information about a patient (i.e., longer context) should facilitate easier predictions."
>
> Yes, we agree with your reasoning here --  Please see our response to the comment directly above for our thoughts on this intuition. We would note, however, that while this generally makes sense, it may not hold for every architecture -- as shown in **Figure 1b,** we observe that certain architectures are better suited for processing longer context EHR data (e.g. see drop in Hyena performance). This underscores the importance of this type of experimental work in rigorously evaluating the performance claims of different architectures.
>
> ### "Could you elaborate on the paper's novelty?"
>
> Thank you for the opportunity to address this question. Please see our **Overall Response** under _“Novelty”_ for details on the novel contributions of this paper, which to reproduce below for your reference:
>
> First, we believe that our work does bring several novel contributions to the field:
>
> 1. We **introduce and quantify three unique characteristics of EHR sequential data** -- copy-forwarding, irregularity, and token complexity progression -- that have been underexplored in the NLP literature. We present novel empirical results demonstrating how these characteristics impact model performance across context lengths.
> 2. We are the **first to train and evaluate long context models for EHR data across multiple context lengths** and including both transformer and subquadratic architectures. We achieve SOTA results across 14 diverse clinical prediction tasks.
> 3. Our work is **distinct from the closest prior work, EHRMamba, in several key ways**. (a) We extend Mamba to multiple context lengths of 1k, 4k, 8k, and 16k tokens, in contrast to the single 2k context length tested in EHRMamba as detailed in **Table 1**. (b) We evaluate Mamba on 14 diverse clinical prediction tasks spanning operational outcomes, lab result prediction, and new diagnosis assignment. In contrast, EHRMamba assesses Mamba's performance on only 6 tasks. (c) Our analysis of three EHR-specific properties is entirely novel in **Table 2**. We stratify our results by each property and measure model robustness within these subpopulations. In contrast, EHRMamba does not assess how models perform across subgroups with varying levels of data complexity. (d) We conduct few-shot evaluations of our models (as low as $k = 8$) in **Appendix Figure 10**, in contrast to EHRMamba which relies on 10k’s of training examples for fine tuning.
> 4. We will (after the anonymization period) be the **first to publicly release the open weights of Llama, Mamba, and Hyena models (16 total) pretrained on real-world EHR data on HuggingFace** for open science and reproducibility. This is particularly impactful in a domain such as healthcare in which publishing the weights of models trained on real-world EHR data is rare.
>
> Second, we do not believe that introducing a novel architecture is necessary to make a positive contribution to the literature. The field of ML for healthcare is currently suffering from a **"reproducibility crisis"** but eager to deploy FM techniques. [2] Without rigorous understanding of their performance, however, new FM architectures will never be adapted in healthcare (e.g. see **Table 1** for the low adoption of non-transformer models within the EHR FM literature). Thus, our experiments can serve as a valuable bridge.
>
> Third, our work contributes to a larger trend in general LLM research towards **data-centric model development and evaluation**, going beyond architectures / training objectives and instead gaining insights via the construction and examination of domain-specific datasets and metrics with meaningful downstream evaluations. [3,4,5] In a fast-moving field like LLMs, it is difficult to distinguish noise from signal and identify which architectures work, for which domains, and under which configurations. We believe that our work offers useful insights for ML researchers hoping to apply sequence models to non-natural-language modalities (e.g. EHRs) where domain-specific properties can skew model performance.
>
> (cont...)

---

> ### Author Response · Authors · 2024-11-18
> **Author Response (cont.)**
>
> (...cont)
>
> [1] Fallahpour, Adibvafa, et al. "EHRMamba: Towards Generalizable and Scalable Foundation Models for Electronic Health Records." arXiv preprint arXiv:2405.14567 (2024).
> [2] McDermott, Matthew BA, et al. "Reproducibility in machine learning for health research: Still a ways to go." Science Translational Medicine 13.586 (2021): eabb1655.
> [3] Liu, Yang, et al. "Datasets for large language models: A comprehensive survey." arXiv preprint arXiv:2402.18041 (2024).
> [4] Guo, Zishan, et al. "Evaluating large language models: A comprehensive survey." arXiv preprint arXiv:2310.19736 (2023).
> [5] Zha, Daochen, et al. "Data-centric artificial intelligence: A survey." arXiv preprint arXiv:2303.10158 (2023).
>
> ### "Is the dataset utilized in the study publicly available?"
>
> Yes, the evaluation dataset used in our paper (EHRSHOT) is publicly available at this link: https://ehrshot.stanford.edu/
>
> In addition, we will make all of our pretrained models’ weights publicly available on HuggingFace after the anonymization period. We will be the first to publicly release the weights of Llama, Mamba, and Hyena models pretrained on EHR data. We believe that this represents another contribution of our work towards more reproducible and open science within the field of ML for healthcare.
>
> Our pretraining corpus of ~2.5M patients, however, is not publicly available given the difficulty of obtaining Privacy Office permission to release such a large dataset of real-world EHR data.

---

> ### Author Response · Authors · 2024-11-25
> **Update**
>
> Dear Reviewer uVBS,
>
> We hope all is well! May we ask your thoughts on our comments above? In our response, we:
>
> 1. **Outlined 4 ways** in which our work provides **novel contributions** to the ML literature, namely:
>
>     (1) Proposing and quantifying **three unique characteristics** of EHR sequential data -- copy-forwarding, irregularity, and token complexity progression -- that have been underexplored in the NLP literature (see **Figure 1** and **Table 2**)
>
>     (2) Being the **first to train and evaluate long context models for EHR data across multiple context lengths** across transformer and subquadratic architectures, achieving **SOTA results** on clinical prediction tasks. (see **Table 1**)
>
>     (3) Providing **few-shot** and **zero-shot** analyses (see newly added **Appendix Sections G and H**)
>
>     (4) Being the **first to publicly release the open weights** of Llama, Mamba, and Hyena models pretrained on real-world EHR data on HuggingFace. Given the **_"reproducibility crisis"_** (McDermott et al. 2021) currently faced by the field of ML for healthcare, this is an especially important contribution.
> 2. **Added Appendix Figure 6** of the distribution of patient sequence lengths in our evaluation dataset, thus providing evidence for **why long context matters** for EHR data.
> 3. **Clarified** that our evaluation dataset is **publicly available.**
>
> We believe that we have resolved your concerns, so please let us know if you have any other questions or concerns. We hope you are able to take into account our response, including additional experiments and clarifications provided, into the final evaluation of our work. If there is no update, we certainly respect your decision, and thank you for the time!
>
> Best,
> The Authors

---

### Official Review · Reviewer_TDwP · 2024-11-03

**Soundness:** 4
**Presentation:** 3
**Contribution:** 3
**Rating:** 8
**Confidence:** 4

**Summary:**

This paper investigates the challenges of using language models to model EHR data. By comparing EHR data with natural language, this paper identifies three unique and significant properties of EHR data (copy-forwarding, irregular time intervals, and disease progression) that make modeling EHR sequences more complex than natural language. To provide evidence for these properties, this paper evaluated the performance and robustness of three language models by varying the repetitiveness, irregularity, and context length of EHR data. The models are pre-trained using a private dataset and evaluated with a publicly available dataset, EHRSHOT.

**Strengths:**

The identified properties of EHR data are convincing. The evaluation of the effects of these properties provides valuable insights into using long-context models to model EHR data. The observations and conclusions in this paper will be helpful for future work to build better foundation models for EHR.

The authors have released the code and plan to release the model checkpoints later. The release of pre-trained and fine-tuned foundational models will benefit the community, considering the small number of such models currently publicly available.

**Weaknesses:**

Although the authors test the performance of different language models, the tokenization strategies of these models remain the same as the one used in EHRSHOT. It would be helpful to see if using other tokenization strategies could improve performance. For example, Section 4.3 indicates that irregular inter-token time intervals are harder to model. This conclusion is based on EHRSHOT’s tokenization, which doesn’t encode time intervals. However, there are other tokenization strategies, such as those used by ExBEHRT and EHRMamba, that do encode time intervals.

This paper uses only one EHR dataset to evaluate language models, which somewhat limits its conclusions to the EHRSHOT dataset. While I understand that unifying different EHR datasets is highly intensive work, it would be valuable to see whether similar observations consistently appear in other EHR datasets, such as MIMIC.

**Questions:**

Some technical details seem to be missing in the paper. For example, it is unclear what the 14 tasks in EHRSHOT are, whether some tasks have highly imbalanced label distributions, and if the sample sizes for these 14 tasks are the same. Since Table 2 reports the mean performance across all 14 EHRSHOT tasks, the lack of such information makes it challenging to assess the actual performance.

When finetuning models, the author wrote "To be consistent with the original EHRSHOT benchmark, we do not finetune our base models – instead, we train a logistic regression head on top of frozen representations generated for each patient by our base models." However, there is no evaluation of CLMBR-T-Base, which is the foundation model released together with EHRSHOT.  Why was CLMBR-T-Base not included in the experiment?

---

> ### Author Response · Authors · 2024-11-18
> **Author Response**
>
> We are grateful for your detailed and thoughtful review of our work. We’re glad that our work offered “valuable insights” for the development of future EHR foundation models, and that you found our experiments “convincing” regarding the importance of considering EHR-specific properties when modeling EHR data, especially with newer long-context models. We also appreciate your acknowledgment of the importance of our forthcoming code and model releases to “benefit the community”, especially “considering the small number of models currently publicly available”. By sharing these resources, we aim to help alleviate the shortage of open pre-trained models in the healthcare sector. We appreciate your constructive feedback as well in improving our manuscript; in addition to the Overall Response, we have addressed each of your comments in sequence below:
>
> ### "Although the authors test the performance of different language models, the tokenization strategies of these models remain the same as the one used in EHRSHOT. It would be helpful to see if using other tokenization strategies could improve performance. For example, Section 4.3 indicates that irregular inter-token time intervals are harder to model. This conclusion is based on EHRSHOT’s tokenization, which doesn’t encode time intervals. However, there are other tokenization strategies, such as those used by ExBEHRT and EHRMamba, that do encode time intervals."
>
> Thank you for the comment. Please see the **Overall Response** section _“Tokenization strategies for dealing with time”_ in which we provide results details on a time-aware tokenization strategy sourced from the literature that was specifically designed for dealing with time. As seen in **Appendix Figure 12**, adding time to our tokenization strategy slightly reduced performance across all models.
>
> ### "This paper uses only one EHR dataset to evaluate language models, which somewhat limits its conclusions to the EHRSHOT dataset. While I understand that unifying different EHR datasets is highly intensive work, it would be valuable to see whether similar observations consistently appear in other EHR datasets, such as MIMIC."
>
> Thank you for the great question. We agree that additional evaluation on other datasets is always desirable -- That is precisely why we are publicly releasing the full weights of all of our trained models on HuggingFace, to enable other researchers to build on our results and evaluate our models on other EHR datasets.
>
> In the interim, given that EHR-OMOP and EHRSHOT are both longitudinal datasets (i.e. they include both general patient visits and ICU visits, which is the specific type of data contained in MIMIC), we believe that the 14 diverse tasks evaluated in our work provide comprehensive coverage across a broad range of clinical scenarios and thus provides a strong basis for the generalizability of our findings. However, we agree that further evaluations are always important, and hope that our open sourced training code + model weights enable future work in investigating the generalization of our findings to arbitrary EHR datasets.
>
>
> ### "Some technical details seem to be missing in the paper. For example, it is unclear what the 14 tasks in EHRSHOT are, whether some tasks have highly imbalanced label distributions, and if the sample sizes for these 14 tasks are the same. Since Table 2 reports the mean performance across all 14 EHRSHOT tasks, the lack of such information makes it challenging to assess the actual performance."
>
> We agree that we did not provide sufficient information on the 14 EHRSHOT tasks in our initial draft, and believe the manuscript has been significantly improved in addressing this feedback.
>
> As mentioned in our **Overall Response** under _“Additional clarity about the 14 EHRSHOT evaluation tasks”,_ to address your comment we have **added 4 pages of additional detail** covering the 14 EHRSHOT tasks to Appendix Section B.  Improvements that we made to the manuscript in response to your suggestion include:
> 1. We now provide a list of 14 EHRSHOT tasks and their label type, prediction time, and time horizon (**Appendix Table 4**)
> 2. We added precise definitions of all 14 tasks, including the EHR codes corresponding to the labeling functions used for identifying each lab test and new diagnosis (**Appendix Section B.1**)
> 3. We now provide the sample sizes (# of patients and # of labels) and class balance (# of positive/negative labels) for each split (train/test/val) for all 14 tasks (**Appendix Table 5**)
> 4. We added plots showing the number of raw clinical events and tokens for each label each split (train/test/val) for all 14 tasks (**Appendix Figure 8**)
>
> (cont...)

---

> ### Author Response · Authors · 2024-11-18
> **Author Response (cont.**
>
> (...cont)
>
> We have also added the following text to our main manuscript in **Section 3.2 (“Evaluation”)** to provide a high-level description of these tasks and point the reader to our new Appendix section:
>
> > We use the remaining 14 tasks from the EHRSHOT benchmark for our evaluations,which are broadly grouped into three categories: Operational Outcomes includes predicting ICU Transfer, 30-day Readmission, and Long Length-of-Stay; Anticipating Lab Test Results involves predicting if a thrombocytopenia, hyperkalemia, hypoglycemia, hyponatremia, or anemia lab will be abnormal; and Assignment of New Diagnoses requires predicting whether a patient will get a new diagnosis of hypertension, hyperlipidemia, pancreatic cancer, celiac disease, or lupus within the next year. For additional details on all 14 tasks, including precise definitions, label counts, statistics on the number of tokens per patient, and evaluation methodology, please see Appendix Section B.
>
> ### "When finetuning models, the author wrote "To be consistent with the original EHRSHOT benchmark, we do not finetune our base models – instead, we train a logistic regression head on top of frozen representations generated for each patient by our base models." However, there is no evaluation of CLMBR-T-Base, which is the foundation model released together with EHRSHOT. Why was CLMBR-T-Base not included in the experiment?"
>
> We do actually include CLMBR-t-base in all of our evaluations, as detailed in our **Overall Response** under _“How we evaluated the Prior SOTA model (CLMBR-t-base).”_ For example, in **Figure 1b** CLMBR-t-base’s performance is shown as the purple dashed line marked “Prior SOTA”. We apologize for the confusion here!
>
> We agree that the comparison between our models and CLMBR-t-base should have been clearer. Thus, we have added several pages of results in the form of **Appendix Tables 7, 8, 9, and 10** which directly compare the performance of all of our models against CLMBR-t-base across all tasks.

---

> > ### Author Response · Authors · 2024-11-25
> > **Update**
> >
> > Dear Reviewer TDwP,
> >
> > We hope all is well! May we ask your thoughts on our comments above? To summarize our response, we:
> >
> > 1. **Added new experiments** using an **improved tokenization strategy** for modeling time (inspired by modern EHR FMs such as CEHR-BERT, CEHR-GPT, and ETHOS) to **Appendix Section D** and **Appendix Figure 12.**
> > 2. **Added ~4 new pages of details on EHRSHOT** to answer your questions regarding the 14 tasks and label prevalences. Please see the additional paragraph in **Section 3.2** of the main manuscript, as well as the newly added **Appendix Section B** and **Appendix Table 4 and 5** and **Appendix Figure 8.**
> > 3. **Clarified that CLMBR-t-base results are already included** in all of our experiments. To provide additional detail, we added **Appendix Tables 7, 8, 9, and 10** which directly compare the performance of all of our models against CLMBR-t-base across all tasks.
> >
> > We believe that we have resolved your concerns, but please let us know if you have other questions. We hope you are able to take into account our response, including additional experiments and clarifications provided, into the final evaluation of our work. If there is no update, we certainly respect your decision, and thank you for the time!
> >
> > Best,
> > The Authors

---

> > > ### Comment · Reviewer_TDwP · 2024-12-02
> > >
> > > Thanks for the detailed replies. The new materials answered my questions Good work!

---

### Official Review · Reviewer_UYtp · 2024-11-03

**Soundness:** 3
**Presentation:** 3
**Contribution:** 3
**Rating:** 6
**Confidence:** 4

**Summary:**

The manuscript benchmarks four foundation model architectures on Electronic Health Records (EHRs) to investigate the impact of context length on the model performance of downstream tasks. Moreover, the authors identified and quantified three challenges present in EHR data and showed that long-context models are better at mitigating the challenges.

**Strengths:**

1. The authors benchmark both transformer-based and subquadratic architectures on EHR data
2. The authors identified and quantified three challenges present in EHR data.
3. The authors conduct experiments to show the effectiveness of long-context models on EHR data.

**Weaknesses:**

1. The design of token repetition measurement is less convincing.
	- Are the proposed models applicable to ICU patients? If yes, routine vital signs and regular lab tests can repeat a lot, but they can continuously show patients' health status. It is tricky to determine whether they are informative.


2. the comprehensiveness of experimental design is limited
	- The investigated methods are limited. They are general architectures for foundation models. However, foundation models designed for EHR, such as [1] and [2], are not included.
	- The authors claimed that irregular time gaps hinder the performance of the models. This is reasonable because the time gap is not encoded during tokenization. It could be interesting to see whether encoding time information would be helpful for some stratified groups although the gain may be minimal overall.

3. The experiment result reported is limited for a comprehensive understanding
	- Prior SOTA mentioned in the manuscript (CLMBR-t-base) is also a transformer-based model. However, the context length of this model is not discussed. Additionally, it is not trained with variable context lengths.
	- Table 2 provides stratified results of the same experiment as Figure 1 (b) and (d). However, it is confusing that CLMBR-t-base and Hyena don't appear in this table.
	- The author hypothesizes that some degree of copy-forwarding can be helpful by emphasizing important diagnoses. This is observed from the CLMBR-t-base but cannot be validated by other models. Moreover, the Brier score of the CLMBR-t-base seems smaller than other models.
	- Standard deviation is not provided when comparing different models. The authors only conduct statistical tests between short- and long- context counterparts. However, neither standard deviation nor statistical testing is reported when comparing different methods.
	- (Minor) The impact of the long-context and proposed properties on pretraining is not discussed. The downstream tasks are enough to show the conclusion but it will be better to see if these factors affect training process.

4. There are some typos in the manuscript
	- In line 139, there's a corrupted citation
	- 3.3.3 title: Diseae -> Disease
	- Reference [Yang et al., 2023a] and [Yang et al., 2023b] are the same
References:
[1] Guo, L. L., Steinberg, E., Fleming, S. L., Posada, J., Lemmon, J., Pfohl, S. R., ... & Sung, L. (2023). EHR foundation models improve robustness in the presence of temporal distribution shift. Scientific Reports, 13(1), 3767.
[2] Fallahpour, A., Alinoori, M., Afkanpour, A., & Krishnan, A. (2024). EHRMamba: Towards Generalizable and Scalable Foundation Models for Electronic Health Records. arXiv preprint arXiv:2405.14567.

**Questions:**

- Statistics of code, event, and token are confusing.


- Diagnosis codes for chronic diseases can frequently appear in a patient’s EHR, often for billing purposes rather than indicating an active health issue. For instance, if a patient with a chronic condition (e.g., COPD or obesity) visits the hospital for an unrelated condition, the chronic disease code may not appear in that visit. In contrast, for acute diseases, the presence of a code in the record typically indicates an active case during that visit. How can token repetition be modeled effectively for these two types of diseases?


- In line 261, the authors reported that the vocabulary has 39818 tokens. Is this the vocabulary of EHRSHOT, or does EHR-OMOP also share it?

- Meanwhile, in line 950 of Appendix C, it is reported that 39811 codes are selected. Is "code" here equivalent to "token"?

- In Table 3 in Appendix A, The number of unique codes in EHR-OMOP is much more than the token vocabulary mentioned earlier. Is this the original vocabulary without removing infrequent codes?

---

> ### Author Response · Authors · 2024-11-18
> **Author Response**
>
> Thank you for the very detailed and thoughtful review of our work. We appreciate your acknowledgment of our efforts to be among the first to benchmark both transformer-based and subquadratic architectures on EHR data, which we believe highlights the value of exploring diverse model architectures for healthcare applications. We are also grateful for your recognition of the importance of the 3 key challenges within EHR data that we identified in our study, and for the soundness of our experimental setup. In addition to the Overall Response above, we will address each of your comments in order below.
>
> ### "The design of token repetition measurement is less convincing. Are the proposed models applicable to ICU patients? If yes, routine vital signs and regular lab tests can repeat a lot, but they can continuously show patients' health status. It is tricky to determine whether they are informative."
>
> Thank you for the question, this is an excellent point. Our dataset contains longitudinal health records, and thus is primarily composed of inpatient visits (which do include ICU stays). However, other EHR datasets such as MIMIC-III, eICU, and HiRCD only contain ICU visits and thus may exhibit this property to a more extreme degree -- your point that a more refined metric which distinguishes between clinically meaningful repetition and non-informative patterns is well taken. As a result, we have added the following to the Future Work subsection of **Section 5 (“Discussion”).**
>
> > Distinguishing between meaningful and non-meaningful repetition (e.g., in ICU stays) could improve model performance in high-repetition settings.
>
> ### "the comprehensiveness of experimental design is limited: The investigated methods are limited. They are general architectures for foundation models. However, foundation models designed for EHR, such as [1] and [2], are not included. References: [1] Guo, L. L., Steinberg, E., Fleming, S. L., Posada, J., Lemmon, J., Pfohl, S. R., ... & Sung, L. (2023). EHR foundation models improve robustness in the presence of temporal distribution shift. Scientific Reports, 13(1), 3767. [2] Fallahpour, A., Alinoori, M., Afkanpour, A., & Krishnan, A. (2024). EHRMamba: Towards Generalizable and Scalable Foundation Models for Electronic Health Records. arXiv preprint arXiv:2405.14567."
>
> Thank you for this feedback. We address this point in our **Overall Response** under the header _“How we evaluated the Prior SOTA model (CLMBR-t-base).”_
>
> Briefly, we do already include the model cited in [1] (CLMBR) in our benchmark -- this is referred to as the “Prior SOTA” model in **Figure 1b.** Per your suggestion, we have also **added 4 pages of results** in the form of **Appendix Tables 7, 8, 9, and 10** which directly compare the performance of all of our models against CLMBR across all tasks.
>
> The EHRMamba model cited in [2] is architecturally identical to the Mamba model we train. However, their paper does evaluate a few different tokenization strategies for EHR data. In our updated manuscript, we have added additional investigations into tokenization strategies as detailed in our **Overall Response** under the header _“Tokenization strategies for dealing with time”._
>
> ### "The authors claimed that irregular time gaps hinder the performance of the models. This is reasonable because the time gap is not encoded during tokenization. It could be interesting to see whether encoding time information would be helpful for some stratified groups although the gain may be minimal overall."
>
> Thank you for your suggestion. Please see our **Overall Response** under _“Tokenization strategies for dealing with time.”_ We agree that incorporating temporal information directly into the tokenization process could provide a more nuanced understanding of patient data, particularly in handling irregular time gaps.
>
> In response, we retrained all of our models using a state-of-the-art tokenization strategy known as Artificial Time Tokens (ATTs), inspired by recently published EHR FMs [1, 2, 3]. We provide results in **Appendix Figure 12** and describe this tokenization strategy in more detail in **Appendix Section D (“Tokenization”).** Surprisingly, we empirically find that such tokenizations do not improve performance.
>
> [1] Pang, Chao, et al. "CEHR-BERT: Incorporating temporal information from structured EHR data to improve prediction tasks." Machine Learning for Health. PMLR, 2021.
> [2] Renc, Pawel, et al. "Zero shot health trajectory prediction using transformer." NPJ Digital Medicine 7.1 (2024): 256.
> [3] Pang, Chao, et al. "CEHR-GPT: Generating electronic health records with chronological patient timelines." arXiv preprint arXiv:2402.04400 (2024).
>
> (cont...)

---

> ### Author Response · Authors · 2024-11-18
> **Author Response (cont.)**
>
> (...cont)
>
> ### "The experiment result reported is limited for a comprehensive understanding. Prior SOTA mentioned in the manuscript (CLMBR-t-base) is also a transformer-based model. However, the context length of this model is not discussed. Additionally, it is not trained with variable context lengths."
>
> Thank you for the question. Please see our **Overall Response** under _“How we evaluated the Prior SOTA model (CLMBR-t-base)”._
>
> In brief, we have updated **Table 1** in response to your feedback to accurately reflect the context length of this model, which was 512 tokens. CLMBR-t-base was not trained with variable context lengths -- as noted in **Section 2.3 (“Related Work”),** this form of fixed context length evaluation was a key limitation of prior work that we address via our work. We have also edited the following text in **Section 4.1** to make this clearer in the main manuscript:
>
> > The dotted purple line is the mean AUROC (0.777) achieved by the best overall prior model, CLMBR-t-base \citep{wornow2023ehrshot}, which had a context length of 512 tokens.
>
> ### "Table 2 provides stratified results of the same experiment as Figure 1 (b) and (d). However, it is confusing that CLMBR-t-base and Hyena don't appear in this table."
>
> Thank you for the feedback. We have added **Appendix Table 14** to provide full results across all models, including Hyena and CLMBR-t-base. We originally only had Mamba, Llama, and GPT in order to save space in the main manuscript given the page limit.
>
> ### "The author hypothesizes that some degree of copy-forwarding can be helpful by emphasizing important diagnoses. This is observed from the CLMBR-t-base but cannot be validated by other models. Moreover, the Brier score of the CLMBR-t-base seems smaller than other models."
>
> We appreciate your bringing this to our attention. There appeared to be a small error in our data processing pipeline, which we have since resolved and have updated **Figure 1b** and **Table 2** accordingly. Overall, the empirical trend is that copy-forwarding worsens model performance across the board, and we have updated our analysis accordingly. This update also resolves your second question about the Brier scores of CLMBR-t-base being smaller for certain quartiles and bucketing strategies, as this no longer appears to be the case for most quartiles.
>
> ### "Standard deviation is not provided when comparing different models. The authors only conduct statistical tests between short- and long- context counterparts. However, neither standard deviation nor statistical testing is reported when comparing different methods."
>
> To address your comment, we conducted additional analyses comparing each model architecture directly to the prior state-of-the-art, CLMBR-t-base, rather than performing inter-model comparisons. In **Appendix Tables 7 through 10**, we present AUROC scores for each model at various context lengths and tasks, focusing on performance changes relative to CLMBR-t-base. To ensure statistical rigor, we calculated 95% confidence intervals for each observed delta with 1,000 iterations of bootstrapping on the test set, which provides a measure of statistical significance for each performance change.
>
> ### "(Minor) The impact of the long-context and proposed properties on pretraining is not discussed. The downstream tasks are enough to show the conclusion but it will be better to see if these factors affect training process."
>
> Thank you for this valuable suggestion. We agree that exploring the impact of long-context settings and EHR-specific properties on the pretraining phase could provide additional insights. While our study primarily focuses on downstream tasks to validate model effectiveness, investigating these factors during pretraining may reveal nuances in training dynamics that could affect overall model performance. We have noted this in the Future Work subsection of **Section 5 (“Discussion”).**
>
> > Exploring these EHR-specific properties in the pretraining phase could also reveal training nuances such as convergence speed and stability, influencing model performance.
>
> ### "There are some typos in the manuscript. In line 139, there's a corrupted citation"
>
> Thank you for bringing this to our attention -- it has been resolved!
>
> ### "3.3.3 title: Diseae -> Disease"
>
> Resolved!
>
> ### "Reference [Yang et al., 2023a] and [Yang et al., 2023b] are the same"
>
> Resolved!
>
> (cont...)

---

> ### Author Response · Authors · 2024-11-18
> **Author Response (cont.)**
>
> (...cont)
>
> ### "Statistics of code, event, and token are confusing."
>
> Thank you for your question, and we apologize for the confusion. We have made this distinction clearer with the following changes:
>
> First, to clarify the distinction between an “event”,  “code” and “token”, we’ve added a description in the Tokenization subsection of **Section 3 (“Methods”).**
>
> > Each clinical “event” in a patient’s timeline has a single “code” associated with it. Each “code” then gets converted into a single “token” within our vocabulary via the following process. First, all unique codes $c \in \mathcal{C}$ that occur at least once in our training dataset are assigned a unique token. Second, all codes that are associated with categorical values are assigned a unique token for each possible associated categorical value. Third, all codes associated with numerical values are assigned a unique token for each decile within the range of values attained in our training dataset. After sorting all tokens by their information content, the top 39811 tokens were kept as our vocabulary, and all models share this same vocabulary. Please see Appendix Section \ref{sec:appendix_tokenization} for additional details on the token generation and selection process.
>
> Second, there are several figures that mention either tokens or events. We have added additional context to their captions to make this distinction clearer, and added an explanation around why we measure one or the other within each figure. Overall, we use events to quantify irregularity and repetitiveness because events capture the broad structure of patient visits and interactions over time, which is essential for understanding time intervals and repeated diagnoses. In contrast, we use tokens to quantify disease progression because tokens capture finer-grained details such as the severity or specific characteristics of clinical findings, allowing us to assess how model perplexity responds to increasing complexity in patient health data over time.
>
> > **Figure 2** caption -  “EHR data exhibits a high degree of variation in time intervals between events. From left to right, we measure the mean, standard deviation, and inter-quartile range (IQR) of time intervals between events, reflecting the irregular timing of clinical interactions. EHR-OMOP is the 0.5M patients in the EHR-OMOP validation set. The x-axis (log scale) represents the metric in seconds, ranging from $10^1$ to $10^9$. The y-axis measures the number of sequences with those values. Here, we focus on event intervals to capture the temporal structure of clinical encounters and highlight patterns in patient healthcare utilization.”
>
> > **Figure 3** caption - - “EHR data exhibits a higher degree of repetition than natural language, as measured by $n$-gram repetition rates. From left to right, we measure $n =1, 2, 3, 4$. EHR-OMOP (blue) is the 0.5M patients in the EHR-OMOP validation dataset, ``WikiText" (orange) is the WikiText-103 training dataset of high quality Wikipedia articles \citep{merity2016pointer}.We analyze $n$-gram repetition at the event level to reflect the structure of recurring clinical entries, capturing patterns unique to EHR data.The x-axis represents the $n$-gram repetition rate (i.e., percentage of $n$-grams that are repeated at least once within a sequence, where higher is more repetitive) and the y-axis shows the frequency of sequences with that repetition rate in each dataset.”
>
> > **Appendix Figure 5** caption - Distributions of patient data from the EHR-OMOP dataset across (A) training and (B) validation splits, showing both event-level and code-level counts. The x-axis is log-scaled to capture the wide range in the number of events per patient, the number of unique patients per code, and the distribution of events associated with each code.
>
> > **Appendix Figure 6** caption - For each EHRSHOT task, we plot the CDF of the number of \textbf{raw clinical events} (left column) and \textbf{tokens} (right column) available to the model when making its prediction. In other words, the number of events/tokens preceding each label's prediction time point. The blue line represents all prediction times, while the orange line represents only predictions associated with a positive label. Note that unlike the raw event counts, all token counts are capped at the maximum context length of the models we test (16k), hence the spike at the end of the CDF.
>
> (cont...)

---

> ### Author Response · Authors · 2024-11-18
> **Author Response (cont.)**
>
> (...cont)
>
> ### "Diagnosis codes for chronic diseases can frequently appear in a patient’s EHR, often for billing purposes rather than indicating an active health issue. For instance, if a patient with a chronic condition (e.g., COPD or obesity) visits the hospital for an unrelated condition, the chronic disease code may not appear in that visit. In contrast, for acute diseases, the presence of a code in the record typically indicates an active case during that visit. How can token repetition be modeled effectively for these two types of diseases?"
>
> Thank you for the great question! It relates closely to your earlier observation that repetition patterns in ICU settings are clinically informative as opposed to other areas in the EHR where repetition primarily serves for billing purposes. Our current study does not explicitly differentiate between these types of repetition, but we agree that this would make for exciting future work that builds on the EHR-specific properties we introduce. We have added the following to the Future Work subsection of **Section 5 (“Discussion”).**
>
> > Distinguishing between meaningful and non-meaningful repetition (e.g., in ICU stays) could improve model performance in high-repetition settings.
>
> To brainstorm for a bit, one suggestion might be to use an ontology to upweight codes corresponding to acute diseases, or to artificially copy-forward chronic diseases with high information content. Empirically, we find in our host institution’s EHR dataset (EHR-OMOP) that codes across all disease types are frequently copied forward for billing and quality rating purposes. We attempt to take this phenomenon into account via our tokenization strategy of including the top 39811 tokens based on their information content, which is calculated following the process detailed in [1].
>
> [1] Steinberg, Ethan, et al. "Language models are an effective representation learning technique for electronic health record data." Journal of biomedical informatics 113 (2021): 103637.
>
> ### "In line 261, the authors reported that the vocabulary has 39818 tokens. Is this the vocabulary of EHRSHOT, or does EHR-OMOP also share it? Meanwhile, in line 950 of Appendix C, it is reported that 39811 codes are selected. Is "code" here equivalent to "token"?"
>
> Thank you for the clarification question. This vocabulary is used by all models across all datasets -- e.g. both EHRSHOT and EHR-OMOP. There are 39811 tokens in this vocabulary which are derived from codes (e.g. “LOINC/718-7”) and 7 special tokens: [ '[BOS]', '[EOS]', '[UNK]', '[SEP]', '[PAD]', '[CLS]', '[MASK]']. So, the total vocabulary size is 39818. To clarify this, we’ve added the following explanation in **Appendix Section D (“Tokenization”).**
>
> > In addition, seven special tokens—[BOS], [EOS], [UNK], [SEP], [PAD], [CLS], and [MASK]—are included, resulting in a total vocabulary size of 39818 tokens.
>
> ### "In Table 3 in Appendix A, The number of unique codes in EHR-OMOP is much more than the token vocabulary mentioned earlier. Is this the original vocabulary without removing infrequent codes?"
>
> Yes, you are correct. The number of unique codes mentioned in **Appendix Table 3** constitutes the original vocabulary before we conduct our token selection process, as detailed in **Appendix Section D (“Tokenization”).** This procedure (essentially sorting codes in order of decreasing number of occurrences) takes us from a set of 3,144,978 possible codes down to 39811.

---

> ### Author Response · Authors · 2024-11-25
> **Update**
>
> Dear Reviewer UYtp,
>
> We hope all is well! May we ask your thoughts on our comments above? To summarize our response, we:
>
> 1. **Added new experiments** using an **improved tokenization strategy** for modeling time (inspired by modern EHR FMs such as CEHR-BERT, CEHR-GPT, and ETHOS) to **Appendix Section D** and **Appendix Figure 12**.
> 2. **Clarified** how our results **compare to CLMBR-t-base**, a SOTA EHR FM
> 3. **Added Hyena/CLMBR-t-base stratification results** to our Brier score evaluations in **Appendix Table 14**.
> 4. **Resolved questions**  about Brier scores with CLMBR-t-base
> 5. **Added** several of your suggested **ways to improve the 3 metrics we propose** to the Future Work part of **Section 5**.
> 6. **Fixed all typos**
> 7. **Added clarifications** about our **vocabulary / tokenization** strategy used per your comments.
>
> We believe that we have resolved your concerns, so please let us know if you have any other questions. We hope you are able to take into account our response, including additional experiments and clarifications provided, into the final evaluation of our work. If there is no update, we certainly respect your decision, and thank you for the time!
>
> Best,
> The Authors

---

### Official Review · Reviewer_wRBt · 2024-11-05

**Soundness:** 4
**Presentation:** 4
**Contribution:** 4
**Rating:** 10
**Confidence:** 4

**Summary:**

This paper analyzes qualities unique to EHR data that are challenging when scaling up sequence lengths. They identify the key properties of copy-forwarding (EHR data is repetitive), irregular time intervals (time intervals vary greatly between tokens), disease complexity (EHR data actually has generally increasing perplexity overtime--unlike text data--as far out future trajectories are hard to predict). They evaluate their models in zero-shot, few-shot, and linear probing regimes, varying context lengths and architectures, and even sharing compute time estimates via token throughputs.

The authors share metrics for quantifying the severity of these three properties on EHR datasets, and demonstrate that severity of these properties correlates with worse performance. They additionally show that longer sequence lengths can help.

Every weakness I pointed out in the initial review has been thoroughly addressed by the authors, this paper is both a clear accept and a major contribution in the EHR machine learning space as it provides a broad benchmark and insights on training autoregressive models in this space.

**Strengths:**

1. The authors define metrics for evaluating the severity of these properties on any dataset.

2. They demonstrate that as the RR metric and irregularity metric increase, model performance decreases (via patient stratification experiments).

3. The demonstration that perplexity does not generally decrease with sequence length is a major deviation from text data where it is well known to decrease with sequence length. This is because future EHR data is less predictable from previous tokens. The assumption that perplexity reduces with context length for EHR

4. The authors demonstrate that for all three properties, increasing sequence length helped improve performance (either via improved brier scores or perplexity).

**Weaknesses:**

The authors provide in Figure 4 plots of perplexity over tokens for different context-length models. The GPT model has wildly varying perplexity over token positions which is described by the authors as being caused by "training instability". I think a more thoughtful analysis of the issue here is required, because it would otherwise look like the cause is a bug.

Why isn't transfer-learning or few-shot included. A major problem in the evaluation is that representations are obtained by averaging the last L token representations from the transformer (This is a one-liner in the appendix and really should be added to the main paper and be clearly communicated in the limitations section). It would be great to see these results in the few-shot setting. I imagine that the performance improvements as you increase sequence length would be even more extreme.

This paper should include comparisons to linear attention models that practitioners are interested in.

This paper does not communicate compute budgets, such as wall-times and the hardware used for these sequence lengths. Could a plot communicating performance vs the compute-load be provided to help justify whether these improvements are worth the added compute time.

Since you trained an autoregressive sequence model, you could perform a zero-shot evaluation where given the past tokens, you autoregressively generate the future tokens and analyze this generated future trajectory for the binary classification task. This paper does not demonstrate whether these results generalize to the zero-shot. I think that analysis is out of scope for this work but should be mentioned in the limitations.

**Questions:**

I included my questions in the Weakness section, but I'll summarize the actionables below:

1. **The most critical improvement to this paper**: Include a few-shot/transfer-learning results, the use of mean pooling the last L tokens for all evaluations is not a well supported embedding strategy. Alternatively, you have a zero-shot capable model, why not evaluate it in the zeroshot setting as past works do [1,2,3]? At the moment the limited embedding strategy greatly diminishes the generalizability of your results. If this were resolved, I would significantly increase my rating for the paper.
2. Add compute time results, and analysis of the diminishing or increasing predictive performance returns as you increase compute times (via increasing sequence lengths) across methods.
3. A more complete Analysis of the erratic GPT-model Perplexity behavior
4. Add a comparison with linear attention models


[1] Renc, Pawel, et al. "Zero shot health trajectory prediction using transformer." NPJ Digital Medicine 7.1 (2024): 256.

[2] Kraljevic, Zeljko, et al. "Foresight—a generative pretrained transformer for modelling of patient timelines using electronic health records: a retrospective modelling study." The Lancet Digital Health 6.4 (2024): e281-e290.

[3] McDermott, Matthew, et al. "Event Stream GPT: a data pre-processing and modeling library for generative, pre-trained transformers over continuous-time sequences of complex events." Advances in Neural Information Processing Systems 36 (2023): 24322-24334.

---

> ### Author Response · Authors · 2024-11-18
> **Author Response**
>
> Thank you so much for your very thorough and constructive review of our manuscript. We appreciate your recognition of the value in defining and quantifying the 3 metrics we introduce -- irregularity, copy-forwarding, and progression -- for evaluating models on sequential datasets, especially given the “major deviation” that our work reveals versus prior literature in NLP. In addition, we appreciate your clarity in suggested improvements -- in particular, you mentioned that adding few-shot/zero-shot results and improving our embedding strategy would significantly increase your rating for the paper. We address each of your comments below, but please let us know if there are further issues we could address to improve the manuscript :
>
> ### "The authors provide in Figure 4 plots of perplexity over tokens for different context-length models. The GPT model has wildly varying perplexity over token positions which is described by the authors as being caused by "training instability". I think a more thoughtful analysis of the issue here is required, because it would otherwise look like the cause is a bug."
>
> Thank you for the opportunity to clarify this. As we initially suspected, the issue turned out to be the absolute positional embeddings used in the original GPT-2 architecture. Per your comment, we retrained all of our GPT models using rotary positional embeddings (ROPE) [1] and found that the strange perplexity spikes were largely resolved, while the overall trend of increased perplexity with increased context length remained. Please see **Appendix Figure 11** for this updated plot.
>
> Given that the original GPT architecture used absolute positional embeddings, we keep our original results in our main manuscript the same in order to stay true to the architecture.
>
> We believe this is a useful finding to highlight for the EHR FM community, as per **Table 1** virtually all decoder-only EHR FMs are based on the original GPT rather than Llama architecture. However, as our new results demonstrate, the use of ROPE and other improvements offered by Llama (e.g. RMSNorm, SwiGLU, etc.) offer clear improvements over GPT for EHR data, especially at longer contexts.
>
> We have updated the following portion of **Section 4 (“Results”)** to reflect these new findings thanks to your comment:
>
> > For GPT, results are mixed: longer contexts (2k and 4k) achieve lower perplexities at later tokens but exhibit significant spikes. This appears to be caused by GPT's usage of absolute positional embeddings -- replacing them with rotary positional embeddings (ROPE) \cite{su2024roformer} largely resolved these spikes as seen in Appendix Figure \ref{fig:appendix_gpt_rope}. Thus, despite its popularity in the EHR FM community (see Table \ref{table:prior_work}), we recommend discontinuing the GPT architecture in favor of Llama or other more modern decoder-only architectures.
>
> [1] Su, Jianlin, et al. "RoFormer: enhanced transformer with rotary position embedding. CoRR abs/2104.09864 (2021)." arXiv preprint arXiv:2104.09864 (2021).
>
>
> ### "A major problem in the evaluation is that representations are obtained by averaging the last L token representations from the transformer (This is a one-liner in the appendix and really should be added to the main paper and be clearly communicated in the limitations section). "
>
> You are correct -- this is NOT what we did. Instead, **we used the embedding of the last token in the sequence** as our representation for each patient in all of our evaluations (i.e. we did **NOT** mean pool over multiple tokens). Please see our code for references to model checkpoints ending in `-persist_chunk:last_embed:last` in all of our analyses as evidence of this.
>
> We had tried both strategies in preliminary experiments (i.e. mean pooling the last L tokens v. just using the last token’s embedding), but seem to have miscopied which version we used in our manuscript. We greatly appreciate the close read and for raising this to our attention -- we have fixed the manuscript accordingly in **Appendix Section B.2 (“Evaluation Procedure”):**
>
> > Initial experiments indicated that setting $A$ to simply return the last vector in the sequence (i.e. the most recent token in a patient's timeline) and $S$ to the most recent $L$ tokens in a patient's timeline prior to the timepoint at which the prediction for a task is made performed the best.
>
> We have also added the following to **Section 3.2 (“Evaluation”)** of our main manuscript for clarity:
>
> > For our evaluations, we use the same context length that was used during pretraining. We thus sample the last $\min\{ L, |T_i|\}$ tokens for each patient prior to the relevant prediction time for a task, then take the embedding of the last token in that sequence as our representation for that patient.
>
> We hope that this resolves your concern regarding the generalizability of our embedding strategy.
>
> (cont...)

---

> ### Author Response · Authors · 2024-11-18
> **Author Response (cont.)**
>
> (...cont)
>
> ### "Why isn't transfer-learning or few-shot included...It would be great to see these results in the few-shot setting. I imagine that the performance improvements as you increase sequence length would be even more extreme."
>
> Thank you for raising this point, we agree that the paper would be significantly improved by including few-shot results. In response, we have added **Appendix Section G (“Few-shot learning on EHRSHOT”)** which includes few-shot results for all models and context lengths on all EHRSHOT tasks in **Appendix Tables 11, 12, and 13** as well as **Appendix Figure 10.**
>
> The results confirm that for models which benefit from longer contexts (e.g. Mamba and Llama), we see consistent performance gains of longer context across all values of $k$ (i.e. # of shots). Please see the updated manuscript for full details on our few-shot experiments, as the section is too large to reproduce in full below.
>
> ### "This paper should include comparisons to linear attention models that practitioners are interested in."
>
> Thank you for the suggestion! Yes, we definitely agree that adding linear attention models to our evaluations would be beneficial to readers. We had originally flagged this as future work in **Section 5 (“Discussion”):**
>
> > Sixth, other promising transformer alternatives, such as linear attention models \cite{arora2024based}, hybrid architectures \cite{stripedhyena, lieber2024jamba}, and recurrent models \cite{rwkv}, should be explored in future work that builds upon the framework introduced here.
>
> …but agree that it would be nice to include preliminary results on linear attention models in this work.
>
> Thus, per your suggestion **we are now training a linear attention model.** Given compute constraints on our shared cluster, however, it will take a few more days to complete. We will update our response once the results are available, so thank you for your patience. In the meantime, please let us know if we have resolved your other comments or if you have any additional follow-up questions on our responses!
>
> ### "This paper does not communicate compute budgets, such as wall-times and the hardware used for these sequence lengths. Could a plot communicating performance vs the compute-load be provided to help justify whether these improvements are worth the added compute time."
>
> Thank you for the feedback. Below, we provide a table depicting EHRSHOT performance v. max throughput (i.e. tokens/sec) during inference on a single h100 GPU. Following the benchmarking procedure in the original Mamba paper [1], we have each model generate 128 tokens on a 2048 length prompt across batch sizes ranging from 1 to 128. We then take the max throughput achieved by each model, excluding batch sizes that led to out-of-memory errors. Our results are below:
>
> | Model   |   Context Length |  Max Throughput (tokens/sec)|  Overall AUROC |
> |:-------------|-----------------:|-------------:|----------:|
> | gpt2         |              512 |      744.519 |  0.792351 |
> | gpt2         |             1024 |      743.835 |  0.783288 |
> | gpt2         |             2048 |      741.761 |  0.799709 |
> | gpt2         |             4096 |      746.849 |  0.79148  |
> | llama        |              512 |     1156.82  |  0.78899  |
> | llama        |             1024 |     1157.05  |  0.789721 |
> | llama        |             2048 |     1791.95  |  0.800335 |
> | llama        |             4096 |     1795.43  |  0.796923 |
> | mamba        |             1024 |     2205.27  |  0.784284 |
> | mamba        |             4096 |     2295.82  |  0.80187  |
> | mamba        |             8192 |     2297.37  |  0.798959 |
> | mamba        |            16384 |     2289.14  |  0.807706 |
>
> While it is difficult to give precise wall clock estimates of training time given the nature of the shared compute cluster on which we trained our models, we believe these numbers would be useful to a practitioner hoping to deploy such a model in practice, as models’ life-cycles get increasingly spent on inference rather than training [2].
>
> [1] Gu, Albert, and Tri Dao. "Mamba: Linear-time sequence modeling with selective state spaces." arXiv preprint arXiv:2312.00752 (2023).
> [2] Sardana, Nikhil, et al. "Beyond chinchilla-optimal: Accounting for inference in language model scaling laws." arXiv preprint arXiv:2401.00448 (2023).
>
> (cont...)

---

> ### Author Response · Authors · 2024-11-18
> **Author Response (cont.)**
>
> (...cont)
>
>
> ### "Since you trained an autoregressive sequence model, you could perform a zero-shot evaluation where given the past tokens, you autoregressively generate the future tokens and analyze this generated future trajectory for the binary classification task. This paper does not demonstrate whether these results generalize to the zero-shot. I think that analysis is out of scope for this work but should be mentioned in the limitations."
>
> This is a great suggestion! We have **added zero-shot results** for a subset of EHRSHOT tasks and our strongest models (llama and mamba). The results have been added to **Appendix Section H** and the results table in **Appendix Table 15.**
>
> ### "The most critical improvement to this paper: Include a few-shot/transfer-learning results, the use of mean pooling the last L tokens for all evaluations is not a well supported embedding strategy. Alternatively, you have a zero-shot capable model, why not evaluate it in the zeroshot setting as past works do [1,2,3]? At the moment the limited embedding strategy greatly diminishes the generalizability of your results. If this were resolved, I would significantly increase my rating for the paper."
>
> Thank you for making a great point! As noted above, we have **added few-shot learning results and resolved the embedding strategy issue** you raised, and are working on **adding zero-shot results**. We agree that these changes have greatly strengthened the paper per your comment, but please let us know if we’re missing anything!
>
> 1. **Embedding Strategy:** You are correct that mean pooling over the last L tokens is not well-supported, and this is NOT what we did. Instead, we used the embedding of the last token in the sequence as our representation for each patient in all of our evaluations.  We have fixed the manuscript accordingly in **Appendix Section B.2 (“Evaluation Procedure”)** and **Section 3.2 (“Evaluation”)** (please see comment above for more details).
>
> 2. **Few-Shot Learning:** We have added **Appendix Section G (“Few-shot learning on EHRSHOT”)** which contains few-shot results for all models and context lengths on all EHRSHOT tasks. Specifically, please see **Appendix Tables 11, 12, and 13** as well as **Appendix Figure 10.** The results confirm that for models which benefit from longer contexts (e.g. Mamba and Llama), we see consistent performance gains of longer context across all values of $k$ (i.e. # of shots).
>
> 3. **Zero-Shot Learning:** We have **added zero-shot results** to **Appendix Section H** and the results table in **Appendix Table 15.** They show that the zero-shot capabilities of our models are significantly weaker than their few-shot and finetuned performance.
>
>
> (cont...)

---

> ### Author Response · Authors · 2024-11-18
> **Author Response (cont.)**
>
> (...cont)
>
>
> ### "Add compute time results, and analysis of the diminishing or increasing predictive performance returns as you increase compute times (via increasing sequence lengths) across methods."
>
> Please see our note above about how compute time relates to performance for each model and context length.
>
> ### "A more complete Analysis of the erratic GPT-model Perplexity behavior"
>
> Please see our note above about the GPT-model perplexity behavior, and how switching from absolute positional embeddings to ROPE appeared to mitigate the perplexity spikes experienced when using the original GPT-2 architecture as seen in **Appendix Figure 11**.
>
> ### "Add a comparison with linear attention models"
>
> Please see our comment above about linear attention models -- in brief, we are training a linear attention model and will update our response once that finishes this week. In the interim, we have added a note in the Future Work part of **Section 5 (“Discussion”)** mentioning linear attention models.

---

> ### Author Response · Authors · 2024-11-25
> **Update + Zero-Shot / Linear Attention Results**
>
> Dear Reviewer wRBt,
>
> We hope all is well! May we ask your thoughts on our comments above? To summarize our response, we:
>
> 1. **Fixed** the embedding error that you noted
> 2. Addressed the **GPT perplexity** erraticism
> 3. **Added few shot results** for all models and tasks
> 4. Added **compute budgets** above which demonstrate the improved efficiency of the Mamba-based models
>
> In addition, we have two updates:
>
> 1. We have **added zero-shot results** for a subset of EHRSHOT tasks and our strongest models (llama and mamba). The results have been added to **Appendix Section H** and the results table in **Appendix Table 15.**
> 2. We **provide initial results with a linear attention model below.** We train the **Based** architecture [1] for 2B tokens and evaluate it on all 14 EHRSHOT tasks. Results for each EHRSHOT task group for our 2k and 4k **based** models are shown below (with our top performing model, Mamba-16k, included for reference):
>
> | Model     | Context Length     |   Average AUROC |
> |:----------|:---------|------:|
> | *Task Group: Anticipating Lab Test Results* | |
> | mamba | 16483 | 0.855 |
> | based  | 4096 | 0.852 |
> | based  | 2048 | 0.846 |
> | *Task Group: Operational Outcomes* | |
> | mamba | 16483 | 0.831 |
> | based  | 4096 | 0.82  |
> | based  | 2048 | 0.819 |
> | *Task Group: Assignment of New Diagnoses* | |
> | mamba | 16483 | 0.756 |
> | based  | 2048 | 0.747 |
> | based  | 4096 | 0.736 |
>
> We believe that we have resolved your concerns, especially the _"most critical improvement[s] [that] would significantly increase my rating for the paper."_, but please let us know if you have other questions or concerns. We hope you are able to take into account our response, including additional experiments and clarifications provided, into the final evaluation of our work. If there is no update, we certainly respect your decision, and thank you for the time!
>
> Best,
> The Authors
>
> [1] Arora, Simran, et al. "Simple linear attention language models balance the recall-throughput tradeoff." arXiv preprint arXiv:2402.18668 (2024).

---

> > ### Comment · Reviewer_wRBt · 2024-11-25
> >
> > Hi Authors,
> >
> > Every weakness I found in the original work was addressed, so I significantly increased my rating. I think this paper is a fantastic and insightful contribution in the EHR modeling space, so great work!

---

> > > ### Author Response · Authors · 2024-11-25
> > > **Author Comment**
> > >
> > > Dear Reviewer wRBt,
> > >
> > > Thank you so much for the prompt response and for your detailed review of our work. We greatly appreciated your feedback and believe the manuscript is much stronger thanks to your comments, so thank you again for the time and we appreciate the consideration!

---

### Official Review · Reviewer_ACwG · 2024-11-08

**Soundness:** 3
**Presentation:** 3
**Contribution:** 3
**Rating:** 6
**Confidence:** 4

**Summary:**

This paper explores enhancing EHR data processing models by assessing the impact of context length and model architecture on clinical prediction tasks. Four architectures—GPT, Llama, Hyena, and Mamba—were tested with various context lengths, with Mamba (16k token context) achieving state-of-the-art results in 9 of 14 EHRSHOT benchmark tasks. The study identifies three EHR-specific factors influencing model performance: copy-forwarding (repeated diagnoses), irregular time intervals, and disease progression. Findings show longer contexts generally improve performance, with robustness in EHR data handling, though results vary across architectures, as Hyena showed performance drops beyond 4k tokens. This research advances medical data prediction and offers insights on processing long-sequence data, despite limitations like transformer model computational demands and single-institution data.

**Strengths:**

1. Introduce non-Transformer architectures (such as Mamba) to process medical data.

2. The paper demonstrates the potential of the Mamba model’s long-context capabilities for future clinical applications.

3. The advantage of this paper is that Mamba can perform well with linear complexity.

**Weaknesses:**

1. Both GPT and LLaMA have a maximum context length of only 4096 tokens, so it's not appropriate for the authors to conduct tests with 16K length as a benchmark.

2.The authors should provide additional details about what the 14 tasks in EHRSHOT include to facilitate better comparison.

3. The paper mentions using the standard deviation of time intervals to divide patients into four groups (Q1-Q4). Regarding the groups with the most regular time intervals and the most irregular time intervals, the standards vary across different diseases, and testing of time intervals should be conducted according to specific diseases.

4. The paper mentions achieving an AUROC of 0.807, but it's confusing that the specific content of the 14 tasks is not listed.

**Questions:**

Since the author's testing only showed good results with small Mamba models, it's uncertain whether larger Mamba models would perform better than traditional GPT.

---

> ### Author Response · Authors · 2024-11-18
> **Author Response**
>
> Thank you so much for your review of our work. We appreciate your recognition of our paper's contribution in introducing non-transformer architectures, such as Mamba, for EHR modeling tasks and the potential for such long-context models to positively impact future clinical applications. We appreciate your time and thoughtful feedback, and in addition to our **Overall Response** above, provide a response to each point raised in your review below.
>
> In particular, we appreciate your questions regarding the EHRSHOT dataset -- we have added substantially more information to the manuscript in response -- as well as your questions about model/context size. Please let us know if you have any further questions!
>
> ### "Both GPT and LLaMA have a maximum context length of only 4096 tokens, so it's not appropriate for the authors to conduct tests with 16K length as a benchmark."
>
> Thank you for the opportunity to address your concern. We agree that benchmarking models with different maximum context lengths has its limitations. For the purposes of fair benchmarking, we did run all models at the smaller context lengths (e.g. 1k, 4k), so there are head-to-head comparisons between the different models in our results.
>
> When we tried training GPT and Llama at 16k tokens we received out-of-memory errors. Given our focus on practicality -- and the fact that the typical hospital has even fewer compute resources than we had access to -- we’d argue that these models’ inability to efficiently scale to longer contexts represents an intrinsic advantage of other architectures, and this is reflected in the size of context we report. In the future, we would like to train longer context transformer models on better hardware, and we believe this represents exciting future work. To echo your comment, we have added the following note to the Limitations subsection of **Section 5 (“Discussion”).**
>
> > While our findings highlight the potential for long-context models to successfully model EHR data, several limitations should be considered. Firstly, we could not evaluate transformer-based models at context lengths beyond 4k tokens due to limited computational resources.
>
> ### "The authors should provide additional details about what the 14 tasks in EHRSHOT include to facilitate better comparison."
>
> Thank you for your feedback, this is an excellent suggestion. As mentioned in our **Overall Response** under _“Additional clarity about the 14 EHRSHOT evaluation tasks”,_ to resolve your comment we have added 4 new pages of detail covering the 14 EHRSHOT tasks to **Appendix Section B**. We believe this has greatly strengthened the paper, as changes made include:
> 1. We now provide a list of 14 EHRSHOT tasks and their label type, prediction time, and time horizon (**Appendix Table 4**)
> 2. We added precise definitions of all 14 tasks, including the EHR codes corresponding to the labeling functions used for identifying each lab test and new diagnosis (**Appendix Section B.1**)
> 3. We now provide the sample sizes (# of patients and # of labels) and class balance (# of positive/negative labels) for each split (train/test/val) for all 14 tasks (**Appendix Table 5**)
> 4. We added plots showing the number of raw clinical events and tokens for each label each split (train/test/val) for all 14 tasks (**Appendix Figure 8**)
>
> We have also added the following text to our main manuscript in **Section 3.2 (“Evaluation”)** to provide a high-level description of these tasks and point the reader to our new Appendix section:
>
> > We use the remaining 14 tasks from the EHRSHOT benchmark for our evaluations, which are broadly grouped into three categories: Operational Outcomes includes predicting ICU Transfer, 30-day Readmission, and Long Length-of-Stay; Anticipating Lab Test Results involves predicting if a thrombocytopenia, hyperkalemia, hypoglycemia, hyponatremia, or anemia lab will be abnormal; and Assignment of New Diagnoses requires predicting whether a patient will get a new diagnosis of hypertension, hyperlipidemia, pancreatic cancer, celiac disease, or lupus within the next year. For additional details on all 14 tasks, including precise definitions, label counts, statistics on the number of tokens per patient, and evaluation methodology, please see Appendix Section B.
>
> (cont...)

---

> ### Author Response · Authors · 2024-11-18
> **Author Response (cont.)**
>
> (...cont)
>
>
> ### "The paper mentions using the standard deviation of time intervals to divide patients into four groups (Q1-Q4). Regarding the groups with the most regular time intervals and the most irregular time intervals, the standards vary across different diseases, and testing of time intervals should be conducted according to specific diseases."
>
> Thank you for your comment. We did not conduct this evaluation in our manuscript given the fact that most patients in our evaluation dataset (EHRSHOT) have several comorbidities, and thus clustering patients by “specific diseases” was beyond the scope of our investigation into long context EHR FMs. However, we agree that this could be a very interesting direction for future work, and we appreciate your bringing this suggestion to our attention! We would be excited to see research that builds on the 3 EHR-specific metrics we propose -- irregularity, copy-forwarding, and disease progression -- by evaluating how disease status impacts each metric.
>
> In response, we have added the following note in the Future Work section of **Section 5 (“Discussion”)** calling out this limitation in our approach:
>
> > Moreover, disease status may confound the regularity of time intervals between events, which future work could explore by stratifying results based on specific disease trajectories.
>
> ### "The paper mentions achieving an AUROC of 0.807, but it's confusing that the specific content of the 14 tasks is not listed."
>
> Please see comment above regarding the specific content of the 14 EHRSHOT tasks, which we now provide in **Appendix Section B** per your feedback.
>
> ### "Since the author's testing only showed good results with small Mamba models, it's uncertain whether larger Mamba models would perform better than traditional GPT."
>
> Thank you for your comment. We agree that model size is certainly an important consideration, and thus have added the following text to the Future Work subsection of **Section 5 (“Discussion”)** to echo the point you make:
>
> > Second, model sizes were kept consistent across architectures to isolate the impact of context length. Preliminary findings suggest smaller Mamba models with 16k tokens perform well, which may reduce the need for larger models unsuitable for resource-constrained settings. Future work should quantify the impact of model size on performance..
>
> As our primary focus of this work was isolating the impact of context length, we purposely maintained similar (small) model sizes (~110M params) across all architectures in our experiments, as detailed in **Appendix Table 6**. Our results show that even smaller Mamba models with a 16k context length outperform other methods (i.e. achieving SOTA on the EHRSHOT benchmark), which could mitigate the need to scale up model size. We believe that this is an interesting finding in and of itself -- that smaller models are able to perform well on these tasks -- and that these results offer practical benefits to researchers seeking to deploy such models in resource-constrained on-premise hospital environments.
>
> Finally, we would note that other research has shown scaling laws up to 1B parameters indicating that Mamba can exceed GPT performance, and thus we believe that similar findings would hold if we had the compute resources to scale up our models to that size. [1]
>
> [1] Gu, Albert, and Tri Dao. "Mamba: Linear-time sequence modeling with selective state spaces." arXiv preprint arXiv:2312.00752 (2023).

---

> ### Author Response · Authors · 2024-11-25
> **Update**
>
> Dear Reviewer ACwG,
>
> We hope all is well! May we ask your thoughts on our comments above? To summarize our response, we:
>
> 1. **Added ~4 new pages of details on EHRSHOT** to answer your questions regarding the 14 tasks. Please see the additional paragraph in **Section 3.2** of the main manuscript, as well as the newly added **Appendix Section B** and **Appendix Table 4 and 5** and **Appendix Figure 8**.
> 2. **Provided results across all models at several smaller context lengths for fair comparisons** (e.g. see the results @ 4k in **Figure 1b**, where Mamba-4k is still the superior model), and explained why we limit GPT / Llama to a max context length of 4k tokens (i.e. their quadratic computational complexity with context length prevents these models from efficiently scaling to longer contexts).
> 3. **Clarified** why we evaluated **Mamba at a smaller size** -- i.e. we wanted to isolate the impact of context length, so we needed to control for \# of parameters across models. Other work (e.g. Gu and Dao 2023) has already evaluated scaling Mamba models and shown competitive performance with GPT at 1B+ parameters.
> 4. **Added an explanation** as to why standard deviations were calculated across all patients rather than disease subgroups.
>
> We believe that we have resolved your concerns, so please let us know if you have any other questions. We hope you are able to take into account our response, including additional experiments and clarifications provided, into the final evaluation of our work. If there is no update, we certainly respect your decision, and thank you for the time!
>
> Best,
> The Authors

---

> > ### Comment · Reviewer_ACwG · 2024-12-02
> > **Reply to author**
> >
> > Dear author, thank you for answering my questions and thoroughly addressing the detailed issues while providing relevant experimental results.

---

### Author Response · Authors · 2024-11-18
**Author Overall Response**

We would like to thank all our reviewers for taking the time to review and provide detailed feedback on our manuscript. We deeply appreciate their time, and addressing their comments has greatly improved the manuscript.

This work is the **first systematic evaluation of context length on modeling EHR data** across transformer (Llama, GPT) and subquadratic (Mamba, Hyena) architectures. Whereas most prior work (**Table 1**) considers models at a single context length <= 1k tokens, we evaluate models up to 16k tokens and achieve **SOTA results** on the EHRSHOT clinical prediction benchmark with Mamba-16k. We also **propose 3 metrics** for sequential data -- frequency of copy-forwarding, irregularity of inter-token time intervals, and the increasing complexity of tokens over time -- and show that EHR data exhibit all 3 properties to a larger degree than natural language. We empirically show that each property correlates negatively with model performance, highlighting new avenues for research in modeling sequential data.

We are glad reviewers appreciated our work for its **“solid technical execution and experimental design”** (uVBS) towards providing **“valuable insights into using long-context models to model EHR data”** (TDwP). Reviewers appreciated our contribution to the literature in **“introduc[ing] non-transformer architectures (such as Mamba) to process medical data”** (ACwG) and demonstrating the **“effectiveness of long-context models on EHR data”** (UYtp) towards **“future clinical applications”** (ACwG). Reviewers also found our analysis which **“identified and quantified” (UYtp) three previously underexplored EHR-specific properties to be **“convincing”** (TDwP) and **“helpful for future work to build better foundation models for EHR”** (TDwP) in a **“generally…rich technical/experimental paper”** (uVBS). Finally, reviewers appreciated the open source nature of our work and mentioned that our release of **“pre-trained and fine-tuned foundational models will benefit the community”** (TDwP).

In addition to the strengths noted above, the reviewers provided constructive feedback which we have implemented and which has significantly improved the manuscript. We thank them for their comments. In response, we have:

1. Uploaded an **updated PDF** of our manuscript with **changes implemented in blue text.**
2. We provide an **Overall Response** below to common feedback across reviewers, namely questions around (1) additional clarity on the 14 evaluation tasks, (2) EHRSHOT stratification results, (3) comparisons to prior SOTA model CLMBR-t-base, (4) tokenization strategies for dealing with time, and (5) novelty.
3. We have added **detailed line-by-line responses to each reviewer** under their respective sections.


## Common Themes Across Reviewers


### 1. Additional clarity about the 14 EHRSHOT evaluation tasks

Two reviewers (ACwG, TDwP) asked for additional details on the 14 tasks in the EHRSHOT benchmark used for evaluation. We agree with their suggestions, and thank the reviewers for bringing this to our attention.

In response to these comments, we have **added roughly 4 pages of additional detail** covering the 14 EHRSHOT tasks to **Appendix Section B**.  While we can’t reproduce it fully here, improvements that we made in response to reviewers include:

1. We now provide a list of 14 EHRSHOT tasks and their label type, prediction time, and time horizon (**Appendix Table 4**)
2. We added precise definitions of all 14 tasks, including the EHR codes corresponding to the labeling functions used for identifying each lab test and new diagnosis (**Appendix Section B.1**)
3. As asked by TDwP, we now provide the sample sizes (# of patients and # of labels) and class balance (# of positive/negative labels) for each split (train/test/val) for all 14 tasks (**Appendix Table 5**)
4. We added plots showing the number of raw clinical events and tokens for each label each split (train/test/val) for all 14 tasks (**Appendix Figure 6**)

We have also added the following text to our main manuscript in **Section 3.2 (“Evaluation”)** to provide a high-level description of these tasks and point the reader to our new Appendix section:

>We use the remaining 14 tasks from the EHRSHOT benchmark for our evaluations, which are broadly grouped into three categories: Operational Outcomes includes predicting ICU Transfer, 30-day Readmission, and Long Length-of-Stay; Anticipating Lab Test Results involves predicting if a thrombocytopenia, hyperkalemia, hypoglycemia, hyponatremia, or anemia lab will be abnormal; and Assignment of New Diagnoses requires predicting whether a patient will get a new diagnosis of hypertension, hyperlipidemia, pancreatic cancer, celiac disease, or lupus within the next year. For additional details on all 14 tasks, including precise definitions, label counts, statistics on the number of tokens per patient, and evaluation methodology, please see Appendix Section B.

(cont...)

---

> ### Author Response · Authors · 2024-11-18
> **Response (cont.)**
>
> (...cont)
>
> ### 2. EHRSHOT stratification results
>
> Please note that in response to a reviewer’s comment, we found a bug in our processing pipeline for **Table 2**. The conclusions remain the same, but numbers have been slightly updated. Separately, we have also added results for CLMBR-t-base to **Table 2**, and added **Appendix Table 14** with full results for all models.
>
> ### 3. How we evaluated the Prior SOTA model (CLMBR-t-base)
>
> Two reviewers (UYtp, TDwP) raised questions about the prior SOTA model on the EHRSHOT benchmark, CLMBR-t-base. We thank them both for raising their concerns, and have worked to improve the clarity of our manuscript in this regard.
>
> First, in response to UYtp’s comment that “the context length of this model is not discussed,” we have updated **Table 1** to accurately reflect the context length of this model, which was 512 tokens. Please note that CLMBR-t-base was not trained with variable context lengths -- as noted in **Section 2.3 (“Related Work”)**, this form of fixed context length evaluation was a key limitation of prior work which we address via our work. We have also edited the following text in **Section 4.1** to make this clearer in the main manuscript:
>
> > The dotted purple line is the mean AUROC (0.777) achieved by the best overall prior model, CLMBR-t-base \citep{wornow2023ehrshot}, which had a context length of 512 tokens.
>
> In response to TDwP’s comment that “there is no evaluation of CLMBR-T-base”, we'd note that we do already include the performance of the publicly available CLMBR-t-base model in all of our evaluations, and follow the same evaluation procedure for all models that was used in the original EHRSHOT benchmark for CLMBR-t-base. In **Figure 1b**, for example, CLMBR-t-base’s performance is shown as the purple dashed line marked “Prior SOTA”.
>
> This should also help resolve UYtp’s comment that “foundation models designed for EHR, such as [1] and [2], are not included”, as citation [1] references the paper _Guo, L. L., Steinberg, E., Fleming, S. L., Posada, J., Lemmon, J., Pfohl, S. R., ... & Sung, L. (2023). EHR foundation models improve robustness in the presence of temporal distribution shift. Scientific Reports, 13(1), 3767_ which uses the same exact CLMBR-style model that we use as the “Prior SOTA” baseline for all of our evaluations.
>
> These points about making this comparison clearer between our models and CLMBR-t-base is well taken, and so to provide a more detailed analysis we have **added several pages of results** in the form of **Appendix Tables 7, 8, 9, and 10** which compare the performance of all of our models against CLMBR-t-base across all tasks.
>
>
>
> ### 4. Tokenization strategies for dealing with time
>
> Two reviewers (TDwP, UYtp) asked about tokenization strategies that might be better suited for encoding time information. While we preserved the same tokenization strategy as EHRSHOT in our initial results for the fairest possible comparison, we agree that there are other strategies in the literature that merit consideration.
>
> In response, we retrained all of our models using the state-of-the-art “Artificial Time Token (ATT)” strategy to encode time intervals, inspired by CEHR-BERT [1] and other recently published EHR FMs [2,3]. We provide full details on this tokenization strategy in **Appendix Section D (“Tokenization”)**. In brief, we inject tokens into the patient’s timeline whenever there is a gap of more than one day between tokens. We define new [DAY X], [WEEK X], [MONTH X], and [LONG TERM] tokens (where X can take on various integer values) to explicitly represent intervals of time between tokens. This tokenization strategy enhancement directly embeds temporal patterns within the token sequence, providing contextual information about the intervals between clinical events.
>
> We report our results in **Appendix Figure 12**. As predicted by UYtp (“the gain may be minimal overall”), we do not see performance gains when encoding time information via this tokenization strategy. In fact, while the overall trends stay the same, we actually see slightly worse performance across most models with the inclusion of ATT tokens. However, we acknowledge that there are many possible tokenization strategies that could be used, and leave such investigations to future work. We hope that our manuscript serves to motivate such efforts through our introduction of metrics such as inter-token irregularity to EHR data.
>
> We have added the following to **Section 4.1 (“Results”)** to explicitly call this out in the manuscript as well:
>
> > To more explicitly model the passage of time, we also train a version of our models using the Artificial Time Tokens (ATT) technique originally proposed in CEHR-BERT \cite{cehr-bert}. However, as shown in Appendix Figure \ref{fig:appendix_att_tokens}, we see slightly worse performance with this tokenization strategy.
>
> (cont...)

---

> > ### Author Response · Authors · 2024-11-18
> > **Response (cont.)**
> >
> > (...cont)
> >
> > [1] Pang, Chao, et al. "CEHR-BERT: Incorporating temporal information from structured EHR data to improve prediction tasks." Machine Learning for Health. PMLR, 2021.
> > [2] Renc, Pawel, et al. "Zero shot health trajectory prediction using transformer." NPJ Digital Medicine 7.1 (2024): 256.
> > [3] Pang, Chao, et al. "CEHR-GPT: Generating electronic health records with chronological patient timelines." arXiv preprint arXiv:2402.04400 (2024).
> >
> > ### 5. Novelty
> >
> > The question of novelty (uVBS) was raised, as we do not introduce a new architecture in this work. Two reviewers (UYtp, TDwP) also had questions about how our work relates to EHRMamba [1]. We have three responses to these questions.
> >
> > First, we believe that our work does bring several novel contributions to the field:
> >
> > 1. We **introduce and quantify three unique characteristics of EHR sequential data** -- copy-forwarding, irregularity, and token complexity progression -- that have been underexplored in the NLP literature. We present novel empirical results demonstrating how these characteristics impact model performance across context lengths.
> > 2. We are the **first to train and evaluate long context models for EHR data across multiple context lengths** and including both transformer and subquadratic architectures. We achieve SOTA results across 14 diverse clinical prediction tasks.
> > 3. Our work is **distinct from the closest prior work, EHRMamba, in several key ways**. (a) We extend Mamba to multiple context lengths of 1k, 4k, 8k, and 16k tokens, in contrast to the single 2k context length tested in EHRMamba as detailed in **Table 1**. (b) We evaluate Mamba on 14 diverse clinical prediction tasks spanning operational outcomes, lab result prediction, and new diagnosis assignment. In contrast, EHRMamba assesses Mamba's performance on only 6 tasks. (c) Our analysis of three EHR-specific properties is entirely novel in **Table 2**. We stratify our results by each property and measure model robustness within these subpopulations. In contrast, EHRMamba does not assess how models perform across subgroups with varying levels of data complexity. (d) We conduct few-shot evaluations of our models (as low as $k = 8$) in **Appendix Figure 10**, in contrast to EHRMamba which relies on 10k’s of training examples for fine tuning.
> > 4. We will (after the anonymization period) be the **first to publicly release the open weights of Llama, Mamba, and Hyena models (16 total) pretrained on real-world EHR data on HuggingFace** for open science and reproducibility. This is particularly impactful in a domain such as healthcare in which publishing the weights of models trained on real-world EHR data is rare.
> >
> > Second, we do not believe that introducing a novel architecture is necessary to make a positive contribution to the literature. The field of ML for healthcare is currently suffering from a **"reproducibility crisis"** but eager to deploy FM techniques. [2] Without rigorous understanding of their performance, however, new FM architectures will never be adapted in healthcare (e.g. see **Table 1** for the low adoption of non-transformer models within the EHR FM literature). Thus, our experiments can serve as a valuable bridge.
> >
> > Third, our work contributes to a larger trend in general LLM research towards **data-centric model development and evaluation**, going beyond architectures / training objectives and instead gaining insights via the construction and examination of domain-specific datasets and metrics with meaningful downstream evaluations. [3,4,5] In a fast-moving field like LLMs, it is difficult to distinguish noise from signal and identify which architectures work, for which domains, and under which configurations. We believe that our work offers useful insights for ML researchers hoping to apply sequence models to non-natural-language modalities (e.g. EHRs) where domain-specific properties can skew model performance.
> >
> > [1] Fallahpour, Adibvafa, et al. "EHRMamba: Towards Generalizable and Scalable Foundation Models for Electronic Health Records." arXiv preprint arXiv:2405.14567 (2024).
> > [2] McDermott, Matthew BA, et al. "Reproducibility in machine learning for health research: Still a ways to go." Science Translational Medicine 13.586 (2021): eabb1655.
> > [3] Liu, Yang, et al. "Datasets for large language models: A comprehensive survey." arXiv preprint arXiv:2402.18041 (2024).
> > [4] Guo, Zishan, et al. "Evaluating large language models: A comprehensive survey." arXiv preprint arXiv:2310.19736 (2023).
> > [5] Zha, Daochen, et al. "Data-centric artificial intelligence: A survey." arXiv preprint arXiv:2303.10158 (2023).

---

### Meta-Review · Area_Chair_qrJB · 2024-12-19

**Metareview:**

This is an interesting systematic evaluation and experimental analytics paper on foundation models (FMs) for EHRs. The paper is very comprehensive and could be impactful for the field of AI for EHR. The authors benchmarked both transformer-based (GPT and Llama) and subquadratic architectures (Mamba and Hyena) on EHR data. The authors evaluated and discussed the increased performance with longer contexts via comprehensive experiments. The authors also measured 3 EHR-specific properties, copy-forwarding, irregular inter-token time intervals, and disease progression, impact models at different context lengths. In sum: the study is EHR application-driven, the experimental results are comprehensive, the discussions are insightful, and the supplementary material contains the implementation code. After the rebuttal stage, most reviewers supported this paper.

PC/SAC/AC and all reviewers will monitor this submission to ensure the reproductivity of this work - the authors mentioned they will release all of the model checkpoints and code. However, the supplementary material doesn't contain the models. 4 FMs on EHRs based on GPT, Llama, Mamba, Hyena pre-trained on their 2.5M patient data are required by the ICLR community. It will also get good citations for this paper.

**Additional Comments On Reviewer Discussion:**

The authors have sufficiently addressed most comments from all the reviewers. Most reviewers have increased their scores accordingly during the discussions. Reviewer uVBS has not responded to the authors during the discussions. However, I think his/her questions have already been partially addressed by the authors.

---

> ### Public Comment · ~Michael_Wornow1 · 2025-02-07
> **Model release**
>
> Hi,
>
> Thank you for the comment! Our models and code can be found at this Github: https://github.com/som-shahlab/long_context_ehrs

---

### Decision · Program_Chairs · 2025-01-22

Accept (Poster)